# Molecular insights into the effect of 1,6-hexanediol on FUS phase separation

Tongyin Zheng [1,8], Noah Wake [2,8], Shuo-Lin Weng [3,8], Theodora Myrto Perdikari[4,8], Anastasia C Murthy[5], Jeetain Mittal [3,6,7✉] & Nicolas L Fawzi [1✉]

## Abstract

The alkanediol 1,6-hexanediol has been widely used to dissolve liquid–liquid phase-separated condensates in cells and in vitro, but the details of how it perturbs the molecular interactions underlying liquid–liquid assembly remain unclear. In this study we use a combination of microscopy, nuclear magnetic resonance (NMR) spectroscopy, molecular simulation, and biochemical assays to probe how alkanediols suppress phase separation and why certain isomers are more effective. We show that alkanediols of different lengths and configurations are all capable of disrupting phase separation of the RNA-binding protein Fused in Sarcoma (FUS), although potency varies depending on both geometry and hydrophobicity, which we measure directly. Alkanediols induce a shared pattern of changes to the chemical environment of the protein, to different extents depending on the specific compound. Furthermore, we use lysozyme as a model globular protein to demonstrate that alkanediols decrease the proteins' thermal stability, which is consistent with the view that they disrupt phase separation driven by hydrophobic groups. Conversely, 1,6-hexanediol does not disrupt charge-mediated phase separation, such as FUS RGG-RNA and poly-lysine/poly-aspartic acid condensates. All-atom simulations show that hydroxyl groups in alkanediols mediate interactions with the protein backbone and polar amino acid side chains, while the aliphatic chain allows contact with hydrophobic and aromatic residues, providing a molecular picture of how amphiphilic interactions disrupt FUS phase separation.

**Keywords** NMR Spectroscopy; Molecular Simulation; Hydrophobicity; Protein Folding; RNA
**Subject Categories** Pharmacology & Drug Discovery; Proteomics; Structural Biology

## Introduction

Cellular biochemistry is spatiotemporally tuned by intracellular compartments known as biomolecular condensates. Despite lacking a phospholipid bilayer, condensates concentrate specific but heterogeneous biomolecules through a process called phase separation (Banani et al, 2017; Shin and Brangwynne, 2017; Yang et al, 2020). The formation, dissolution, and localization of membraneless puncta are controlled by a plethora of factors, such as the presence of RNA species (Roden and Gladfelter, 2021; Shin and Brangwynne, 2017) and post-translational modifications within intrinsically disordered regions (IDRs) (Martin and Mittag, 2018; Snead and Gladfelter, 2019). Important studies on the nuclear pore complex (NPC), a multiprotein structure embedded in the nuclear envelope that serves as a permeability barrier potentially by phase separation of phenylalanine/glycine repeat domains, pioneered the use of alkanethiols to disrupt disordered domain interactions and interfere with NPC functions (Ribbeck and Gorlich, 2002). Furthermore, alkanediols were also shown to cause the dissociation of nuclear pore complex components (Shulga and Goldfarb, 2003). More recently, cytoplasmic and nucleoplasmic condensates such as stress granules (SGs) (Wolozin and Ivanov, 2019) have also been shown to be sensitive to dissolution by alkanediols (Kroschwald et al, 2015; Wheeler et al, 2016). Moreover, alkanediols have been employed to assess the reversibility of liquid nuclear and cytoplasmic puncta (Kroschwald et al, 2015). For example, 1,6-hexanediol, 1,2-cyclohexanediol, and 1,5-pentanediol are capable of dissolving nucleopores (Jaggi et al, 2003; Ribbeck and Gorlich, 2002), while 1,6-hexanediol also disrupts cytoplasmic granules (Tulpule et al, 2021), potentially by disrupting weak hydrophobic interactions. In yeast, when 1,6-hexanediol was used in tandem with compounds that increase cell permeability like digitonin, P granules dissolved but stress granules with a more solid-like character remained intact (Kroschwald et al, 2017). In mammalian cells, 1,6-hexanediol has been used to probe the liquidity of condensates including stress granules, though prolonged exposure is cytotoxic and generated abnormal cell morphologies that complicate these analyses (Wheeler et al, 2016). Although these studies provide compelling evidence that

[1]Department of Molecular Biology, Cell Biology & Biochemistry and Robert J. and Nancy D. Carney Institute for Brain Science, Brown University, Providence, RI, USA. [2]Therapeutic Sciences Graduate Program, Brown University, Providence, RI, USA. [3]Department of Chemistry, Texas A&M University, College Station, TX, USA. [4]Center for Biomedical Engineering, Brown University, Providence, RI, USA. [5]Molecular Biology, Cell Biology & Biochemistry Graduate Program, Brown University, Providence, RI, USA. [6]Artie McFerrin Department of Chemical Engineering, Texas A&M University, College Station, TX, USA. [7]Interdisciplinary Graduate Program in Genetics and Genomics, Texas A&M University, College Station, TX, USA. [8]These authors contributed equally: Tongyin Zheng, Noah Wake, Shuo-Lin Weng, Theodora Myrto Perdikari. ✉E-mail: jeetain@tamu.edu; nicolas_fawzi@brown.edu

hexanediols and similar compounds can be used to assess the physical properties of in vitro reconstituted droplets and cellular condensates, the details of how these compounds perturb the molecular interactions underlying liquid-like assembly remain unclear.

FUS (Fused in Sarcoma) is an RNA-binding protein whose phase separation has been extensively characterized. FUS has been in the biophysical spotlight due to its disease-causing mutations (Naumann et al, 2018; Patel et al, 2015) and its interactions with nucleic acids (Daigle et al, 2013; Loughlin et al, 2019; Sama et al, 2014), poly(ADP ribose) (Altmeyer et al, 2015; Rhine et al, 2022) and transcription factors (Owen et al, 2021). FUS phase separation is also modulated by phosphorylation (Monahan et al, 2017), arginine methylation (Hofweber et al, 2018) and N-terminal acetylation (Bock et al, 2021). Under physiological conditions, FUS can shuttle between the nucleus and the cytoplasm and form interactions with other proteins through its serine-glutamine-tyrosine-glycine (SQYG) rich N-terminal disordered region, arginine-glycine (RGG) motifs (Chong et al, 2018) and globular RNA binding domains (Deng et al, 2014). The aberrant function of membraneless puncta containing FUS has been associated with pathology, particularly in neurodegenerative disease and cancer (Alberti and Dormann, 2019; Boija et al, 2021; Ryan and Fawzi, 2019; Trnka et al, 2021). This has led to an emerging area of research focused on developing potential therapeutics targeting protein phase separation (Babinchak et al, 2020; Klein et al, 2020; Mitrea et al, 2022; Schmidt et al, 2022). Although the molecular interactions holding together the condensed phase of FUS, such as hydrophobic, $sp^2/\pi$ contacts, and hydrogen bonds have been studied in detail (Murthy et al, 2019; Murthy et al, 2021; Wake et al, 2025; Zheng et al, 2020), little is known at the atomic level about how non-covalent interactions with alkanediols perturb the phase behavior of FUS. Some have suggested that 1,6-hexanediol acts as "mini-detergent" that disrupts hydrophobic contacts (Hedtfeld et al, 2024; Ribbeck and Gorlich, 2002) while others have hypothesized that particular alkanediol-protein interaction geometries explain observations that some alkanediol isomers are more potent at inhibiting phase separation than others (Gu et al, 2023). To fill this gap, we employ microscopy, solution-state NMR, molecular dynamics (MD) simulations, and biochemical assays to unravel the molecular-level details of the impact of alkanediols on the phase separation, structure, and motions of FUS. In particular, we seek to probe the hypothesis that alkanediols act as hydrophobic disruptors of phase separation and clarify why some alkanediols are more potent than others.

# Results

## Effect of alkanediols on FUS LC phase separation

Alkanediols such as 1,6-hexanediol (1,6-HD), 2,5-hexanediol (2,5-HD), 1,4-butanediol (1,4-BD) and 1,5-pentanediol (1,5-PD) have been extensively used as FUS hydrogel melting (Lin et al, 2016) and phase separation prevention agents (Berkeley et al, 2021; Li et al, 2021; Liu et al, 2021). We sought to determine how these alkanediols as well as 1,2-hexanediol (1,2-HD) and 1,2-cyclohexanediol (1,2-CHD) alter the capacity of purified protein to form liquid droplets in vitro using the isolated low-complexity (LC)

domain (residues 1–163) of FUS as a model. First, we tested the impact of 1,6-HD on FUS LC phase separation (0% to 5%) using differential interference contrast (DIC) (Fig. 1A) and fluorescence microscopy (Appendix Fig. S1A). FUS LC droplet area linearly decreases between 0% and 5% concentrations of 1,6-HD (Fig. 1B). To complement these microscopy assays, we measure the protein saturation concentration ($C_{sat}$) by centrifuging phase separated samples to pellet FUS LC droplets in the presence of different concentrations of 1,6-HD (Fig. 1C) and then quantified the protein remaining in the supernatant by UV absorbance (Appendix Fig. S1B), which also show a linear dependence on 1,6-HD concentration. However, by 5% 1,6-HD, no FUS LC droplets were present. Therefore, to quantitatively assess the effects of 1,6-HD at higher concentrations up to the 10% 1,6-HD concentration sometimes used in cellular studies, we used the longer FUS LC-RGG1 segment, which has a decreased $C_{sat}$ (i.e., an increased ability to undergo phase separation) compared to FUS LC (Wake et al, 2025). Similarly, we observed that FUS LC-RGG1 $C_{sat}$ linearly decreases across the 0% to 10% 1,6-HD concentration range (Appendix Fig. S1C).

To compare the effect of various alkanediols, we imaged FUS LC droplets after treatment with an intermediate amount (2%) of each alkanediol (an amount suggested to be in the optimal range for cell-based experiments (Klein et al, 2020; Sabari et al, 2018; Shi et al, 2021)) using DIC as well as fluorescence (Fig. 1D; Appendix Fig. S1D). In separate samples, we quantified FUS LC phase separation by measuring $C_{sat}$ in the presence or absence of each alkanediol at 1% or 2% concentration (Fig. 1E). The impact on phase separation as measured by quantitative microscopy and $C_{sat}$ measurements are highly correlated (Fig. 1F), confirming the quantitative differences in disruption of FUS LC phase separation between the various alkanediols. Compared to the rest of the alkanediol series, 2,5-HD and 1,4-BD resulted in lower concentrations of protein remaining in the dispersed phase and higher droplet area fractions (Fig. 1F), suggesting that these alkanediols were the least effective in disrupting the condensed phase. 1,5-PD reduced the extent of phase separation to an intermediate level, less than 1,6-HD and 1,2-HD. Together, the data suggest that alkanediols with different molecular structures all disrupt phase separation but to differing extents.

## Phase separation driven by charge-charge interactions is insensitive to alkanediols

In FUS, the N-terminal LC is followed by multiple domains involved in RNA-binding—the RNA-recognition motif (RRM) and zinc finger (ZnF) domains, flanked by interdomain linkers containing disordered RGG boxes composed of closely spaced arginine-glycine-glycine repeats and aromatic residues (e.g., RGG[Y/F]RGG) (Thandapani et al, 2013) that contribute to phase separation via protein-protein (Wang et al, 2018) and protein-RNA interactions (Chong et al, 2018). Previous studies have shown that the isolated RRM and ZnF domains of FUS bind RNA rather weakly, but the inclusion of RGG2 domain in RRM-RGG2 results in a many-fold increase in RNA-binding affinity, likely due to the presence of positively charged arginines that are well-known for promoting multivalency in disordered protein-RNA complexes (Loughlin et al, 2019; Ozdilek et al, 2017). Inspired by recent studies on the effect of hexanediol in the organization of RNA granules (Fuller et al, 2020) and chromatin (Itoh et al, 2021; Liu

et al, 2021; Shi et al, 2021; Ulianov et al, 2021), as well as reports on the action of chemotherapeutics on nucleolar proteins and ribosomal RNA (rRNA) synthesis (Sutton and DeRose, 2021), we studied the potential of alkanediols to inhibit phase separation of FUS in the presence of RNA as a model for the types of protein-RNA interactions that contribute to biomolecular condensates formed in cells. As in previous studies (Burke et al, 2015; Monahan et al, 2017), we imaged full-length (FL) FUS after addition of TEV protease to cleave an N-terminal maltose-binding protein (MBP) tag that prevents phase separation (Fig. 2A; Appendix Fig. S2A). As expected, alkanediols disrupted the phase separation of FL FUS (Fig. 2A,B). As we showed previously, addition of polyuridylic acid (polyU RNA) enhances FUS phase separation. However, we found that phase separation is only modestly decreased and still occurs even with the addition of 5% alkanediols, and the small differences in impacts on phase separation do not appear to correlate with the "strength" of the alkanediol in disrupting FUS phase separation without RNA. In other words, 1,6-HD and other alkanediols do not fully dissolve full-length FUS condensates with RNA.

Having already shown that RNA does not interact with FUS LC (Burke et al, 2015), we focused on an RGG domain of FUS as a model for the disordered, charged domains of FUS that may interact with RNA. FUS RGG3, which does not undergo phase separation by itself at these conditions, readily forms droplets upon the addition of RNA (Appendix Fig. S3). Notably, the presence of alkanediols did not alter this behavior. For three different types of RNA (polyA, torula yeast RNA extract, and polyU) that include homopolymeric unstructured RNA and physiologically structured RNAs, we quantified the $C_{sat}$ of FUS RGG3 in the phase-separated RGG3-RNA mixture by measuring the absorbance at both 260 and 280 nm to separate the contribution of protein and RNA to the readings. We found that $C_{sat}$ values were nearly unchanged in the presence of 1,6-HD (Fig. 2C). We also tested the impact of 2.5% 1,6-hexanediol on mixtures of FUS RGG3 and RNAs across various RNA-to-protein ratios and observed no significant disruption of phase separation at RNA-to-protein ratios up to 1:1 (Fig. 2D).

Given that the co-phase separation of RGG3-RNA may rely predominantly on charge-charge interactions, we asked whether

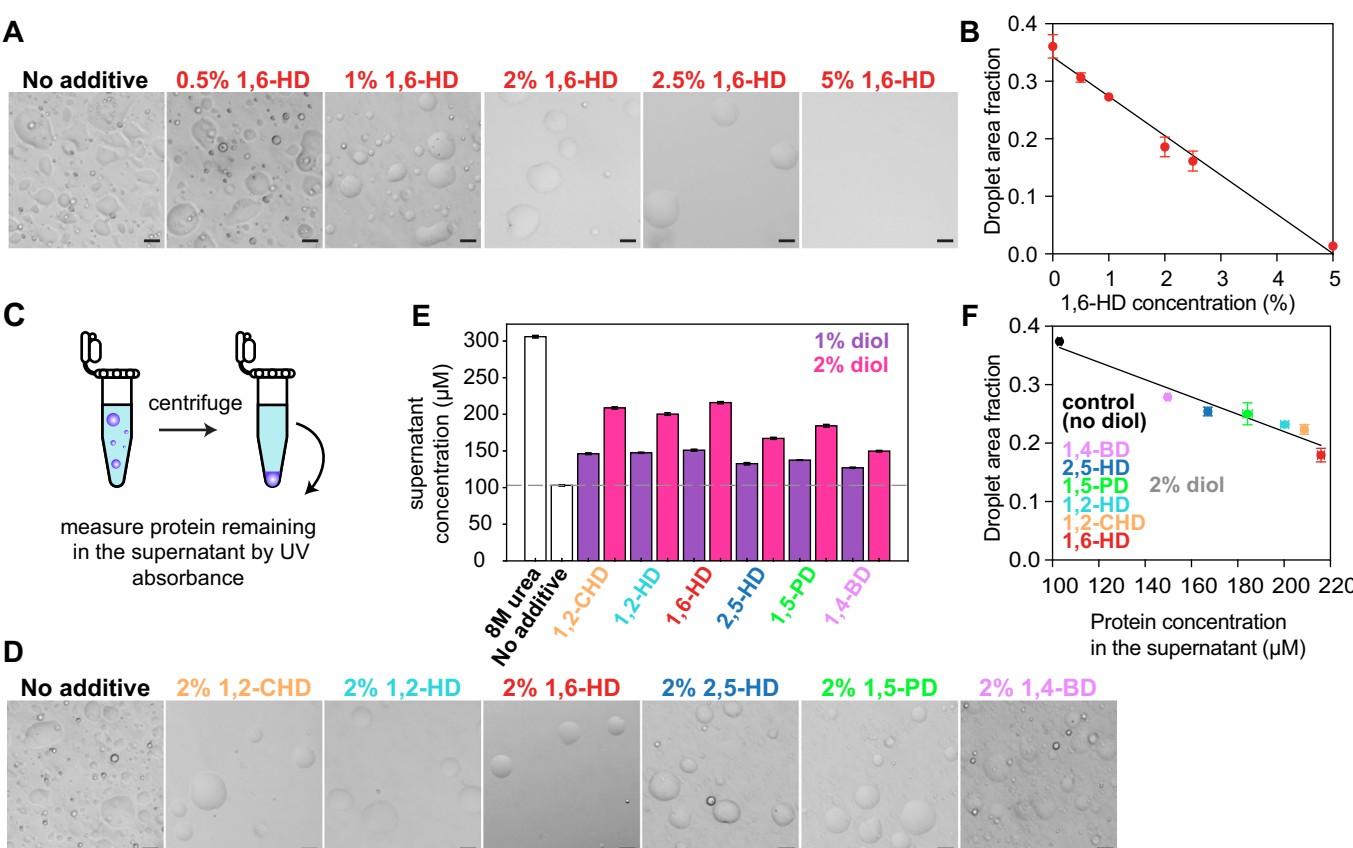

**Figure 1. Alkanediols quantitatively alter FUS LC phase separation.**

(A) DIC micrographs of 300 μM FUS LC in buffer alone (50 mM MES pH 5.5, 150 mM NaCl) and increasing concentrations of 1,6-hexanediol. Error bars represent s.d. of $n = 3$ technical replicates. (B) Extent of FUS LC phase separation in the presence of 0% to 5% 1,6-HD quantified by fluorescence microscopy. (C) Schematic of experimental setup for quantifying phase separation measuring the concentration of protein remaining in the supernatant after centrifugation of droplets. (D) DIC micrographs of 300 μM FUS LC in buffer alone (50 mM MES pH 5.5, 150 mM NaCl) (control) or various alkanediols with different aliphatic chain length or configuration premixed with buffer at 2%. Error bars represent s.d. of $n = 3$ technical replicates. (E) Phase separation assay that measures the saturation concentration of FUS LC in the presence of 1% or 2% of different alkanediols. Error bars represent s.d. of $n = 3$ technical replicates. (F) Phase separation quantification using fluorescence microscopy versus supernatant concentration assay shows a high level of agreement between the two methods. Error bars represent s.d. of $n = 3$ technical replicates. Data Information: Scale bars represent 20 μm. Source data are available online for this figure.

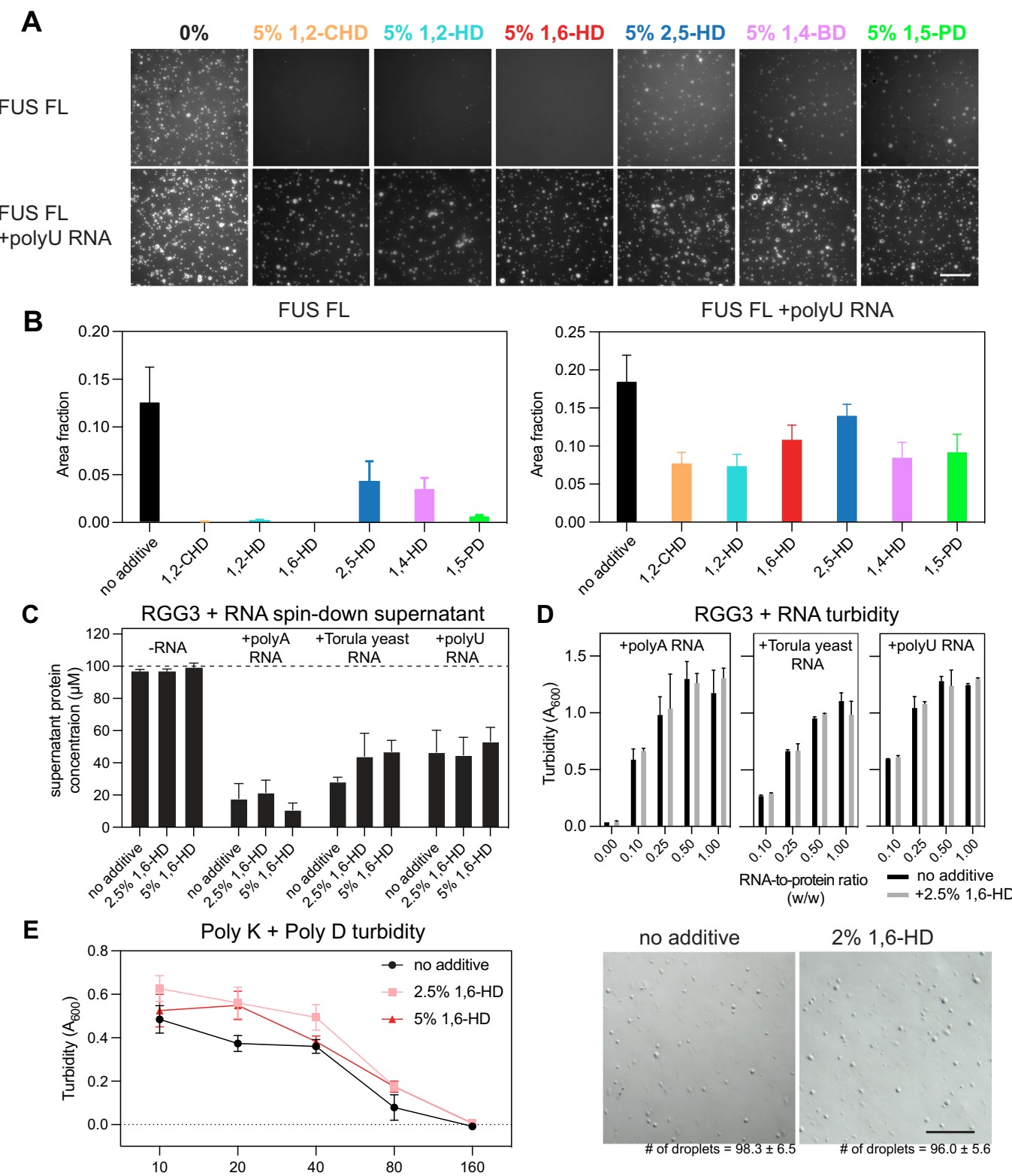

**Figure 2. Alkanediols do not effectively disrupt charge-driven phase separation.**

(A) Fluorescence microscopy images of 5 μM FUS full-length without (top) and with (bottom) polyuridylic acid (0.05 mg/ml polyU, 5:1 protein-to-RNA ratio w/w) after treatment with 5% w/v premixed solutions of aliphatic alcohols and 20 μM ThT. Scale bars represent 50 μm. (B) Phase separation of 5 μM FUS full-length without (left) and with (right) polyU RNA quantified by fluorescence-positive area fraction. Error bars represent s.d. of $n = 3$ technical replicates. (C) The effect of 2.5% or 5% 1,6-HD on phase separation of 100 μM FUS RGG3 in the presence of 1:1 (w/w) RNA measured by spin-down supernatant concentration. Error bars represent s.d. of $n = 3$ technical replicates. (D) The effect of 2.5% 1,6-HD on phase separation of 100 μM FUS RGG3 in the presence of RNAs across RNA-to-protein ratios, measured by turbidity at 600 nm wavelength. Error bars represent s.d. of $n = 3$ technical replicates. (E) (left) The effect of 2.5% or 5% 1,6-HD on phase separation of the mixture of 2 mM poly-L-lysine and 2 mM poly-L-aspartate measured by turbidity at 600 nm. Error bars represent s.d. of $n = 3$ technical replicates. (right) DIC micrographs of 2 mM poly-L-lysine and 2 mM poly-L-aspartate mixture with and without 2% 1,6-HD added show similar droplet counts (average of triplicate) in a square with edge length 200 μm. Data information: Scale bars represent 50 μm. Source data are available online for this figure.

1,6-HD is similarly incapable of disrupting other simpler complex coacervation driven by charged groups. To test this, we examined the phase separation behavior of poly-L-lysine (polyK) and poly-L-aspartate (polyD) peptide mixture (Fig. 2E). As previously reported (Cakmak et al, 2020; Perry et al, 2015), the negatively charged polyD and positively charged polyK peptides readily phase separated upon mixing, and the turbidity of this peptide mixture decreased sharply with increasing NaCl concentration, which screens charge-charge interactions between the peptides. Importantly, upon the addition of up to 5% 1,6-HD, we saw no apparent decrease in phase separation by microscopy; instead, the 1,6-HD additive appeared to slightly enhance turbidity, possibly due to the effect on charge screening for solutions containing large volume fractions of alcohols like 1,6-HD, that have decreased dielectric constants.

Together, these findings suggest that while 1,6-hexanediol readily disrupts phase separation of FUS LC, it does not appear to disrupt coacervation involving RNA interaction with FUS RGG or oppositely charged peptide interactions.

## FUS LC chemical environment and molecular motion are altered by alkanediols

We then used NMR spectroscopy to obtain more detailed insight into how 1,6-HD interferes with FUS LC phase separation. Leveraging the uniquely sensitive, residue-by-residue resolution of $^1$H-$^{15}$N heteronuclear single quantum coherence (HSQC) spectra (Fig. 3A), we observed small chemical shift perturbations (CSPs) between the 0% and 5% 1,6-HD conditions throughout FUS LC domain (Fig. 3B), that may arise from weak interactions between 1,6-HD and the protein and/or small changes in the protein conformation or interactions with water. While the CSPs are smaller than those typically associated with strong binding interactions, they are larger than those observed for the weak interactions of karyopherin-β2 with FUS LC (Yoshizawa et al, 2018). The observed CSPs are also not uniform across the sequence. Binning backbone CSPs according to amino acid types showed little to no systematic residue-type specific effects and a broad distribution of changes within each type (Fig. 3C). Consistent with the picture from the amide positions, we also observed small CSPs without clear residue-type specificity for backbone carbonyl ($^{13}$CO) positions in the presence of 1,6-HD (Appendix Fig. S4). Given the view that 1,6-HD is a hydrophobic disruptor, one might expect that hydrophobic amino acids in FUS may experience more chemical environment perturbation. However, neither these backbone CSPs above nor CSPs for sidechains from $^1$H-$^{13}$C HSQC spectra (Appendix Fig. S4B) showed large perturbations at hydrophobic amino acid positions. To examine if the range of 1,6-HD induced CSPs are specific to the FUS LC composition, we examined the backbone amide CSPs for FUS RGG3,

which does not undergo phase separation at these concentrations and conditions and has a distinct sequence composition (Murthy et al, 2021). We observed a similar magnitude of CSPs for FUS RGG3 (Appendix Fig. S5) as for FUS LC, suggesting that 1,6-HD similarly influences the chemical environment of distinct sequences. Therefore, our CSP analysis suggests that, despite certain residues showing larger CSPs, no clear pattern of interaction emerges.

Next, we examined the changes in local molecular motions of FUS LC due to 1,6-HD by measuring the $^{15}$N NMR spin relaxation parameters ($R_1$, $R_2$, hetNOE) at each backbone amide position (Fig. 3D). These values are sensitive to reorientational motions of the amide (NH) bond vector on the ps to ns timescales (Palmer et al, 2001). Decreased $R_2$ and elevated $R_1$ values in the presence of hexanediol suggest faster local reorientational tumbling motions, whereas the magnitude of the hetNOE (i.e., below 0.5) is unchanged and consistent with disorder across the domain, confirming that 1,6-HD does not cause significant structural rearrangements of the isolated LC domain. Interestingly, the decrease in $^{15}$N $R_2$ observed with the addition of 1,6-HD to FUS LC is similar to that caused by introducing phosphomimetic serine-to-glutamate low-complexity domain substitutions (FUS LC 12E) (Appendix Fig. S6A), preventing phase separation (Monahan et al, 2017). Hence, elevated $R_2$ values in wild-type FUS LC without 1,6-HD likely arise from transient favorable contacts between positions that also contribute to driving phase separation (Martin et al, 2020), and are suppressed by charged residue substitution (e.g., FUS LC 12E) or, as shown here, addition of 1,6-HD. Furthermore, addition of 1,6-HD to FUS LC 12E causes much smaller changes compared to the effects on FUS LC (Appendix Fig. S6B,C), suggesting that 1,6-HD and the introduction of the 12 phosphomimetic mutations may weaken the same intramolecular contacts (Appendix Fig. S6).

In an attempt to understand the mechanistic basis for the differing extents of phase separation disruption we observed for each alkanediol, we then sought to compare the effects of the different alkanediols on FUS LC by NMR. We acquired $^1$H-$^{15}$N HSQC spectra of FUS LC domain in the presence of 0%, 2.5%, or 5% of each alkanediol and mapped the CSPs to identify locations with changes in chemical environment (e.g., conformational change or interaction) (Appendix Fig. S7). Interestingly, when we plot each residue's CSP upon addition of each alkanediol compared to those for 1,6-HD, we find a strong correlation (Fig. 3E; Appendix Fig. S8A,B). Furthermore, the magnitude of the CSPs is strongly correlated with the $C_{sat}$ of these molecules in the phase separation assay (Fig. 3F; Appendix Fig. S8C,D), as shown by Pearson's correlation coefficient (PCC = 0.85 at 1% diol, PCC = 0.77 at 2% diol for $^{15}$N CSPs, and PCC = 0.92 at 1% diol, PCC = 0.84 at 2% diol for $^1$H CSPs). Hence, the changes in residue-by-residue

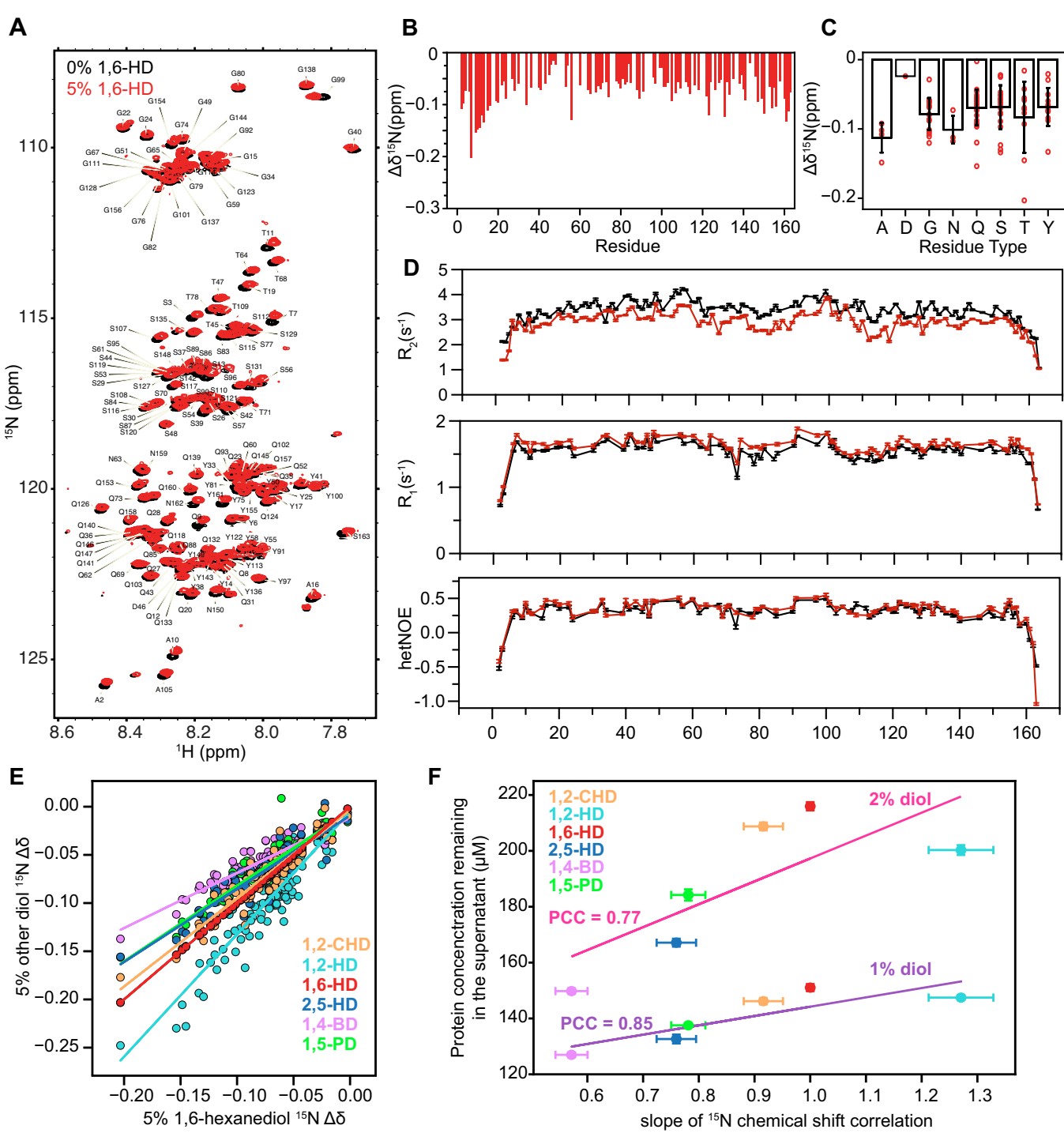

perturbation of the chemical environment are strongly correlated with the extent of phase separation disruption. In summary, we observe that the series of alkanediols show the same pattern of effects on FUS LC with different magnitudes that correlate with the extent of impact on phase separation. This observation suggests that all alkanediols, regardless of structure, have similar types of interactions with FUS. Hence, we do not find evidence here for unique interactions only possible by particular alkanediol isomers, though below we probe the factors that govern the extent of impact on phase separation.

## Alkanediols disrupt hydrophobic interactions

Structural differences (i.e., placement of the hydroxyl groups) have been suggested to make 2,5-HD less hydrophobic than 1,6-HD (Fig. 4A), though the hydrophobicity of these and other alkanediols has, to our knowledge, never been experimentally measured. Using NMR-based concentration quantification experiments, we measured the partition coefficient (logP) between octanol and water, a common measure of hydrophobicity, for each alkanediol (Fig. 4B) (Cheng et al, 2007). We found a range of partition coefficients

**Figure 3. The effect of alkanediols on the chemical environment of FUS LC.**

(A) $^1$H-$^{15}$N heteronuclear single quantum coherence (HSQC) spectra of 30 μM FUS LC in the absence (black) and in the presence (red) of 5% 1,6-hexanediol. Solutions in all cases contain 150 mM NaCl 50 mM MES pH 5.5. (B) FUS SYGQ LC $^{15}$N chemical shift deviations induced by 5% 1,6 hexanediol. Gaps indicate positions with no $^1$H/$^{15}$N resonance (e.g., proline residues that have no backbone $^1$H attached to nitrogen) or with overlapped resonances that were not resolved in these experiments. (C) 1,6-hexanediol induced $^{15}$N chemical shift deviations of FUS LC binned by residue type. Individual residues are plotted as red marks. Error bars represent s.d. of single replicate of chemical shift deviations from (B). (D) NMR spin relaxation parameters $^{15}$N $R_2$, $^{15}$N $R_1$ and ($^1$H) $^{15}$N heteronuclear NOE values for FUS LC at 850 MHz $^1$H frequency indicate slightly faster molecular motions in the presence of 1,6-hexanediol. Data are plotted as best model parameter and uncertainty corresponding to one s.d. from single replicate. (E) Comparison of $^{15}$N Δδ in 5% 1,6-hexanediol with every other co-solvent. (F) Slope extracted from the correlation presented in (E) versus protein remaining in the supernatant shows that the chemical shift differences induced by each diol are correlated with the capacity of FUS LC to phase separate in each condition (PCC = 0.85 at 1% diol, PCC = 0.77 at 2% diol). Solid lines represent linear fits. Data are plotted as best model parameter and uncertainty corresponding to one s.d. from single representative replicate. Source data are available online for this figure.

showing that the longer alkanediols are more hydrophobic and the 2,5-HD isomer is indeed less hydrophobic than 1,6-HD. We then compared the measured logP with the slope of the NMR $^{15}$N chemical shift perturbations and with the $C_{sat}$ of each diol from FUS LC phase separation assays (Fig. 4C). We saw a strong correlation between the NMR shifts and logP, indicating that more hydrophobic alkanediols induce higher CSPs. Additionally, the correlation between $C_{sat}$ and logP suggests that the hydrophobicity of the diols is linked to their ability to disrupt phase separation. However, 2,5-HD and 1,2-HD are outliers in the correlation between $C_{sat}$ and logP and removing these outliers significantly improves the correlation (PCC 0.76 to 0.99). This suggests that although hydrophobicity plays the dominant role, the unique geometries of these branched alkanediols do introduce additional contributions to their impact on phase separation, making them different from the alkanediols with hydroxyl groups at the terminal positions.

Taking inspiration from prior studies that have established the capability of aliphatic alcohols such as ethanol and propanol to destabilize protein structures by weakening hydrophobic interactions (Chong et al, 2015; Miyawaki and Tatsuno, 2011; Nakata et al, 2023), we sought to determine whether 1,6-HD, like its simpler counterparts, could exert a similar influence on hydrophobic interactions and protein structures. Specifically, we used Circular Dichroism (CD) as a function of temperature to assess the impact of 1,6-HD on the thermal stability of lysozyme—a protein that folds around a hydrophobic core (Fig. 4D). Interestingly, 10% 1,6-HD reduced the melting temperature ($T_m$) of lysozyme by approximately 10 °C. Next, we expanded our analysis to include 2,5-HD and other representative simple alcohols, such as ethanol and 1-propanol. We found that the extent of lysozyme $T_m$ reduction for 1,6 HD is comparable to propanol, which has the same number of linear aliphatic carbons per hydroxyl group (Fig. 4E), while 2,5-HD and ethanol have less impact on the $T_m$. Furthermore, we compared the structure-destabilizing capacity of the alcohols to their effect on FUS phase separation (Fig. 4E,F). For these experiments, we used a longer segment, FUS LC-RGG1 that we recently characterized, due to its increased ability to undergo phase separation compared to FUS LC (Wake et al, 2025), allowing quantification of the impact of 10% 1,6-HD and matching those we used in the CD studies. Interestingly, the lysozyme $T_m$ reduction of each alcohol correlates moderately (PCC = 0.77) with their individual potency in reducing phase separation (Fig. 4F). This observation suggests a shared mechanism for alcohol-induced reduction in phase separation and folding stability. Among all molecules tested, those with longer aliphatic chains (i.e., propanol and 1,6-HD) exhibited greater

potency in disrupting lysozyme thermal stability and FUS LC-RGG1 phase separation, again suggesting that the hydrophobicity of the alcohol plays a pivotal role in destabilizing both folding and phase separation.

## 1,6-Hexanediol perturbs protein-protein interactions via direct contacts with protein

To gain more atomistic details about the impact of alcohols on FUS LC phase separation, we performed atomistic MD simulations of an isolated single chain of FUS LC solvated with water and ion molecules ("in explicit solvent") in the presence of 5% 1,6-HD or 2,5-HD and compared these simulations to those without hexanediols. The parameters for both hexanediols, modeled with GAFF2 force field (He et al, 2020) (see Methods for more details), were validated by comparing the densities and molar volumes of water solutions with experimental data (Romero et al, 2007) (Appendix Fig. S9 and Appendix Tables S1 and S2).

Analysis of the protein conformation, including the radius of gyration ($R_g$) distribution and the average intrachain distance ($R_{ij}$), revealed that both alkanediols induced a more expanded structure for the LC domain compared to simulations without them (Fig. 5A and inset). Additionally, the expansion was more pronounced with 1,6-HD compared to 2,5-HD, which is consistent with 1,6-HD's stronger potency in disrupting FUS LC phase separation. Notably, FUS LC with 1,6-HD exhibited the largest $R_{ij}$ expansion, especially for residues that are furthest apart (|i − j| > 100), consistent with disruption of long-range protein-protein interactions. Contact analysis (see Methods) further supported these findings, showing a decrease in intramolecular protein-protein contacts in presence of either alkanediols, with a greater reduction observed with 1,6-HD (Fig. 5B). Particularly, 1,6-HD significantly disrupted mid- to long-range contacts, as indicated by a region of reduced values near the top of the contact heat map, as well as the distribution of average protein-protein contacts across various ranges of residue distances (Fig. 5C), which aligns with the observed $R_{ij}$ expansion shown in Fig. 5A.

Previous research has indicated that small molecules can influence proteins through direct interactions (Das and Mukhopadhyay, 2009; Nakata et al, 2023). To explore the potential disruptive mechanism of hexanediol, we investigated protein-alkanediol contacts in our simulations, which revealed extensive contacts formed between alkanediols and the entire protein chain (Fig. 5D; Appendix Fig. S10A), consistent with NMR data showing widespread CSPs throughout the protein. Notably, 1,6-HD formed more contacts with protein than 2,5-HD. Moreover, the radial

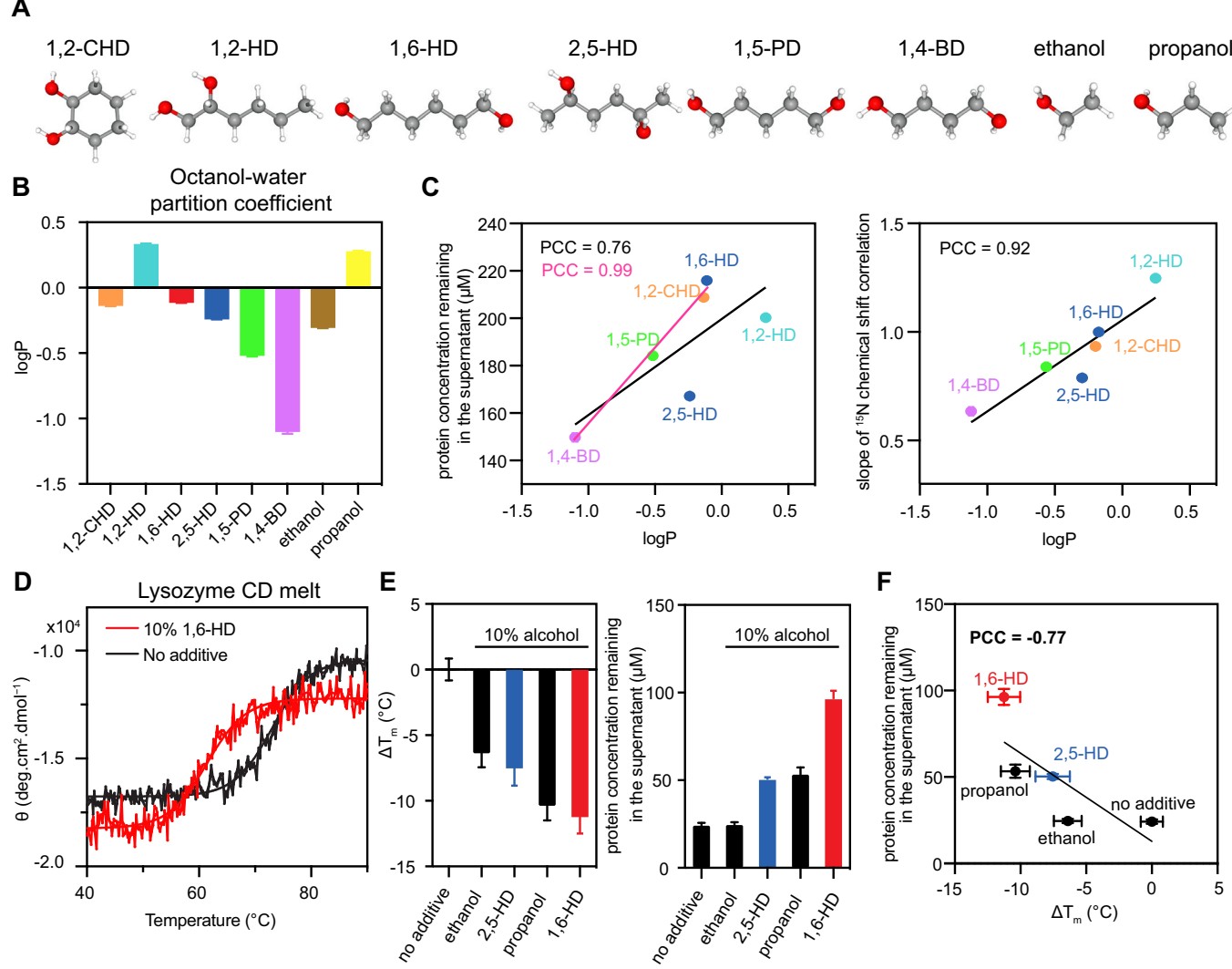

**Figure 4. Alkanediols destabilize protein structure and hydrophobic assembly.**

(A) Ball and stick representation of the structure of each alkanediol and simple alcohol studied. (B) NMR measured values of the octanol partition coefficient of each alkanediol, a measure of hydrophobicity. LogP corresponds to the logarithm of the ratio of the concentration of the solute in octanol versus in water. Error bars represent s.d. of $n = 3$ technical/chemical replicates. (C) Correlation of logP presented in (B) versus $^{15}N$ $\Delta\delta$ slopes in Fig. 3F (top), or versus protein remaining in the supernatant presented in Fig. 1E (bottom) shows that the effect of the alkanediols on protein backbone chemical shifts and FUS phase separation is correlated with the hydrophobicity. Pearson correlation coefficient (PCC) with all data points is shown in black, while the correlation excluding 2,5-HD and 1,2-HD is indicated in magenta. (D) Circular Dichroism data showing the effect of 10% 1,6-HD on the thermostability of lysozyme. (E) (left) The melting temperature of lysozyme decreases with the addition of simple alcohols as well as diols. (right) Saturation concentration of FUS LC-RGG1 with and without the addition of simple alcohols or diols. Error bars represent the standard deviation of three replicates. Data are plotted as best model parameter and uncertainty corresponding to one s.d. from single representative replicate. (F) Correlation between saturation concentration of FUS LC-RGG1 and melting temperature change ($\Delta T_m$) of lysozyme with different alcohols. Horizontal error bars and vertical error bars correspond to the values in (E). Source data are available online for this figure.

distribution functions (RDF, g(r)) for protein residues with hexanediols showed more close contact formation and surface enrichment for 1,6-HD (Fig. 5E; Appendix Fig. S10B). These findings suggest stronger interactions between 1,6-HD and protein, consistent with its greater potency in disrupting FUS phase separation.

Further details from residue-diol contact maps show that hexanediol forms contacts with different residues using different parts of the molecule (Fig. 5F; Appendix Fig. S10C). 1,6-hexanediol formed more contacts with Tyr using the aliphatic region and with

Gln using the polar ends. Interestingly, the geometric differences between 1,6-HD and 2,5-HD lead to differences in the number of protein contacts. Complementarily, comparison of 1,6-HD and 2,5-HD NMR $^1H$-$^{13}C$ HSQC spectra showed that the middle aliphatic methylene group carbon of 1,6-HD was more shielded (lower chemical shift in ppm) than that of 2,5-HD (Appendix Fig. S11), suggesting a less polar, more hydrophobic chain, which aligns with 1,6-HD's ability to better interact with hydrophobic amino acid side chains. Notably, the central section of 2,5-HD also shows lower contact frequency with Tyr than that of 1,6-HD (Fig. 5F), which

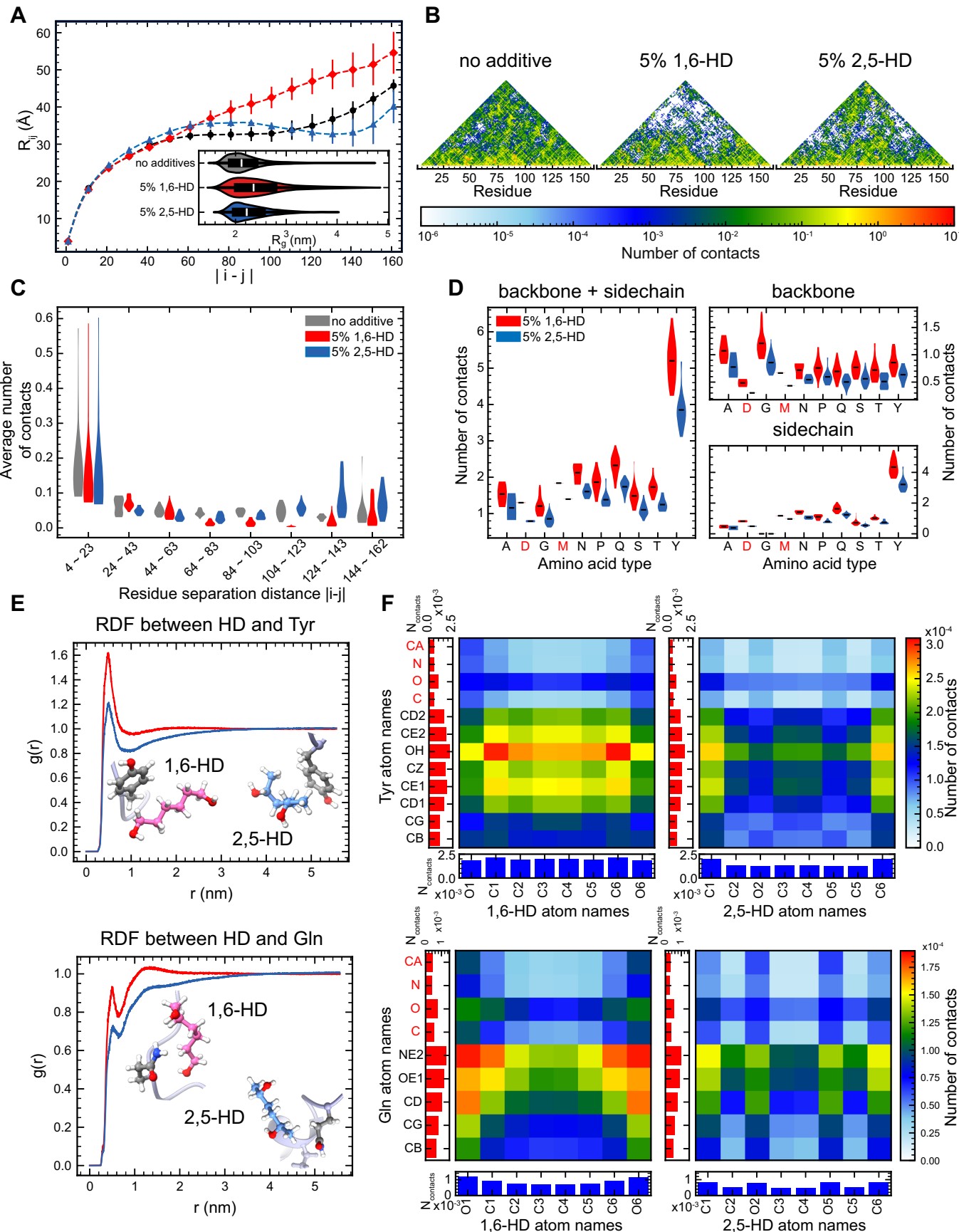

◄ **Figure 5.   1,6-HD perturbs long-range protein-protein interactions via direct contacts with protein.**

(A) The average intramolecular distance $R_{ij}$ between the $i^{th}$ and $j^{th}$ residues, calculated from atomistic simulations of single chains FUS LC with 100 mM NaCl without additives, with 5% 1,6-HD, or with 5% 2,5-HD. Error bars represent standard error of the mean computed over 3 independent trajectories for each condition. Inset: Distribution of radius of gyration ($R_g$) calculated from the simulations. (B) Protein-protein contact profiles calculated from the simulations. From left to right show the result from no additive, 5% 1,6-HD, and 5% 2,5-HD simulations, respectively. (C) The distribution of average protein-protein contacts in different ranges of residue distances. (D) The distribution of protein-hexanediol contacts calculated from the simulations, binned by residue type. (E) Radial distribution functions (RDF, g(r)) between the center of mass (COM) of sidechain heavy atoms of Tyr/Gln and heavy atoms of hexanediols. Corresponding interaction example snapshots are inset. (F) Atomistic contact maps of Tyr/Gln with hexanediols, calculated from the atomistic simulations. The left column and the bottom row of each panel show the one-dimensional summation of corresponding atomistic contacts. Source data are available online for this figure.

may be due to a combination of differences in steric hindrance, polarity, and hydrophobicity. In other words, the linear geometry of 1,6-HD appears to facilitate contacts with protein residues, in contrast to the branched heavy atom chain of 2,5-HD. Taken together, these data suggest that direct interaction between protein residues and hexanediols contribute to phase separation disruption and that some of the difference between 1,6-HD and 2,5-HD not fully explained by differences in hydrophobicity can be explained by changes in contacts caused by their distinct molecular geometries as proposed previously (Gu et al, 2023).

## Discussion

In this study, we demonstrated that alkanediols of different lengths and configurations, including 1,4-BD, 1,5-PD, 1,2-HD, 1,2-CHD, 1,6-HD, and 2,5-HD, can effectively decrease phase separation of FUS. Although very high (60%) hexanediol concentrations have been suggested to cause the ordering of an unstructured region in a protein crystal (Buhrman et al, 2003), our data suggest no change to the overall disordered structure of FUS LC in the concentration range (up to 10%) used in cellular studies. In fact, faster reorientational motions for FUS LC with 1,6-HD likely stem from decreased self-interaction within FUS molecules. Among tested compounds, 1,6-HD stands out as the most effective disruptor of FUS phase separation. Indeed, 2,5-HD is less effective than 1,6-HD in dissolving nuclear speckles and subsequent recruitment of a proline-glutamine-rich splicing factor (Levone et al, 2021) and melting FUS LC hydrogels and intracellular puncta (Kato and McKnight, 2018), and it shows significantly less impact on FUS phase separation (Figs. 1 and 3). However, we show here that despite being less potent than 1,6-HD, 2,5-HD and shorter alkanediols are still able to reduce phase separation via a qualitatively similar mechanism. Taken together, these results suggest that specific interactions uniquely available to particular alkanediol isomers (e.g., simultaneous hydrogen bonding of both hydroxyl groups only geometrically possible in 1,6-HD and not 2,5-HD) do not explain the differences in their action. Rather, on the atomic level, the aliphatic segment of 1,6-HD allows interactions with hydrophobic and aromatic amino acids such as tyrosine, while the polar hydroxyl groups enable hydrogen bonds with polar residues such as glutamine, suggesting it may compete with both of these residue types for protein-protein contacts that drive phase separation (Murthy et al, 2019; Wake et al, 2025; Wang et al, 2018). Furthermore, these simulations and experiments provide insight into how hydrophobicity and geometry contribute to the differences in potency between 1,6-HD and 2,5-HD. Specifically, the

linear aliphatic chain present in 1,6-HD is less polar and more accessible for interaction with hydrophobic protein residues, while the primary alcohol in 1,6-HD is more polar than the secondary alcohol in 2,5-HD, hence more favorable for dipole-dipole interactions. Both factors combine to make 1,6-HD more amphiphilic than 2,5-HD, consistent with 1,6-HD's greater ability to suppress phase separation than 2,5-HD.

Consistent with earlier biophysical studies on FG-rich nucleoporins and the synaptonemal complex (SC) that proposed that the capacity of aliphatic alcohols to dissolve the NPC and SC is correlated with the hydrophobicity of each alkanediol (Ribbeck and Gorlich, 2002; Rog et al, 2017), we measured the hydrophobicity of each alkanediol and found that it correlates extremely well with the ability of alkanediols with terminal hydroxyl groups (1,6-HD, 1,5-PD, 1,4-BD) to disrupt phase separation. However, the unique structures of 1,2-HD and 2,5-HD result in less disruption of phase separation than predicted by hydrophobicity alone, suggesting that the geometry of the molecule does play a role in determining the capacity of the alcohols to disrupt FUS LC phase separation. Consistent with disruption of the hydrophobic driving forces for folding, we also show that aliphatic alcohols, including 1,6-HD, can also disrupt protein folding stability, suggesting that alkanediols may also unfold cellular proteins. Intriguingly, 1,2-HD also disrupted TEV protease activity (Appendix Fig. S2A), complementing previous findings that 1,6-HD disrupts kinase and phosphatase activity (Düster et al, 2021). Therefore, our data suggest that disruptions beyond only disordered domain contacts may contribute to cellular changes caused by hexanediol treatment, and the mechanism of these changes should be inferred cautiously. Nevertheless, our biochemical data suggest that interference with hydrophobic interactions constitutes a major mechanism underpinning the disruptive function of 1,6-HD in the context of biochemical FUS phase separation, as was also suggested by the impact of Hofmeister series salts on FUS phase separation (Murthy et al, 2019).

Some RNA-rich condensates in cells show insensitivity to 1,6-HD treatment (Ahmed et al, 2021; Sato et al, 2022). Our data show that the alkanediol-sensitivity of FUS condensates enriched in RNA is reduced compared to FUS without RNA (Fig. 2A), and FUS RGG-RNA and polyK-polyD complex coacervation are largely unaffected by hexanediol treatment, suggesting that charge-charge interactions mediating phase separation are not impacted by amphiphilic alcohols. Although hexanediol-dependent dissolution has been successfully used to distinguish solid from liquid forms of particular membraneless organelles, such as those stabilized by hydrophobic interactions (Kroschwald et al, 2017), the fact that certain condensates are not susceptible to 1,6-hexanediol does not necessarily mean they are solid. Instead, these condensates may primarily involve charge-charge interactions and may indeed be liquid. As

a corollary, the large number of membraneless organelles found to be susceptible to 1,6-hexanediol may also therefore imply that these are primarily stabilized by hydrophobic and not charge-charge interactions. However, these observations that interactions between disordered RGG peptides and unstructured RNA or total RNA extracts are not disrupted by 1,6-HD should not be taken to mean that no cellular RNA-protein interactions are disrupted by 1,6-HD. Indeed, FUS and many other phase-separating RNA-binding proteins contain RRMs and other domains that interact with single stranded RNA, not primarily via the charged backbone of RNA, but rather via the bases using hydrophobic/aromatic residues (Loughlin et al, 2019; Ozdilek et al, 2017; Ozguney et al, 2024), which may be disrupted by 1,6-HD. Thus, the effectiveness of hexanediol is context-dependent and results of cellular experiments adding HD should be interpreted with careful consideration of the condensate composition (e.g., sequence characteristics and nucleic acid partitioning).

Given the expanding knowledge of the roles of biomolecular condensates in many physiological processes (Lyon et al, 2021) and in diseases (Alberti and Dormann, 2019; Alberti and Hyman, 2021; Jiang et al, 2020; Lu et al, 2021), many efforts have focused on understanding how to modulate phase separation and how potential small molecule therapeutics partition into condensates (Dai et al, 2021; Klein et al, 2020; Wheeler et al, 2016). For example, mitoxantrone and other chemotherapy drugs with targets that reside in nuclear condensates selectively concentrate in nuclear condensates (Chong et al, 2015). This selective concentration may be due to direct interactions between the compound and the disordered protein—although these direct interactions are weak, they can show similarly sized NMR chemical shift perturbations at 3 orders of magnitude lower concentrations (500 μM vs ~400 mM) (Uechi et al, 2024) than perturbations generated by alkanediols (Fig. 3), suggesting that small molecules other than hexanediol may be able to alter condensate properties. Our results show that NMR-based observations combined with molecular simulation can provide unique insight into the molecular origins of phase separation modulation and could contribute to the rational design of possible therapies for altering disordered protein phase separation.

# Methods

### Reagents and tools table

| Reagent/Resource | Reference or Source | Identifier or Catalog Number |
|---|---|---|
| **Recombinant DNA** | | |
| pTHMT FUS 1-526, | AddGene | cat # 98651 |
| pRP1B FUS LC-RGG1 (1-284) | Fawzi Lab | pRP1B_FUS_LC-RGG1 |
| pTHMT FUS RGG3 (453-507) | Fawzi Lab | pTHMT_FUS_RGG3 |
| **Oligonucleotides and other sequence-based reagents** | | |
| Poly-L-lysine hydrochloride | Alamanda Polymers | cat # 26124 |
| Poly-L-aspartic acid sodium salt | Alamanda Polymers | cat # 34345 |
| Polyadenylic acid potassium salt | Sigma-Aldrich | cat # P9403 |
| Polyuridylic acid potassium salt | Sigma-Aldrich | cat # P9528 |

| Reagent/Resource | Reference or Source | Identifier or Catalog Number |
|---|---|---|
| Torula yeast RNA extract | Sigma-Aldrich | cat # R6625 |
| **Chemicals, Enzymes and other reagents** | | |
| 1,2-hexanediol | Sigma-Aldrich | cat # 213691 |
| 1,6-cyclohexanediol | Sigma-Aldrich | cat #141712 |
| 1,6-hexanediol | Sigma-Aldrich | cat # 240117 |
| 2,5-hexanediol | Sigma-Aldrich | cat # H11904 |
| 1,4-butanediol | Sigma-Aldrich | cat # 493732 |
| 1,5-pentanediol | Sigma-Aldrich | cat # P7703 |
| Lysozyme | Sigma-Aldrich | cat # L4919 |
| **Software** | | |
| NMRPipe | (Delaglio et al, 1995) | |
| CCPNmr Analysis | (Vranken et al, 2005) | |
| NMRFAM-Sparky | (Lee et al, 2015) | |
| Fiji (ImageJ) | (Schindelin et al, 2012) | |
| ACPYPE python package | (Sousa da Silva and Vranken, 2012) | |
| GAFF2 force field | (He et al, 2020) | |
| GROMACS-2022 MD package | (Abraham et al, 2015) | |
| AMBER03ws force field | (Best et al, 2014) | |
| Amber22 package | (Case et al, 2022) | |
| ParmEd package | (Shirts et al, 2017) | |
| MDAnalysis-2.5.0 | (Gowers et al, 2016) | |
| Bruker Topspin 3.2 | http://bruker.com/topspin | |
| Matlab | https://www.mathworks.com/products/matlab.html | |
| **Other** | | |
| Fluorescence microscope | Zeiss | Axiovert 200 M |
| NMR spectrometer | Bruker | Avance III 850 MHz |
| CD spectropolarimeter | JASCO | J-815 |

## General information

1,2-hexanediol (#213691), 1,6-cyclohexanediol (#141712), 1,6-hexanediol (#240117), 2,5-hexanediol (#H11904), 1,4-butanediol (#493732) and 1,5-pentanediol (#P7703) were purchased from Sigma-Aldrich.

## Protein purification

FUS LC containing a TEV cleavable N-terminal histidine tag (RP1B FUS LC, AddGene #127192), full-length FUS with a TEV cleavable N-terminal histidine/maltose-binding protein (MBP) fusion tag (pTHMT FUS 1-526, AddGene #98651), FUS LC-RGG1 containing a TEV cleavable N-terminal histidine tag (pRP1B FUS LC-RGG1 (1-284)) and FUS RGG3 with a TEV cleavable N-terminal

histidine/MBP fusion tag (pTHMT FUS RGG3 (453-507)) were expressed in BL21 Star (DE3) *Escherichia coli* cells (Life Technologies) grown at 37 °C to an OD of 0.60–0.90 before induction with 1 mM IPTG for 4 h. Isotopically labeled protein was produced by expression in M9 minimal media supplemented with $^{15}$N-ammonium chloride or $^{13}$C-glucose (Cambridge Isotopes). Histidine tagged FUS LC and FUS LC-RGG1 was purified as described previously (Wake et al, 2025). In brief, cells were lysed using an Avestin homogenizer and the lysate was cleared by centrifugation at 14,000 g for 1 h. The insoluble fraction was applied to a 5 mL HisTrap HP column (Cytiva) equilibrated with 8 M urea 20 mM NaPi pH 7.4 300 mM NaCl 10 mM imidazole and eluted with a gradient from 10 to 300 mM imidazole. The protein was diluted with 20 mM NaPi pH 7.4 such that the final urea concentration was 1 M and incubated with TEV protease overnight. A subtractive nickel affinity step was performed, and the protein in the flow through was then buffer exchanged into 20 mM CAPS pH 11.0 and concentrated for storage by centrifugal filtration. MBP-FUS 1-526 was purified as previously described (Burke et al, 2015). FUS RGG3 was purified as previously described (Murthy et al, 2021). All protein preparations showed UV absorbance ratio, $A_{260\ nm}/A_{280\ nm}$, of 0.5 to 0.6, indicating minimal to no residual nucleic acid contamination. Proteins were stored as frozen aliquots at −80 °C.

## Phase separation assays

Phase separation of FUS LC in the presence of hexanediols was quantified by measuring the absorbance at 280 nm of the dilute phase using a NanoDrop spectrophotometer. Samples were prepared by diluting FUS LC stored in 20 mM CAPS pH 11.0 to a final protein concentration of 300 μM into 20 mM MES (pH adjusted with Bis-Tris) pH 5.5 150 mM NaCl with and without hexanediols. After dilution, samples were spun at 14,000 × g for 10 min at 22 °C. Phase separation of FUS LC-RGG in the presence of hexanediols was quantified the same way as FUS LC.

Turbidity of RGG3 was measured as previously described (Murthy et al, 2021) by measuring the absorbance at 600 nm of samples in a 96-well clear plate (Costar) using a Cytation 5 Cell Imaging Multi-Mode Reader (BioTek). Samples were prepared in 50 mM MES pH 5.5, 150 mM NaCl buffer. For turbidity of FUS FL, TEV protease was added to 10 μM MBP-FUS FL in 50 mM HEPES pH 7 with 150 mM NaCl and, after 60 min, buffer containing 10% alkanediols and RNA, for a final concentration of 5% alkanediol. To remove background absorbance, the turbidity of a no TEV control (i.e., with TEV storage buffer) for each condition was subtracted from the turbidity of the experimental conditions. Experiments were conducted in triplicate and averaged. To test the effect of different alkanediols on RNA-FUS phase separation, the poly-adenylic acid (polyA), polyuridylic acid (polyU), and torula yeast RNA extract type VI (Sigma #R6625) were desalted into the appropriate buffer using a Zeba 0.5 ml spin column.

To test the effect of 1,6-HD on polyD + polyK phase separation, the peptides were dissolved in 50 μM MES pH 5.5 buffer, turbidity of 2 mM polyK + 2 mM polyD mixtures with 10–160 mM NaCl was measured in a 96-well clear plate (Costar) using the Cytation 5 reader by measuring the absorbance at 600 nm. Experiments were conducted in triplicate and averaged. Poly-L-lysine hydrochloride

(PLKC10, #26124) and poly-L-aspartic acid sodium salt (PLD10, #34345) were purchased from Alamanda Polymers.

## Microscopy

Visualization of phase separation of FUS LC, MBP-FUS FL and FUS RGG3 was performed by differential interference contrast microscopy on a Zeiss Axiovert 200 M Fluorescence microscope. All samples were spotted on a 24 × 40 × 1.5 mm coverslip for imaging using a 40x water immersion microscope objective. Images were processed using ImageJ (NIH). Fluorescence microscopy samples were prepared as 200 μM FUS LC or (60 μM FUS FL) in a 50 mM HEPES buffer (pH adjusted to 7.0 using Bis-Tris) containing 150 mM sodium chloride, 20 μM Thioflavin T as a fluorescent dye, and varying volume fractions of alkanediols. All samples were spotted onto the coverslip and droplets were allowed to settle for 2 min prior to capturing four independent fields of view for quantification. The droplet area ratio was determined using an in-house Matlab (2023b) script that calculates the percentage of fluorescently illuminated pixels corresponding to droplet area. Droplet counting for the polyK-polyD droplets were performed using ImageJ.

## NMR spectroscopy

NMR experiments were recorded at 850 MHz using a Bruker Avance III spectrometer with a $^1$H/$^{15}$N/$^{13}$C TCl cryoprobe and $z$ field gradient coil. NMR titrations of $^{15}$N-labeled FUS LC with 0, 2.5, or 5% hexanediols were conducted at 25 °C in 50 mM MES, 150 mM NaCl pH 5.5 including 10% $^2$H$_2$O and 2 μM sodium trimethylsilylpropane sulfonate (DSS). All data were processed with NMRPipe software package (Delaglio et al, 1995) and visualized with NMRFAM-Sparky (Lee et al, 2015) or CcpNMR Analysis (Vranken et al, 2005). Chemical shifts and intensity ratios were normalized by subtracting the $^{15}$N chemical shift values and dividing the signal intensity in the absence of hexanediol from all other datasets. DSS was used as the direct reference for the $^1$H chemical shift, while the $^{13}$C and $^{15}$N chemical shifts were referenced indirectly using DSS using their gyromagnetic ratios. Specifically, adjustments in the nitrogen and carbon dimensions were calculated based on the following formula (Harris et al, 2001):

$$V_X = \frac{V_{1H} * X}{100}, \text{where } X_{15N} = 10.132912 \text{ and } X_{13C} = 25.144953$$

Molecular motions in the presence of 1,6-hexanediol were probed using $^{15}$N $R_1$, $^{15}$N $R_2$ and heteronuclear NOE experiments using standard pulse sequences (hsqct2etf3gpsitc3d, hsqct1etf3gp-sitc3d, hsqcnoef3gpsi, respectively, from Bruker Topspin 3.2). Interleaved experiments comprised 256 × 4096 total points in the indirect $^{15}$N and direct $^1$H dimensions, respectively, with corresponding acquisition times of 74 ms and 229 ms, sweep width of 20 ppm and 10.5 ppm, centered at 117 ppm and 4.7 ppm, respectively. $^{15}$N $R_2$ experiments had an interscan delay of 2.5 s, a Carr-Purcell-Meiboom-Gill (CPMG) field of 556 Hz, and total $R_2$ relaxation CMPG loop-lengths of 16.5 ms, 264.4 ms, 181.8 ms, 33.1 ms, 115.7 ms, 82.6 ms, and 165.3 ms. $^{15}$N $R_1$ experiments had an interscan delay of 1.2 s, and total $R_1$ relaxation loop-lengths of 100 ms, 1000 ms, 200 ms, 800 ms, 300 ms, 600 ms, and 400 ms.

Heteronuclear NOE experiments were conducted with an interscan delay of 5 s.

## Circular Dichroism spectrometry of lysozyme

0.1 mg/mL lysozyme (Sigma #L4919) solution was prepared in 5 mM sodium phosphate buffer at pH 7. The solution was then filtered through a 0.22 μm membrane filter. Circular dichroism (CD) spectra of lysozyme were recorded using a JASCO J-815 spectropolarimeter equipped with a Peltier temperature control system. Molar residue ellipticity (MRE) at 222 nm was monitored in a temperature range between 25 and 90 °C both in the presence and absence of 10% alcohol additives.

## Single chain MD simulations of FUS LC with or without hexanediol additives

Force field parameters for 1,6- and 2,5-hexanediol were obtained from the open source ACPYPE python package (Sousa da Silva and Vranken, 2012) with the GAFF2 force field (He et al, 2020).

Initial equilibration simulations were performed using the classical MD package GROMACS-2022 (Abraham et al, 2015), employing periodic boundary conditions and a 2 fs integration time step. All systems were simulated in explicit solvent, modeled using the AMBER03ws force field and explicit TIP4P/2005s water model (Best et al, 2014) with improved salt (NaCl) parameters (Luo and Roux, 2009).

FUS LC initial protein structure was placed in a $12 \times 12 \times 12$ nm octahedral box, and 330 hexanediol molecules were added to achieve a concentration of ~5%. The system was then solvated, with counter ions added to achieve electroneutrality and a salt concentration of 100 mM. It then underwent energy minimization using the steepest descent algorithm, followed by a 100 ps NVT equilibration. Velocity rescaling algorithm (Bussi et al, 2007) was used for temperature equilibration at 300 K with a coupling constant of 0.1 ps. Subsequently, a 100 ps NPT equilibration was conducted using the Parrinello-Rahman barostat (Parrinello and Rahman, 1980) with a coupling constant of 2 ps for pressure control.

The production runs were performed with MD package Amber22 (Case et al, 2022). After GROMACS equilibration, the package ParmEd (Shirts et al, 2017) was used to convert GROMACS files to Amber files. Hydrogen mass repartitioning (Hopkins et al, 2015) was applied during conversion to enable a timestep of 4 fs for production runs. Following the conversion, energy minimization was applied with protein position restraint, followed by a 2 ns NVT equilibration with reduced protein position restraint. With Berendsen barostat (Berendsen et al, 1984), a 500 ps NPT equilibration with further reduced protein position restraint was conducted. After the equilibration steps, the production run simulations were performed in the NPT ensemble. Langevin dynamics (Pastor, 1994) were used to control temperature at 300 K (friction coefficient = $1 \text{ ps}^{-1}$), a Monte Carlo barostat (Åqvist et al, 2004) was applied to control pressure at 1 bar (coupling constant = 1), and the SHAKE algorithm, as implemented in Amber22, was used for constraining hydrogen-containing bonds.

After production runs, the Amber NetCDF trajectory files were converted to Gromacs compressed trajectory files via CPPTRAJ (Roe and Cheatham, 2013) package. Python package MDAnalysis-2.5.0 (Gowers et al, 2016) was used for further analysis. A contact

was considered formed if any heavy atom from two residues or hexanediol was within 4.5 Å.

## Data availability

NMR data: spectra BMRB 52913 (https://bmrb.io/data_library/summary/index.php?bmrbId=52913). DNA data: Recombinant protein expression plasmids Addgene (https://www.addgene.org/Nicolas_Fawzi/). MD Simulation data: 1,6- and 2,5-hexanediol MD simulation parameters Bitbucket (https://bitbucket.org/jeetain/all-atom_ff_refinements/src/master/hexanediol_parameters/). Microscope date: DIC microscopy images BioImages Archive S-BIAD1666 (https://www.ebi.ac.uk/biostudies/bioimages/studies/S-BIAD1666). Scripts: Microscopy image processing script Github (https://github.com/WakeN-1/ImageSegmentation).

The source data of this paper are collected in the following database record: biostudies:S-SCDT-10_1038-S44318-025-00431-2.

## Peer review information

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

## Acknowledgements

We thank Dr. Mandar Naik for NMR assistance and the Structural Biology Core Facility at Brown University. Research was supported in part by NIGMS R01GM147677 (to NLF), Human Frontier Science Program RGP0045/2018 (to NLF), grant 1845734 from the National Science Foundation (to NLF), and NIGMS R35GM153388 (to JM). TZ was supported in part by a Pape Adams Postdoctoral Award from the Carney Institute for Brain Science at Brown University and a Milton Safenowitz Postdoctoral Fellowship (23-PDF-629) from the ALS Association. NW was supported in part by an NIGMS training grant at Brown University (T32GM139793). ACM was supported in part by NIGMS training grant at Brown University (T32GM007601, T32GM136566) and NSF graduate fellowship (1644760, to ACM). This content solely reflects the authors and does not necessarily represent the official views of the funding agencies. We gratefully acknowledge the computational resources provided by the Texas A&M High Performance Research Computing (HPRC).

## Author contributions

**Tongyin Zheng**: Investigation; Methodology; Writing—review and editing. **Noah Wake**: Validation; Investigation; Methodology; Writing—original draft. **Shuo-Lin Weng**: Investigation; Methodology; Writing—review and editing. **Theodora Myrto Perdikari**: Formal analysis; Investigation; Methodology; Writing—original draft; Writing—review and editing. **Anastasia C Murthy**: Conceptualization; Formal analysis; Supervision; Funding acquisition; Investigation; Methodology; Writing—original draft. **Jeetain Mittal**: Conceptualization; Supervision; Funding acquisition; Methodology; Writing—review and editing. **Nicolas L Fawzi**: Conceptualization; Supervision; Funding acquisition; Investigation; Methodology; Writing—original draft; Writing—review and editing.

Source data underlying figure panels in this paper may have individual authorship assigned. Where available, figure panel/source data authorship is listed in the following database record: biostudies:S-SCDT-10_1038-S44318-025-00431-2.

## Disclosure and competing interests statement

The authors declare no competing interests.

