## [Peer Review File · The EMBO Journal]

Molecular insights into the effect of 1,6-hexanediol on FUS phase separation

Tongyin Zheng, Noah Wake, Shuo-Lin Weng, Theodora Perdikari, Anastasia Murthy, Jeetain Mittal, and Nicolas Fawzi

Corresponding author(s): Nicolas Fawzi (nicolas_fawzi@brown.edu) , Jeetain Mittal (jeetain@tamu.edu)

Review Timeline:

Submission Date:	3rd Jun 22
Editorial Decision:	8th Jul 22
Revision Received:	14th May 23
Editorial Decision:	15th Sep 23
Revision Received:	22nd Nov 24
Editorial Decision:	22nd Jan 25
Revision Received:	7th Mar 25
Accepted:	13th Mar 25

Editors: Karin Dumstrei and Ioannis Papaioannou

Transaction Report:

Dear Nick,

Thank you for submitting your manuscript to the EMBO Journal. Your study has now been seen by three referees and their comments are provided below.

While referee #3 is not convinced that we gain enough insight for consideration here, referees #1 and 2 are more supportive and find the analysis important. I have looked at everything carefully and while I see the points raised by referee #3, I am also in agreement with the comments made by the other two referees about the importance of this study for the field. I would therefore like to invite you to submit a revised version.

It would be helpful to discuss the revisions further and I am available to do so via email or a video. Let me know when it is a good time for you.

Looking forward to discussing your revisions further

With best wishes

Karin

Karin Dumstrei, PhD
Senior Editor
The EMBO Journal

I have attached a PDF with helpful tips on how to prepare the revised version.

Guide For Authors: <https://www.embopress.org/page/journal/14602075/authorguide>

We realize that it is difficult to revise to a specific deadline. In the interest of protecting the conceptual advance provided by the work, we recommend a revision within 3 months (6th Oct 2022).

As a matter of policy, competing manuscripts published during this period will not negatively impact on our assessment of the conceptual advance presented by your study. However, we request that you contact the editor as soon as possible upon publication of any related work, to discuss how to proceed.

If you require more time to complete the revisions let me know as as I can grant an extension.

Use the link below to submit your revision:

Referee #1:

Perdikari et al., try to elucidate the mechanism of dissolution of biomolecular condensates (BMC) by alkanediol isomers. They show that the alkanediol isomers act all similarly as solvation agents by favoring protein-solvent hydrogen bond interactions rendering them unsuitable for therapeutic applications. They additionally suggest that some alkanediol can no longer disrupt FUS BMC that contain polyA RNA showing the limitations of alkanediols to study protein-RNA phase separation.

This study represents an important step towards understanding the mechanistic role alkanediols have on BMCs. By showing that the tested alkanediol isomers have all similar functions they demonstrate that many studies in the field are indeed comparable, which until date has only been assumed. While one could argue that this paper is of technical nature, its importance lies in shattering the recent criticism for using alkanediol isomers as phase separation inhibitors. The field will benefit by the detailed and methodological analysis presented here. For this reason, I fully support publication of this work in EMBO Journal, after

revision to address the following points for criticism.

Major points of criticism:

- Some qualitative statements are made throughout the manuscript; however, no quantification is shown in support of these statements. In particular, the authors should quantify the number of droplets and their perimeter and apply statistical analysis to the following panels: 1A, 2A, 2C, S11A, S13.
- Many DIC images seem to have been recorded with different illumination strengths. If this is the case, the authors should repeat these experiments using the same illumination strength for all conditions.
- No statistical analysis has been shown for quantitative Figures. The authors should perform and integrate statistical analysis of the following Figures: 1B, 1C at $t=0$, 2B at $t=0$, 2D, S11B.
- As purity affects the final concentration of biomolecules and BMC formation is concentration dependent, I think it is important to show the purification analysis for all the protein constructs used in this study.
- For 2,5% and 5% 2,5-HD there seems to be a difference in the effectiveness of droplet dissolution. Several concentrations of alkanediol isomer have been used in the field over the years (usually ranging from 1-10% w/v alkanediol concentration). It would, therefore, be particularly useful if the authors performed a dose response ranging from 1 to 10% for all alkanediol isomers used in this manuscript.
- In Figure 2B 1,2-HD treated samples in the RNA+ condition shows an increase in turbidity. While a microscopy-based timeline has been established for diols that do not disrupt phase separation the other alkanediols are missing. For completion, the authors should perform microscopy-based timeline recordings of all the diols with and without RNA.
- In Figure 1 and Figure 2 the images taken by DIC do not necessarily reproduce perfectly the later findings by turbidity measurements and supernatant protein concentration determination. For example:
 - o 1B: The concentration difference between 5% 2,5-HD and 5% 1,5-PD is minimal, however, the images in 1A show a clear difference between the 2,5-HD and 1,5-PD conditions. Can the authors explain this discrepancy?
 - o Discrepancy between 1B and 1C: If you compare 5% 2,5-HD with 5% 1,5-PD then the absorbance should reflect the same trend as in 1C. Why doesn't it? Could the author please comment?
 - o Figure 1A and 2A of the untreated samples should look the same. Can the authors please comment on the differences between these images?

It would be helpful to show an experimental schematic on how this data was acquired for readers to better understand the findings.

- Some NMR experiments and their interpretation are hard to understand. It would be beneficial for the broad readership of EMBO Journal - especially those unfamiliar with NMR - to elaborate on the experimental readouts and the conclusions in more depth. Some examples:
 - o Please annotate in the Figure legend for differential chemical shift perturbation plots that blank residues mean that there was no shift detected.
 - o Comment on the strength of the chemical shift perturbation in Figure S12.
 - o "In other words, the sequence-specific effects are largely the same for all diols tested. The observation that the series of alkanediols show a quantitative gradient of effects on LLPS and chemical environment with qualitative similarity suggest that all alkanediols have similar mechanisms of action."

Figure 1D: both 1,2-HD and 1,5-PD show clear outliers. Could the authors comment on those outliers and explain what this might mean for the effectiveness of these diols? Interestingly, 1,2-HD is a good solvent while 1,5-HD is not. Could the outliers also explain the difference in performance? Might there be a sequence specific effect?

Figure S12 shows that some residues are more affected (bigger CSP) while others are not affected at all (empty lanes in the residue space) and there are differences in the CSP pattern between the alkanediol isomers. Is this not a direct indication that some residues are more affected than others?

o "We quantified the chemical shift perturbations between the control and 1,6-hexanediol conditions and observed weak non-specific interactions throughout FUS SYGQ LC domain."

Figure 3D and 3C and S14: There are differences in the chemical shift perturbation of the different amino acids. Additionally, in 3B some amino acids are not perturbed at all. Based on these findings it is very hard to understand how the authors come to the conclusion that there is no sequence-specific effect.

o "Amide 1H resonance positions as a function of temperature are a sensitive indicator of hydrogen bonding status. In general, temperature coefficients more negative than -4.5 ppb/K are observed for non-hydrogen bonded amides (e.g. amides open to solvent interaction, not engaged in protein-protein hydrogen bonds) whereas values more positive than -4.5 ppb/K are usually observed for amides participating in intramolecular hydrogen bonds (Cierpicki et al, 2002)."

Looking at Figure 5 only 4 residues show a shift over the described 4.5ppb/K threshold in presence of 1,6-hexanediol. The way

this experiment is described in the results section suggests that there is no change in solvation for most of the residues. Could the authors please address this point and revise the description of the findings?

o Figure 5B and 5C are hard to interpret as very little explanation is given on why the data analysis was performed this way. It would be beneficial for the readers to have more detailed explanation.

- The authors state: "To further test this hypothesis and exclude the possibility that the cleaved MBP tag or TEV protease interferes with the ability of hexanediol to dissolve these condensates, we used FUS RGG3 (purified apart from an MBP-tag) that phase separates readily upon addition of RNA."

o The description of the RGG3 purification in the method section is unclear and it looks like the purification of a RGG3 also is via an MBP-tag (though a His tag is also mentioned). The authors should clarify and correct the experimental description as necessary.

o One possible variable affecting FUS LLPS is the cleavage efficiency of the TEV protease, which is not shown in this manuscript. The authors should assess the cleavage efficiency of their TEV by performing a western blot.

o Comparing how the MBP tag or the TEV influences the full-length FUS cannot be answered by using another construct like the RGG3. Each protein construct might react differently to the presence of the MBP and the TEV. If the authors wish to stay in this line of argumentation, they should perform an additional purification step after the MBP cleavage to remove the TEV and MBP from the full length protein. One for instance could use magnetic beads with a His6-tag to achieve this. Of course, this additional purification step should be qualitatively assessed using western blot, for example.

- The authors claim that the presence of the PolyA RNA inhibits the action of alkanediols as solvents.

o While the droplets look mostly spherical, suggesting that they are indeed liquid, their dynamicity should be directly tested. For instance, one could perform a temperature-induced reversibility test or do serial dilution experiments to formally show that the droplets are dynamic.

o It is unclear why the authors chose PolyA as a model RNA. Their claim that the inhibited dissolution of the droplets by alkanediol isomers is due to the presence of electrostatic interactions should be validated with different RNAs as well, including scrambled sequences.

o While I understand the authors' view that PolyA RNA interactions are most likely electrostatic, one cannot exclude other interaction types. To support their claim, the authors should experimentally test this.

- "Moreover, our findings show a correlation between the atomic-level impact of alkanediol treatment on the chemical environment experienced by a disordered protein and the impact on phase separation."

The results in Figure 1D, 1E and SI2 indicate that all alkanediol isomers work similarly. However, to truly claim that they act exactly the same I recommend to repeat the same NMR experiment that was done with the 1,6-hexanediol, also with the 2,5-hexanediol. While the smaller effects of 2,5-hexanediol on BMCs might mask its effect on phase separation in bulk NMR experiments, this technical issue might be circumvented by using higher concentrations of the 2,5-hexanediol, as there seems to be a concentration dependence influence (as shown in Figure 1B).

Minor criticism:

- The authors should carefully re-read the manuscript and correct some spelling, formatting and grammatical mistakes.

Examples:

o 1-6 hexanediol (introduction) vs 1,6-hexanediol (results).

o Some numbers and units are not separated by a space.

o In the reference list some references seem to be formatted wrongly.

o Fulllength should be written as "full-length"

- "Low-complexity domain and fulllength FUS LLPS is decreased varyingly, while LLPS of FUS RGG-RNA condensates is even enhanced by some alkanediols."

This sentence is overstating what is seen in Figure 3. If the quantifications requested above confirms this, then the sentence would be justified. However, if this would not be the case, the authors should re-phrase and tone-down this claim.

- "Moreover, the formation or stability of some cytoplasmic and nucleoplasmic condensates, such as stress granules (SGs) (Wolozin & Ivanov, 2019) and the nuclear pore complex (NPC) are sensitive to the cellular uptake of some small organic aliphatic alcohols(Shulga & Goldfarb, 2003; Kroschwald et al, 2015a; Wheeler et al, 2016)."

The sentence should be changed to "Moreover, the formation and stability of some cytoplasmic and nucleoplasmic [...]"

- "For example, 1-6 hexanediol, 1-2 hexanediol and 1-5 pentanediol are capable of dissolving nucleopores (Ribbeck et al, 2002; Patel et al, 2007), RNA granules and cytoplasmic granules (Tulpule et al, 2021) by putatively disrupting weak hydrophobic interactions."

Please remove the "putatively disrupting weak hydrophobic interactions" as this was suggested, but not shown, in the Tulpule et al., 2021 study, and the current manuscript under revision actually demonstrates that this is not true.

- Some additional information in the introduction would be beneficial for the broad readership of EMBO Journal to better understand the work. Examples:

o LLPS, hydrogels, maturation, aggregation

o FUS domain overview and attributed functions thereof.

o A more in-depth description of the pipeline that is described by the authors to study phase separation disrupting compounds for therapeutic applications.

• "[...]nucleic acid-mimic biopolymers (Altmeyer et al, 2015),"

Can the authors specify what is meant with this statement?

• "Alkanediols of shorter carbon chain length such as butanediol and pentanediol reduced the extent of phase separation but did so less than hexanediols with hydroxyl groups at positions 1,6 and 1,2."

It would be helpful for readers with little chemical background to integrate the chemical structures in one of the supplementary figures.

• Figure 1D: For consistency the color code legend should be added in the panel as done for all other panels.

• Following figures are missing scale bars in some panels: 1A, 2A, S13. For consistency I advise the authors to always use the same style of scale bars (white vs black).

• "As we showed previously, addition of RNA (here we use polyadenylic acid (polyA) RNA) enhances phase separation."

Looking at the images in 2A this statement is not true of the untreated condition. Therefore the authors should revise this sentence.

• Could the authors please provide an image of the RGG3 domain without RNA as a comparison in figure 2C?

• "Importantly, we find that phase separation is modestly decreased but still present even with addition of aliphatic alcohols (Figure 2A) as seen by high turbidity values (Figure 2B)."

This sentence does not reflect the findings seen in figure 2A. The authors should revise this sentence.

• In Figure 2E it is very hard to see and distinguish the different colors. One possibility to make it more visible to the reader is to integrate an additional panel with a higher magnification of the starting points into the plot.

• In my opinion the authors should dedicate a small paragraph in the discussion on how alkanediols do not affect aggregates and hydrogels in context of their findings.

• "Our data show that the alkanediol-sensitivity of FUS condensates enriched in RNA is clearly reduced (Figure 2), underscoring that such electrostatic interactions mediated by arginine-specific contacts are not impacted by amphiphilic alcohols." Several studies in vitro and in cells have shown that one can abolish RNA-containing BMCs by using alkanediols. Many of those condensates are thought to contain RNA-protein interactions of electrostatic nature. Can the authors comment on the discrepancy between their study and these findings?

• "A series of studies on chromatin behavior in cells under the effect of 1,6-hexanediol or 2,5-hexanediol showed that these alcohols actually enhance interactions, condensing and suppressing the mobility of chromatin (Itoh et al, 2021)."

While it is true that this study shows one example of enhanced condensation, the same study also reports that several other biomolecular condensates containing RNAs are in fact dissolved with increasing hexanediol concentration. I therefore propose that the authors rewrite the sentence as follows:

"A series of studies on chromatin behavior in cells under the effect of 1,6-hexanediol or 2,5-hexanediol showed that these alcohols enhance interactions in some cases, condensing and suppressing the mobility of chromatin (Itoh et al, 2021)."

• "Finally, it will be interesting to directly probe why 1,6-hexanediol can disrupt phase separation of a histidine-rich disordered domain but not its interactions with RNA-polymerase II C-terminal domain (Lu et al, 2018)."

This statement is unclear to me. Could the authors please elaborate in this paragraph what exactly is meant?

• In general, I miss a more detailed discussion on the different types of RNA interactions that exist in FUS and how these findings are translatable to the behavior of FUS with regard to RNA binding and subsequent phase separation.

• The authors do not specify in which E. Coli strains the different protein constructs were expressed and how the constructs were stored. To improve reproducibility of the data, this information should be added.

• How were the proteins stored?

• There is information missing on the microscopy part of the material and methods. The readers will benefit from a more detailed description of these experiments. For example: the objective used is currently missing.

• Please add descriptions for the axes in 4B.

Referee #2:

The manuscript by Perdikari et al. explored how alkanediols regulates FUS liquid-liquid phase separation. As one of the widely used alkanediols, 1,6-hexanediol has been long used in dissolving nuclear pores, membraneless organelles, and disordered protein biomolecular condensates. But how it works remains poorly understood. In this work, the authors firstly evaluated the effect of 6 different diols on the phase separation of FUS full-length (FL), low complexity (LC) and RGG3 domain, respectively. Then, they applied NMR spectroscopy to study the interaction between diols and FUS. The results show that 1,6-hexanediol (1,6-HD) disrupts FUS phase separation mainly by forming protein-solvent hydrogen bonds and increasing FUS local motions, but not by disrupting the interaction with specific hydrophobic/aromatic residues. Altogether, this manuscript sheds light on the molecular mechanism underlying regulation of alkanediols in FUS phase separation. The work is interesting and is of potential importance to the field. However, to strengthen this work, the authors may need to analyze and interpret the data carefully, and conduct additional experiments according to my suggestion listed below.

Major concerns:

1. The authors examined the effects of 6 different alkanediols on phase separation of FUS full-length (FL), low complexity (LC) and RGG3 domain. They found that the phase separation of the FUS LC domain was most sensitive to the addition of different alkanediols, as manifested by the disruption of the droplets upon addition of diols. However, the droplets formed by FUS RGG3 and FL with RNA (polyadenylic acid, polyA) are strongly resistant to all types of alkanediols. Moreover, 1,2-cyclohexanediol and 1,2-CHD promote the phase separation of FUS RGG3 and FL with RNA. To fully support these claims, the authors need to present the DIC micrographs with a higher resolution and magnification, since the current DIC micrographs are too small to be clearly observed and compared.

Moreover, the DIC micrographs do not match well with the quantitative turbidity data. For example, the data in Figure 1A showed that the inhibitory effect of addition of 5% 1,5-pentanediol (1,5-PD) was significantly stronger than that of 5% 1,4-butanediol (1,4-BD), but in Figure 1B, the inhibitory effects of the two compounds are opposite. In addition, when describing the effect of the addition of alkanediol on the co-separation of FUS FL and RNA, the authors described that "phase separation is modestly decreased but still present even with addition of aliphatic alcohols", but I can't get the same conclusion from the bottom panel of Figure 2A (e.g., 5% 1,2-hexanediol (1,2-HD), 1,6-HD, and 1,5-PD). The authors need to address the inconsistency.

2. The authors explored whether the different effects of alkanediols on FUS LC phase separation could be attributed to particular contacts between different alkanediols and FUS by performing NMR titrations. The authors conclude that "all alkanediols have similar mechanisms of action" and "do not find evidence for specific interactions caused by particular alkanediols or isomers". However, the NMR data in Figure 3B and SI Figure 2 indicate that the N-terminal residues (~ 6-12) show obvious larger chemical shift deviations than the other residues upon addition of 5% 1,6-HD. Similar results were observed for the titrations of FUS LC by addition of 5% 1,2-HD and 1,2-CHD.

In addition, two Tyrosine residues at the terminal of FUS LC, e.g., Y6 and Y161, revealed obvious larger chemical shift deviations upon addition of 5% 1,6-HD, as well as addition of 5% 1,2-CHD, 5% 1,2-HD and 5% 1,4-BD, but smaller changes with 5% 2,5-hexanediol (2,5-HD) and 5% 1,5-PD. The different slopes in Figure 1D were determined by the point with the largest change, which also suggested the critical role of chemical shift deviations of certain amino acid that were probably tyrosine. Thus, whether these residues play a role in phase separation of FUS-LC, and mediating the interaction with different diols? The authors might need to carefully analyze the titration results to address.

3. The authors measured ¹HN resonances over a range of temperatures by using NMR to examine whether position-specific changes in hydrogen bonding in FUS LC occur in the presence of 1,6-hexanediol. The data showed that serine and tyrosine display the largest difference upon adding 1,6-hexanediol. From the view of experimental design, other diols, at least 2,5-HD should be included as a negative control. Additional diols with the same experiment are appreciated, which would be useful to understand the differences caused by different alkanediols, and certainly strength the conclusions.

Minor concerns:

1. In SI Figure 1B, "12-HD ..." should be "1,2-HD ...".

2. The authors selected six types of alkanediols to study the mechanism of their effects on FUS phase separation, taking into account the length of the alkane chain (4-6 C), the conformation of the alkane chain (linear and cyclic), and the position of the hydroxyl group. It would be meaningful to analyze the effect of different types of alkanediols on phase separation from the perspective of alkanediol structure. Even if there is no more data to support it, additional discussion could be added.

Referee #3:

The paper by the Fawzi lab addresses how different alcohols impact on protein condensation. This is an interesting topic as many alcoholic compound are used to modulate protein condensation. In their manuscript the authors combine light microscopy and OD600 measurements to determine the degree of phase separation (these methods only show limited correlation though,

see below). Furthermore, the authors exploit NMR spectroscopy and try to detect the effect of 1,6 HD on the chemical shifts of FUS.

In general, the paper is very descriptive (experiments are performed and the results are presented, but not always interpreted in the light of a mechanism or model). In addition, there appear to be no real new hard findings or conclusions: the alcohols sometimes enhance, sometimes decrease LLPS and there are no specific contacts between the compounds and FUS. In that light, the paper fails to provide a clear message. (The intro and discussion are well written and general, but the findings seem not to add much to our understanding of how alcohols influence LLPS, despite the title being "molecular insights"). In addition, I have serious doubts regarding the referencing of the NMR spectra and the extracted CSPs (see below). In that light, I think that the manuscript is not suitable for EMBO journal.

In addition, I have a (larger) number of major and minor remarks.

Figure S1: the panels for 5% 2,5 HD are the same at 2.5 and 5 hours. This must be a mistake.

"with the latter diol being the least efficient among the series of hexanediol isomers being tested". This seems to be the case in figure 1A, but not in figure S1A. How representative are the shown images? How many times were these experiments independently performed?

"both 2.5% and 5% 2,5-HD resulted in lower concentration of protein remaining in the dispersed phase and higher turbidity". Are the results of Fig 1A and Fig 1B in full agreement? Seems to me that the 2,5 HD is indeed in agreement with panel A, but the 1,4 BD is for sure not in agreement (5% 1,4 BD behaves the same as e.g. 1,6 HD and 1,2 CHD in panel B. In panel A %5 1,4 DB behaves different as e.g. 1,6 HD and 1,2 CHD). Also, 2,5 HD and 1,5 PD are the same in B, but different in A. This raises the question as to how reproducible the experiments are, or as to how relevant the readouts (seeing bubbles, measuring some concentration, turbidity) are.

Figure 1C: is that 5% of the additives?

Fig S2: I would like to see some of the raw data. All CSPs are in one direction. This indicates a general shift of the resonances in a specific direction. Are the spectra referenced using e.g. DSS? General resonance shifts are of course expected if 5% (or 2.5%) of another solvent is added. The same happens when measuring spectra at different temperatures or different pressures. Those general CSPs are, however, not interesting as they don't report on any interactions, but merely in changes in the general solvent properties.

Along the same lines, the authors should perform control experiments to differentiate between CSPs caused by "specific" interactions and by changes in the solvent properties. One way of doing this would be to use a folded protein. In case residues not on the surface behave, residues in the core and the FUS IDP resonances respond the same to the solvents there is clearly sign that the CSPs are not directly caused by interactions between the alcohols and the proteins.

Fig S2: error bars are definitely needed in all panels (A-G).

"In other words, the sequence-specific effects are largely the same for all diols tested." . I don't understand how anything about sequence specificity could be extracted from the plots in Fig. 1D.

Fig 1E: I assume that this is at 5% additive. The correlation at 2.5% should also be shown as there the spread in the concentrations (Fig 1B) is much larger. Please add error bars in the plot. Looking at Fig 1B, the error in the concentrations is large compared to the differences between the additives.

"Hence, the changes in residue-by-residue perturbation of the chemical environment are strongly correlated with the mechanism of action." This statement is far too strong I think (see above). There is no insights into any kind of mechanism of action here.

"predicted water solubility (logP) of each diol" Why should there be a correlation? 1,4-Butanediol can be mixed with water at any ratio and 1,6-Hexanediol is soluble up to 50% (w/v). According to the legend P is the octanol partition coefficient (how much of the compound goes into octanol and how much into water) I don't see how this tells something about water solubility of the different compounds. P only tells something about the difference in solubility of the compounds in octanol and in water.

In addition to my confusion regarding the relevance of the partition coefficient I disagree with the statement "similar predicted partition coefficient such as 1,2-hexanediol (which also has significantly lower experimentally-determined solubility than 1,6 hexanediol) and 2,5-hexanediol show different impacts on LLPS." Why are the log(P)s of 1,2 HD (0.65) and 2,5 HD (0.4) very similar, but is the log(P) of 1,6 HD (0.3) significantly different? A difference of 0.25 can not be called similar, while a difference of 0.1 is called significant.

"Importantly, we find that phase separation is modestly decreased but still present even with addition of aliphatic alcohols" To me it looks like there are more "bubbles" after adding any of the alcohols in the presence of RNA (I don't see a decrease in Fig. 2A).

Fig. 2A (bottom row) and Fig 2B (right panel) are clearly inconsistent to me. (maybe the turbidity in Fig 2B decreases over time in the presence of alcohols as there is larger condensates that just sink to the bottom). Like this it makes no sense to me.

For all panels in Fig 2 a control with only RNA should be done, maybe the RNA phase separates too, independent of the protein.

Again, I see no correlation between Fig 2C and Fig 2D. If at all, I see less bubbles for 5% 1,2 CHD than for no alcohols in Fig 3C, whereas Fig 3D indicates more condensation for 5% 1,2 CHD than for no alcohols. The lack of correlation between complementary experiments is disturbing.

". Imaging the RGG3- RNA condensates over time showed that droplets remained spherical " Do the droplets remain liquid like, or are they forming hydrogels that can no longer dissolve. In case the droplets matured, it is clear that alcohols have no effect and that they remain the same in shape over time.

Fig 3A and 3B. These spectra should be references properly. 95% of the shifts go in one direction. This will mask residue specific differences (in case there are any). The CSPs that are extracted here are not relevant I would say.

Fig 3C, D. See my remark above. In the absence of proper referencing these data appear not very valuable to me.

Fig S4 C and D. The plot contain 37% bars for residues that are only present once in the sequence. As the spreads are very large (as can be seen for the residue types that occur often), this is not very informative.

"Hence, the elevated R2 values likely report on dynamic interactions leading to phase separation...". One easy (other) explanation is that the viscosity of the solvent changed by the addition of the alcohol. This then leads to changes in the relaxation rates, fully independent of phase separation effects. One could check this by e.g. recording data on an IDR that does not undergo LLPS.

" the 1H 15N NMR data do not conclusively show if 1,6-hexanediol makes specific contacts that mediate its action or if it primarily alters the solvent environment." I would say that the spectra show that there as no specific contacts. The wording of this sentence is misleading ("do not conclusively show" → "don't show").

A technical question regarding the buffers. Do the authors take the 0% additive buffer and then pipette in the additives directly? This would lower the salt concentration of course. Or do the authors prepare a buffers such that the buffers with additives have exactly the same same concentration as the buffers without additives?

I am confused with the errorbars in Fig. 4D. For Ala, there are 2 datapoints (around 0.002 and around 0.05 or so). The average of those makes sense (around 0.026), but the displays error bar (s.d. I assume, it is not mentioned), makes no sense. How is this calculated?

Again for Fig 4, the real differences in chemical shifts might be obscured in a general shift of the spectra due to changes in the solvent.

Clearly, I agree, there is no correlation between the amide nitrogen and carbonyl carbon CSPs. But why should there be one? Has this ever been systematically investigated. Of course when there is a single well defined binding site the 15N and 13C CSPs are localized and then correlate, but for the very small shifts here I see no reason for this. Please explain.

"show a wide range of chemical shifts " I assume the authors mean "show a wide range of chemical shift perturbations".

It took me a while to figure out what the labels in panels 4E and F (x-axis) are. Please make this clearer.

I strongly disagree with calling Serine, Threonine or Glutamine aliphatic amino acids. Those are polar, uncharged amino acids. Aliphatic amino acids are e.g. Leucine, Valine and Isoleucine (that is a biochemistry I class)

"Surprisingly, many of the backbone 13CO chemical shifts are larger than the 13C aromatic (i.e. tyrosine) and aliphatic sidechain shift perturbations". What is the conclusion of the authors regarding this statement?

Fig 5B definitely needs error bars (that are likely very large).

Fig 5: some primary data should be shown.

Fig 5. Did the authors take into account that different amino acids behave differently with temperature and have different pKa values? The analysis here seems quite limited.

Panel 5C is not of much value I would say.

"Still, our results show that NMR-based observations can provide unique insight into the molecular origins of phase separation modulation and could contribute to the rational design of possible therapies for altering disordered protein phase separation."
What molecular origins of phase separation have been found in this paper?

Response to Reviewers:

We appreciate the reviewers' comments and have taken their suggestions to thoroughly revise the manuscript. Importantly, we have refocused the text on the main message – that popular simplistic conceptions of alkanediols interacting with hydrophobic groups directly as “mini-detergents” and that distinct alkanediols isomers have qualitatively different effects are simply not true. The new finding is that alkanediols do not work by specific aromatic residue interaction and in fact seem to disrupt both hydrogen bonding and water interactions – neither of which are “pi” specific. We emphasize that these insights are consistent with the view many interactions contribute to disordered protein phase separation, not just pi interactions. We have also altered the title to more accurately reflect the findings.

We address each comment below.

Referee #1:

Perdikari et al., try to elucidate the mechanism of dissolution of biomolecular condensates (BMC) by alkanediol isomers. They show that the alkanediol isomers act all similarly as solvation agents by favoring protein-solvent hydrogen bond interactions rendering them unsuitable for therapeutic applications. They additionally suggest that some alkanediol can no longer disrupt FUS BMC that contain polyA RNA showing the limitations of alkanediols to study protein-RNA phase separation.

This study represents an important step towards understanding the mechanistic role alkanediols have on BMCs. By showing that the tested alkanediol isomers have all similar functions they demonstrate that many studies in the field are indeed comparable, which until date has only been assumed. While one could argue that this paper is of technical nature, its importance lies in shattering the recent criticism for using alkanediol isomers as phase separation inhibitors. The field will benefit by the detailed and methodological analysis presented here. For this reason, I fully support publication of this work in EMBO Journal, after revision to address the following points for criticism.

Major points of criticism:

- Some qualitative statements are made throughout the manuscript; however, no quantification is shown in support of these statements. In particular, the authors should quantify the number of droplets and their perimeter and apply statistical analysis to the following panels: 1A, 2A, 2C, S11A, S13.*

We agree with the reviewer that quantification is important. But we disagree that quantifying droplet number and perimeter is the best way to quantify phase separation as droplet number and area is measured on the surface (not in the bulk) and this change as the droplets fuse, fall, and settle over time. The diols can affect the speed of this as well as the wetting of the surface. Therefore, image quantification does not directly report on phase separation and is challenging to time perfectly the same. Given that we do not have the means to image droplets in dozens of conditions identically and we support the quantification of phase separation with other means, we have instead chosen to refer the reader to other forms of quantification that we do perform. We do provide and have updated in places the quantification of phase separation by measuring the saturation concentration in Figure 1B and Figure 1D (former Figure 1B) which provides quantification for the assemblies formed in Figure 1A while turbidity data (vs time) is provided for conditions like in 2A (in 2B), 2C (in 2D) as a means of quantifying -- as all quantification of droplet imaging is time dependent and therefore much less precise than a spectroscopic characterization. Appendix Figure S1D and Appendix Figure S4 are imaged at later times and explicitly only used qualitatively.

• Many DIC images seem to have been recorded with different illumination strengths. If this is the case, the authors should repeat these experiments using the same illumination strength for all conditions.

We appreciate the reviewer comment and have repeated some data and replaced several images in Figure 1. However, illumination differences in our DIC images do not take away from the purpose of the images as qualitative demonstration of the presence of droplets. We do not make quantitative conclusions from these images and therefore we are confident the conclusions drawn from the images (the presence of liquid droplets) are solidly supported, especially with the many new images presented.

• No statistical analysis has been shown for quantitative Figures. The authors should perform and integrate statistical analysis of the following Figures: 1B, 1C at t=0, 2B at t=0, 2D, S11B.

We agree that in some places statistical analyses are helpful and we use them to evaluate correlations. However we have repeated the quantitative data in Figure 1B and the difference is much larger than the standard deviation. We have also been editorially advised that statistical tests are discouraged for technical replicates by data use guidelines. We have substantially revised the figures shown the individual data points for what is now Appendix Figure S1C, we are emphasizing in 2D that hexanediol does not have the ability to disrupt these (and given the differences vs time and the amount of time required to set up the assay, we do not want to test for statistical difference here as the result is intended to be qualitative), and in S1G (former S1B) these are predicted values from a database with no experimental uncertainty provided simply as a guide in the SI for the readers.

• As purity affects the final concentration of biomolecules and BMC formation is concentration dependent, I think it is important to show the purification analysis for all the protein constructs used in this study.

We appreciate the reviewer's suggestion. The purity for all proteins by absorbance at 260/280 is provided. We have added the statement:

All protein preparations showed UV absorbance ratio, $A_{260\text{ nm}}/A_{280\text{ nm}}$, of less than 0.6, indicating minimal to no residual nucleic acid contamination.

FUS LC does not stain with Coomassie and does not bind SDS so a gel purity is not clear – we have described this purity extensively in previous papers (see Burke et al. Mol Cell 2015 and Murthy et al NSMB 2019). FUS RGG3 purity is demonstrated in Murthy et al NSMB 2021. We have added a gel showing high purity of FUS full-length (see below Appendix Figure S3). We also refer readers to our previous papers where we have performed extensive NMR analysis on these constructs which show the level of purity is high.

• For 2,5% and 5% 2,5-HD there seems to be a difference in the effectiveness of droplet dissolution. Several concentrations of alkanediol isomer have been used in the field over the years (usually ranging from 1-10% w/v alkanediol concentration). It would, therefore, be particularly useful if the authors per-formed a dose response ranging from 1 to 10% for all alkalediol isomers used in this manuscript.

We have now added an evaluation of the saturation concentration as a function of 1,6 hexanediol concentration from 0.5% to 5% (Figure 1B). We did not go above 5% as for FUS LC we cannot generate easily a higher concentration of FUS LC – i.e. the saturation concentration becomes extraordinarily high at about 5% hexanediol. We show that for 0.5% to 2.5% the response of the saturation concentration is linear. We also show that the correlation between 1% and 2% hexanediol is linear for all diols tested. Therefore we did not provide additional

evaluation for all diols at all concentrations as the amount of effort and sample and replicates needed became too great and we demonstrated effect linearity already several ways.

• In Figure 2B 1,2-HD treated samples in the RNA+ condition shows an increase in turbidity. While a microscopy-based timeline has been established for diols that do not disrupt phase separation the other alkanediols are missing. For completion, the authors should perform microscopy-based timeline recordings of all the diols with and without RNA.

We agree with the reviewer that the curve in Figure 2B for 1,2 HD samples with RNA shows a different shape where the turbidity stays approximately the same. As part of investigating the effect of diols on TEV cleavage to separate MBP from FUS full-length, we realized that 1,2 HD prevents TEV cleavage. Therefore, we attribute this difference observed in figure 2B to lack of cleavage. The addition of RNA induces phase separation of the FUS proteins before cleavage in these conditions. We have added these new data in Appendix Figure S3.

• In Figure 1 and Figure 2 the images taken by DIC do not necessary reproduce perfectly the later findings by turbidity measurements and supernatant protein concentration determination. For example:

o 1B: The concentration difference between 5% 2,5-HD and 5% 1,5-PD is minimal, however, the images in 1A show a clear difference between the 2,5-HD and 1,5-PD conditions. Can the authors explain this discrepancy?

We sought to extensively address this reviewer comment and we thank the reviewer for pointing this out. To restate, the apparent discrepancy is that the apparent saturation concentration appears similar but the droplets do not appear present. At these high concentrations of diols we do not see evidence of significant amount of droplets (no apparent turbidity and no images of droplets) after mixing, but that there seemed to be “missing” protein from the supernatant (the values in the original Figure 1B for saturation concentration should have been the same as in urea if we are above the saturation concentration. We also realized that if we measured saturation concentration by spinning longer we found less protein left in supernatant for the diol samples (but not the urea control or the 0% diol phase separated condition), suggesting that there may be issues with protein adsorption to the walls of the tube in the high diol conditions. Finally, we realized that labeling of 1,4-BD and 2,5-HD (due to stock labeling issue) were inadvertently flipped (which we discovered by assay repetition and repetition of all NMR data with all diols). Therefore, we reworked the phase separation assay to stay in the regime where there are clearly droplets and test the effect of the diols on the amount of protein that phase separates -- we did the experiment at a higher total FUS LC concentration of 300 μ M and with a slightly lower amount of all diols 1% and 2% (note we also did a full range of concentrations for 1,6 hexanediol to ensure we cover the range of diols used in the literature – and we show that the effect of diol concentration on saturation concentration is linear). The new data show that the diols have distinct effects on phase separation and are still correlated with the magnitude of the NMR derived effect as we suggested before. We point out that for the 2.5% hexanediol condition there was much less of a possible discrepancy suggesting that a lower diol concentration (away from the range where there is complete prevention of phase separation) is the best means to quantify the effect.

o Discrepancy between 1B and 1C: If you compare 5% 2,5-HD with 5% 1,5-PD then the absorbance should reflect the same trend as in 1C. Why doesn't it? Could the author please comment?

We have now repeated the assays previously in Figure 1B (see above) so this is now resolved as well. We appreciate the reviewer's careful reading.

o Figure 1A and 2A of the untreated samples should look the same. Can the authors please comment on the differences between these images?

We note that they should not look the same - Figure 1A is FUS LC while Figure 2A is FUS full-length at different concentrations, so the untreated samples should not look the same.

It would be helpful to show an experimental schematic on how this data was acquired for readers to better understand the findings.

We thank the reviewer for this suggestion and have added schematics in Figure 1

• Some NMR experiments and their interpretation are hard to understand. It would be beneficial for the broad readership of EMBO Journal - especially those unfamiliar with NMR - to elaborate on the experimental readouts and the conclusions in more depth. Some examples:

o Please annotate in the Figure legend for differential chemical shift perturbation plots that blank residues mean that there was no shift detected.

We have added additional text.

Gaps indicate positions with no $^1\text{H}/^{15}\text{N}$ resonance (e.g. proline residues that have no backbone ^1H attached to nitrogen) or with overlapped resonances that were not resolved in these experiments.

o Comment on the strength of the chemical shift perturbation in Figure SI2.

We have added the following comment in the text:

Overall, the observed chemical shift perturbations are small compared to what would be expected for a tight-binding molecule, though they are larger than observed when FUS LC is in the presence of excess Karyopherin- β 2 (Yoshizawa et al, 2018). We note several interesting features and analyze the sequence specific spectral changes below (see below).

o "In other words, the sequence-specific effects are largely the same for all diols tested. The observation that the series of alkanediols show a quantitative gradient of effects on LLPS and chemical environment with qualitative similarity suggest that all alkanediols have similar mechanisms of action."

We have reworked this statement to say "In other words, the sequence-specific effects are highly similar for all diols tested, just with a different magnitude, suggesting that all alkanediols affect phase separation by a similar mechanism."

Figure 1D: both 1,2-HD and 1,5-PD show clear outliers. Could the authors comment on those outliers and explain what this might mean for the effectiveness of these diols? Interestingly, 1,2-HD is a good dissolvent while 1,5-HD is not. Could the outliers also explain the difference in performance? Might there be a sequence specific effect?

We appreciate the reviewer's comment, and we note that outliers happen due to uncertainty in assigning peaks in complex spectra. We have reanalyzed the spectra and removed points arising from peaks that were ambiguous, overlapped, or misassigned. Hence, we do not think that these outliers in the raw data explain the difference in phase separation, though we do highlight (see response to comments below) that there are some sequence-specific hexanediol effects.

Figure SI2 shows that some residues are more affected (bigger CSP) while others are not affected at all (empty lanes in the residue space) and there are differences in the CSP pattern between the alkanediol isomers. Is this not a direct indication that some residues are more affected than others?

We agree with the reviewers that some residues are more affected than others. This observation may be due to slight differences in conformation preferences in the structure of FUS LC at each site (see Burke et al Mol Cell 2015) that are differentially altered (due to local conformations as well as due to intramolecular collapse and interactions in FUS LC (see Ryan et al Mol Cell 2018)). We also agree that in FUS there are some differences between the diols, however the correlation plots (for which we have now added the Pearson correlation coefficients) show that the effect of the different diols are highly similar in pattern and differ mostly in magnitude. Taken together, it is quite possible that some residues are more affected than others. We will revise text to make that clear. There is a variance between residues in each residue type. However, the magnitude of the chemical shift can depend on many factors but the data do not suggest major differences between diols.

o "We quantified the chemical shift perturbations between the control and 1,6-hexanediol conditions and observed weak non-specific interactions throughout FUS SYGQ LC domain." Figure 3D and 3C and SI4: There are differences in the chemical shift perturbation of the different amino acids. Additionally, in 3B some amino acids are not perturbed at all. Based on these findings it is very hard to understand how the authors come to the conclusion that there is no sequence-specific effect.

One main message of the paper is that the alkanediols do not work by direct specific interaction with only one or a few residue types. However, as described in our response above, there are certainly differences at each site. We have revised this section of the text to make clear that there are sequence-specific interactions:

We quantified the chemical shift perturbations between the control and 1,6-hexanediol conditions and observed small, sequence-specific chemical shift perturbations common across different alkanediols throughout FUS SYGQ LC domain (**Figure 3B**).

o "Amide 1H resonance positions as a function of temperature are a sensitive indicator of hydrogen bonding status. In general, temperature coefficients more negative than -4.5 ppb/K are observed for non-hydrogen bonded amides (e.g. amides open to solvent interaction, not engaged in protein-protein hydrogen bonds) whereas values more positive than -4.5 ppb/K are usually observed for amides participating in intramolecular hydrogen bonds (Cierpicki et al, 2002)." Looking at Figure 5 only 4 residues show a shift over the described 4.5ppb/K threshold in presence of 1,6-hexanediol. The way this experiment is described in the results section suggests that there is no change in solvation for most of the residues. Could the authors please address this point and revise the description of the findings?

The reviewer brings up a good point that the text is not clear. We have revised the text to emphasize that, while only a few residues shift across this threshold, the threshold is arbitrary and the importance here is that FUS LC shows a change towards increased solvation across the whole protein, but that some sites are affected more than others.

o Figure 5B and 5C are hard to interpret as very little explanation is given on why the data analysis was performed this way. It would be beneficial for the readers to have more detailed explanation.

We appreciate the reviewer's critique. We have now better explained Figure 1B in the text and we have moved Figure 5C to the supplemental (it is just a measure of the goodness of fit of the values in Figure 5A – all values are well fit explained by the linear fit).

• The authors state: "To further test this hypothesis and exclude the possibility that the cleaved MBP tag or TEV protease interferes with the ability of hexanediol to dissolve these

condensates, we used FUS RGG3 (purified apart from an MBP-tag) that phase separates readily upon addition of RNA."

o The description of the RGG3 purification in the method section is unclear and it looks like the purification of a RGG3 also is via an MBP-tag (though a His tag is also mentioned). The authors should clarify and correct the experimental description as necessary.

We have now made clear that the RGG3 is purified to remove the MBP and TEV explicitly before the experiment.

o One possible variable affecting FUS LLPS is the cleavage efficiency of the TEV protease, which is not shown in this manuscript. The authors should assess the cleavage efficiency of their TEV by performing a western blot.

The reviewer makes an excellent point. To summarize the data, we show that all diols prevent phase separation of FUS full-length without RNA as seen in Figure 2B left, but phase separation is enhanced and the diols do not prevent separation with RNA. We have now added a gel (Appendix Figure S3) showing that that MBP-FUS is cleaved by TEV at similar rates in the absence and presence of the diols, except for in 1,2-HD – which also explains the differences observed for 1,2-HD in the adjacent figure panel.

o Comparing how the MBP tag or the TEV influences the full-length FUS cannot be answered by using another construct like the RGG3. Each protein construct might react differently to the presence of the MBP and the TEV. If the authors wish to stay in this line of argumentation, they should perform an additional purification step after the MBP cleavage to remove the TEV and MBP from the full length protein. One for instance could use magnetic beads with a His6-tag to achieve this. Of course, this additional purification step should be qualitatively assessed using western blot, for example.

We agree with the reviewer and have removed this line of argumentation and references to effect of MBP and TEV.

• The authors claim that the presence of the PolyA RNA inhibits the action of alkanediols as solvents.

We clarify that we are not claiming this. We are claiming that the type of contacts that stabilize FUS RGG phase separation with RNA are not perturbed by alkanediols. This makes sense as we are showing here that alkanediols appear to alter FUS LC phase separation by altering the hydrophobic effect and we have previously shown that hydrophobic interactions contribute to phase separation of FUS LC (Murthy et al NSMB 2019). FUS RGG phase separation with RNA has significant contribution from electrostatic interactions (Murthy et al NSMB 2021), hence this difference in effect of hexanediol on phase separation of different parts of FUS is reasonable. A similar point was recently made in <https://www.nature.com/articles/s41467-022-35430-y> where condensates that depend on electrostatic interactions are not disrupted by hexanediol.

o While the droplets look mostly spherical, suggesting that they are indeed liquid, their dynamicity should be directly tested. For instance, one could perform a temperature-induced reversibility test or do serial dilution experiments to formally show that the droplets are dynamic.

In previous work, we demonstrated that the spherical droplets of FUS LC are liquid (Burke et al. Mol Cell 2015). Although there is no evidence that the diols change this nature and we see clear evidence of droplet behavior staying the same, we agree with the reviewers that the material properties could be interesting to study but the droplets we see are dynamic and it is beyond the message of this paper to quantify the material properties of the droplets in detail in every condition. Surely the viscosity will be different as the solvent changes the point on the phase diagram – this is not trivial to measure in every condition is not the point of this work.

o It is unclear why the authors chose PolyA as a model RNA. Their claim that the inhibited dissolution of the droplets by alkanediol isomers is due to the presence of electrostatic interactions should be validated with different RNAs as well, including scrambled sequences.
o While I understand the authors' view that PolyA RNA interactions are most likely electrostatic, one cannot exclude other interaction types. To support their claim, the authors should experimentally test this.

We chose polyA as a simple model for non-specific RNA interactions. FUS RGGs are well known to interact with a wide variety of RNA

<https://academic.oup.com/nar/article/45/13/7984/3855590>

including polyA, with little apparent specificity.

We previously tested the enhancement of FUS phase separation by RNA extract (that contains arbitrary RNAs) (Monahan et al EMBO J 2017; Murthy et al NSMB 2021). It would be interesting to test a wide array of different RNAs but it is not within the scope of this effort. We have revised the text to make clear we are not making detailed claims on the mechanism of hexanediol inability to disrupt these condensates. However, it is relevant for us to note that these condensates are not affected.

• "Moreover, our findings show a correlation between the atomic-level impact of alkanediol treatment on the chemical environment experienced by a disordered protein and the impact on phase separation."

The results in Figure 1D, 1E and SI2 indicate that all alkanediol isomers work similarly. However, to truly claim that they act exactly the same I recommend to repeat the same NMR experiment that was done with the 1,6-hexanediol, also with the 2,5-hexanediol. While the smaller effects of 2,5-hexanediol on BMCs might mask its effect on phase separation in bulk NMR experiments, this technical issue might be circumvented by using higher concentrations of the 2,5-hexanediol, as there seems to be a concentration dependence influence (as shown in Figure 1B).

We did do the NMR titration experiment ^1H ^{15}N HSQC with all diols, as shown in Figure 1E and Appendix Figure S2. We simply show in Figure 3A,B,C,D only the 1,6 data fully (the analyzed data for 3A-D are effectively all summarize in Figure 1E). We elected not to perform the entire complement of NMR experiments (e.g. the ^{13}C NMR) as we do not see evidence that they will show more information.

Minor criticism:

• The authors should carefully re-read the manuscript and correct some spelling, formatting and grammatical mistakes. Examples:

o 1-6 hexanediol (introduction) vs 1,6-hexanediol (results).

o Some numbers and units are not separated by a space.

o In the reference list some references seem to be formatted wrongly.

o Fulllength should be written as "full-length"

We have addressed these minor issues and we thank the reviewer for their suggestions.

• "Low-complexity domain and fulllength FUS LLPS is decreased varyingly, while LLPS of FUS RGG-RNA condensates is even enhanced by some alkanediols."

This sentence is overstating what is seen in Figure 3. If the quantifications requested above confirms this, then the sentence would be justified. However, if this would not be the case, the authors should re-phrase and tone-down this claim.

We have rephrased this section of the manuscript to ensure that the claims are supported.

- *"Moreover, the formation or stability of some cytoplasmic and nucleoplasmic condensates, such as stress granules (SGs)(Wolozin & Ivanov, 2019) and the nuclear pore complex (NPC) are sensitive to the cellular uptake of some small organic aliphatic alcohols(Shulga & Goldfarb, 2003; Kroschwald et al, 2015a; Wheeler et al, 2016)."*

The sentence should be changed to "Moreover, the formation and stability of some cytoplasmic and nucleoplasmic [...]"

We have extensively revised this section to be more accurate.

- *"For example, 1-6 hexanediol, 1-2 hexanediol and 1-5 pentanediol are capable of dissolving nucleopores (Ribbeck et al, 2002; Patel et al, 2007), RNA granules and cytoplasmic granules (Tulpule et al, 2021) by putatively disrupting weak hydrophobic interactions."*
Please remove the "putatively disrupting weak hydrophobic interactions" as this was suggested, but not shown, in the Tulpule et al., 2021 study, and the current manuscript under revision actually demonstrates that this is not true.

We have updated the text to clarify by removing the references to hydrophobic mechanism when referring to that study.

- *Some additional information in the introduction would be beneficial for the broad readership of EMBO Journal to better understand the work. Examples:*
 - o *LLPS, hydrogels, maturation, aggregation*
 - o *FUS domain overview and attributed functions thereof.*
 - o *A more in-depth description of the pipeline that is described by the authors to study phase separation disrupting compounds for therapeutic applications.*

We have added helpful background information via text and references.

- *"[...]nucleic acid-mimic biopolymers (Altmeyer et al, 2015),"*
Can the authors specify what is meant with this statement?

We agree this wording was unclear. We were referring to poly(ADP-ribose) and we have now said this explicitly.

- *"Alkanediols of shorter carbon chain length such as butanediol and pentanediol reduced the extent of phase separation but did so less than hexanediols with hydroxyl groups at positions 1,6 and 1,2."*

It would be helpful for readers with little chemical background to integrate the chemical structures in one of the supplementary figures.

We think this is an excellent idea and we have done so.

- *Figure 1D: For consistency the color code legend should be added in the panel as done for all other panels.*

We have now added this.

- *Following figures are missing scale bars in some panels: 1A, 2A, SI3. For consistency I advise the authors to always use the same style of scale bars (white vs black).*

We have now ensured the images have scale bars.

- *"As we showed previously, addition of RNA (here we use polyadenylic acid (polyA) RNA) enhances phase separation."*

Looking at the images in 2A this statement is not true of the untreated condition. Therefore the authors should revise this sentence.

We have changed this statement.

- *Could the authors please provide an image of the RGG3 domain without RNA as a comparison in figure 2C?*

We have already reported on the lack of phase separation of RGG3 alone in Murthy et al NSMB 2021 and we now clarify this and make reference to it.

- *"Importantly, we find that phase separation is modestly decreased but still present even with addition of aliphatic alcohols (Figure 2A) as seen by high turbidity values (Figure 2B)."*
This sentence does not reflect the findings seen in figure 2A. The authors should revise this sentence.

We have revised this statement for clarity and accuracy.

- *In Figure 2E it is very hard to see and distinguish the different colors. One possibility to make it more visible to the reader is to integrate an additional panel with a higher magnification of the starting points into the plot.*

We have now updated the data and we also clarify that the main message here is that the different points are not that different and that RNA addition can induce phase separation.

- *In my opinion the authors should dedicate a small paragraph in the discussion on how alkanediols do not affect aggregates and hydrogels in context of their findings.*

We have now added to the discussion regarding alkanediols and aggregation.

- *"Our data show that the alkanediol-sensitivity of FUS condensates enriched in RNA is clearly reduced (Figure 2), underscoring that such electrostatic interactions mediated by arginine-specific contacts are not impacted by amphiphilic alcohols."*

Several studies in vitro and in cells have shown that one can abolish RNA-containing BMCs by using alkanediols. Many of those condensates are thought to contain RNA-protein interactions of electrostatic nature. Can the authors comment on the discrepancy between their study and these findings?

- *"A series of studies on chromatin behavior in cells under the effect of 1,6-hexanediol or 2,5-hexanediol showed that these alcohols actually enhance interactions, condensing and suppressing the mobility of chromatin (Itoh et al, 2021)."*

While it is true that this study shows one example of enhanced condensation, the same study also re-ports that several other biomolecular condensates containing RNAs are in fact dissolved with increasing hexanediol concentration. I therefore propose that the authors rewrite the sentence as follows:

"A series of studies on chromatin behavior in cells under the effect of 1,6-hexanediol or 2,5-hexanediol showed that these alcohols enhance interactions in some cases, condensing and suppressing the mobility of chromatin (Itoh et al, 2021)."

We have made the suggested change and clarified that some BMCs are susceptible to alkanediols and some are not, suggesting that RNA-containing BMCs have some contribution to assembly that is not just RNA-RGG region interaction.

- *"Finally, it will be interesting to directly probe why 1,6-hexanediol can disrupt phase separation of a histidine-rich disordered domain but not its interactions with RNA-polymerase II C-terminal domain (Lu et al, 2018)."*

This statement is unclear to me. Could the authors please elaborate in this paragraph what exactly is meant?

We have revised this section.

- *In general, I miss a more detailed discussion on the different types of RNA interactions that*

exist in FUS and how these findings are translatable to the behavior of FUS with regard to RNA binding and subsequent phase separation.

We have now addressed this in the text.

• The authors do not specify in which E. Coli strains the different protein constructs were expressed and how the constructs were stored. To improve reproducibility of the data, this information should be added.

We have now added more detailed information on the protein expression and purification.

• How were the proteins stored?

Proteins were stored as frozen aliquots at -80 deg C. We have added additional information.

• There is information missing on the microscopy part of the material and methods. The readers will benefit from a more detailed description of these experiments. For example: the objective used is currently missing.

We have now added additional details regarding the microscopy in particular including objective used.

• Please add descriptions for the axes in 4B.

We thank the reviewer for pointing out this error and we have now added the axes description on 4B.

--

Referee #2:

The manuscript by Perdikari et al. explored how alkanediols regulates FUS liquid-liquid phase separation. As one of the widely used alkanediols, 1,6-hexanediol has been long used in dissolving nuclear pores, membraneless organelles, and disordered protein biomolecular condensates. But how it works remains poorly understood. In this work, the authors firstly evaluated the effect of 6 different diols on the phase separation of FUS full-length (FL), low complexity (LC) and RGG3 domain, respectively. Then, they applied NMR spectroscopy to study the interaction between diols and FUS. The results show that 1,6-hexanediol (1,6-HD) disrupts FUS phase separation mainly by forming protein-solvent hydrogen bonds and increasing FUS local motions, but not by disrupting the interaction with specific hydrophobic/aromatic residues. Altogether, this manuscript sheds light on the molecular mechanism underlying regulation of alkanediols in FUS phase separation. The work is interesting and is of potential importance to the field. However, to strengthen this work, the authors may need to analyze and interpret the data carefully, and conduct additional experiments according to my suggestion listed below.

Major concerns:

1. The authors examined the effects of 6 different alkanediols on phase separation of FUS full-length (FL), low complexity (LC) and RGG3 domain. They found that the phase separation of the FUS LC domain was most sensitive to the addition of different alkanediols, as manifested by the disruption of the droplets upon addition of diols. However, the droplets formed by FUS RGG3 and FL with RNA (polyadenylic acid, polyA) are strongly resistant to all types of alkanediols. Moreover, 1,2-cyclohexanediol and 1,2-CHD promote the phase separation of FUS RGG3 and FL with RNA. To fully support these claims, the authors need to present the DIC micrographs with a higher resolution and magnification, since the current DIC micrographs are too small to be clearly observed and compared.

We have now added additional microscopy data and we have presented the data with higher resolution and magnification. Though the DIC images are not meant to supply a quantitative picture we think the updated data and presentation improve the paper.

Moreover, the DIC micrographs do not match well with the quantitative turbidity data. For example, the data in Figure 1A showed that the inhibitory effect of addition of 5% 1,5-pentanediol (1,5-PD) was significantly stronger than that of 5% 1,4-butanediol (1,4-BD), but in Figure 1B, the inhibitory effects of the two compounds are opposite.

We agree with the reviewer that Figure 1 was confusing. As described above, Figure 1B was performed too close to the saturation concentration for the values to be a meaningful quantitative measure of the effect of each alkanediol on phase separation. Therefore we repeated the saturation concentration assay in a regime (at higher protein concentration and lower diol concentration) where phase separation is still present at all diol concentrations, allowing us to quantitate the effect on phase separation. Importantly, the data are much easier to interpret together now. We also fixed a labeling error we note (see response to Reviewer 1).

In addition, when describing the effect of the addition of alkanediol on the co-separation of FUS LC and RNA, the authors described that "phase separation is modestly decreased but still present even with addition of aliphatic alcohols", but I can't get the same conclusion from the bottom panel of Figure 2A (e.g., 5% 1,2-hexanediol (1,2-HD), 1,6-HD, and 1,5-PD). The authors need to address the inconsistency.

We note that the modest decrease is shown by turbidity in figure 2B (right) and we have clarified this in the text. We have also added the caveat that the main point is that phase separation is not disrupted markedly and that the imaging is not intended, designed, or set up to be quantitative.

2. The authors explored whether the different effects of alkanediols on FUS LC phase separation could be attributed to particular contacts between different alkanediols and FUS by performing NMR titrations. The authors conclude that "all alkanediols have similar mechanisms of action" and "do not find evidence for specific interactions caused by particular alkanediols or isomers". However, the NMR data in Figure 3B and SI Figure 2 indicate that the N-terminal residues (~ 6-12) show obvious larger chemical shift deviations than the other residues upon addition of 5% 1,6-HD. Similar results were observed for the titrations of FUS LC by addition of 5% 1,2-HD and 1,2-CHD.

In addition, two Tyrosine residues at the terminal of FUS LC, e.g., Y6 and Y161, revealed obvious larger chemical shift deviations upon addition of 5% 1,6-HD, as well as addition of 5% 1,2-CHD, 5% 1,2-HD and 5% 1,4-BD, but smaller changes with 5% 2,5-hexanediol (2,5-HD) and 5% 1,5-PD. The different slopes in Figure 1D were determined by the point with the largest change, which also suggested the critical role of chemical shift deviations of certain amino acid that were probably tyrosine. Thus, whether these residues play a role in phase separation of FUS-LC, and mediating the interaction with different diols? The authors might need to carefully analyze the titration results to address.

We have clarified the statements to make clear that all diols cause similar chemical shift perturbations in a way correlated with their ability to dissolve condensates. We agree that there are some position-specific effects and we clarify that the further detailed NMR data do not attribute these effects to favored interactions with particular residue-types.

As we stated above in response to Reviewer 1, we agree that some residues are more affected than others. This observation may be due to slight differences in conformation preferences in the structure of FUS LC at each site (see Burke et al Mol Cell 2015) that are differentially altered

(due to local conformations as well as due to intramolecular collapse and interactions in FUS LC (see Ryan et al Mol Cell 2018). We also agree that in FUS there are some differences between the diols, however the correlation plots (for which we have now added the Pearson correlation coefficients) show that the effect of the different diols are highly similar in pattern and differ mostly in magnitude. As the reviewer points out, the sites near Y6 and Y161 show the largest shifts and the correlation shows that they are similarly affected across different diols. Taken together, it is quite possible that some residues are more affected than others. We revised text to make that clear. There is a variance between residues in each residue type. However, the magnitude of the chemical shift can depend on many factors but the data do not suggest major differences between diols.

Finally, Lin et al JBC 2017 from Michael Rosen group has shown that specific FUS LC tyrosines do not appear to contribute more to phase separation than others, hence we find it unlikely that position-specific effects dominate the effects of alkanediols on phase separation of FUS LC.

3. The authors measured ¹HN resonances over a range of temperatures by using NMR to examine whether position-specific changes in hydrogen bonding in FUS LC occur in the presence of 1,6-hexanediol. The data showed that serine and tyrosine display the largest difference upon adding 1,6-hexanediol. From the view of experimental design, other diols, at least 2,5-HD should be included as a negative control. Additional diols with the same experiment are appreciated, which would be useful to understand the differences caused by different alkanediols, and certainly strength the conclusions.

We would expect these to be similar just smaller as the effects at a single temperature are so correlated by chemical shifts. Given the amount of additional effort needed for this and that the change in the temperature coefficients are already very small for 1,6 hexanediol, we decided not to perform these experiments and instead make clear the limitations of this analysis (see reviewer 1 reply above).

Minor concerns:

1. In SI Figure 1B, "12-HD ..." should be "1,2-HD ...".

We have fixed this issue, thank you for pointing it out.

2. The authors selected six types of alkanediols to study the mechanism of their effects on FUS phase separation, taking into account the length of the alkane chain (4-6 C), the conformation of the alkane chain (linear and cyclic), and the position of the hydroxyl group. It would be meaningful to analyze the effect of different types of alkanediols on phase separation from the perspective of alkanediol structure. Even if there is no more data to support it, additional discussion could be added.

We appreciate this suggestion and we have added additional analysis including a figure with the structure of the alkanediols and added discussion.

--

Referee #3:

The paper by the Fawzi lab addresses how different alcohols impact on protein condensation. This is an interesting topic as many alcoholic compound are used to modulate protein condensation. In their manuscript the authors combine light microscopy and OD600 measurements to determine the degree of phase separation (these methods only show limited correlation though, see below). Furthermore, the authors exploit NMR spectroscopy and try to detect the effect of 1,6 HD on the chemical shifts of FUS.

In general, the paper is very descriptive (experiments are performed and the results are presented, but not always interpreted in the light of a mechanism or model).

In addition, there appear to be no real new hard findings or conclusions: the alcohols sometimes enhance, sometimes decrease LLPS and there are no specific contacts between the compounds and FUS.

In that light, the paper fails to provide a clear message. (The intro and discussion are well written and general, but the findings seem not to add much to our understanding of how alcohols influence LLPS, despite the title being "molecular insights").

We appreciate the reviewer's comments and we have refocused the text on the main message – that popular simplistic conceptions of alkanediols interacting with hydrophobic groups directly as “minidetergents” and that distinct alkanediols isomers have qualitatively different effects are simply not true. The new finding is that alkanediols do not work by specific aromatic residue interaction and in fact seem to disrupt both hydrogen bonding and water interactions – neither of which are “pi” specific. We emphasize that these insights are consistent with the view many interactions contribute to disordered protein phase separation, not just pi interactions. We have also altered the title to more accurately reflect the findings.

In addition, I have serious doubts regarding the referencing of the NMR spectra and the extracted CSPs (see below). In that light, I think that the manuscript is not suitable for EMBO journal.

We have now addressed the chemical shift questions and referencing in detail (see below) to thoroughly address these issues to make the paper suitable for EMBO Journal.

In addition, I have a (larger) number of major and minor remarks.

Figure S1: the panels for 5% 2,5 HD are the same at 2.5 and 5 hours. This must be a mistake. We thank the reviewer for pointing out this issue. It is a mistake. We have fixed this panel.

"with the latter diol being the least efficient among the series of hexanediol isomers being tested". This seems to be the case in figure 1A, but not in figure S1A. How representative are the shown images? How many times were these experiments independently performed?

We agree with the reviewer that there was an issue regarding consistency and we have addressed it thoroughly and extensively by repeating both microscopy and saturation concentration measurements in a much more robust fashion (see comments for reviewer 1). We are confident these revised experiments now improve the manuscript consistency significantly.

" both 2.5% and 5% 2,5-HD resulted in lower concentration of protein remaining in the dispersed phase and higher turbidity". Are the results of Fig 1A and Fig 1B in full agreement? Seems to me that the 2,5 HD is indeed in agreement with panel A, but the 1,4 BD is for sure not in agreement (5% 1,4 BD behaves the same as e.g. 1,6 HD and 1,2 CHD in panel B. In panel A %5 1,4 DB behaves different as e.g. 1,6 HD and 1,2 CHD). Also, 2,5 HD and 1,5 PD are the same in B, but different in A. This raises the question as to how reproducible the experiments are, or as to how relevant the readouts (seeing bubbles, measuring some concentration, turbidity) are.

As above, we very much appreciate the reviewer's careful reading and have addressed these issues seriously and thoroughly (see reviewer 1).

Figure 1C: is that 5% of the additives?

We have added labeling to explicitly mark the concentration of the diols.

Fig S2: I would like to see some of the raw data. All CSPs are in one direction. This indicates a general shift of the resonances in a specific direction. Are the spectra referenced using e.g. DSS? General resonance shifts are of course expected if 5% (or 2.5%) of another solvent is added. The same happens when measuring spectra at different temperatures or different pressures. Those general CSPs are, however, not interesting as they don't report on any interactions, but merely in changes in the general solvent properties. Along the same lines, the authors should perform control experiments to differentiate between CSPs caused by "specific" interactions and by changes in the solvent properties. One way of doing this would be to use a folded protein. In case residues not on the surface behave, residues in the core and the FUS IDP resonances respond the same to the solvents there is clearly sign that the CSPs are not directly caused by interactions between the alcohols and the proteins.

The reviewer points out that the shifts are primarily in one direction. However, it is clear that the shifts are not uniform at every position – therefore the shifts are not simply due to referencing changes. It is an important question to consider the referencing. For the 1HN and 15N shifts, this is because the shifts arise from changes in interactions with water. The shifts are considerably smaller for 1H positions attached to carbon atoms, supporting our view. Also, that the backbone 15N shifts are correlated with the 1HN shifts and are much larger than the 1HN shifts again is consistent with our conclusions. Furthermore, the very small changes in most 13C positions (and 1H carbon attached positions) suggests that uniform solvent issues are not a problem. Finally, we have performed 1D 1H experiments with DSS to demonstrate that the shifts have at most a 0.006 ppm shift in 1H, much smaller than the observed shifts which go up to 0.020 ppm. However, given that as the reviewer points out, a solvent can affect the shift including the shift of the DSS reference standard.

We also attempted to assess the impact on folded proteins using the RRM of hnRNPA2 as a model folded protein. However we found that even residues in the core shifted, likely due to slight changes in the structure transmitted from the surface – proteins are not static rigid assemblies – and the specific arrangement is perturbed by high percentages of co-solvents.

Fig S2: error bars are definitely needed in all panels (A-G).

Error bars are smaller than visible and therefore not included.

"In other words, the sequence-specific effects are largely the same for all diols tested." . I don't understand how anything about sequence specificity could be extracted from the plots in Fig. 1D.

We have rephrased this section (see reviewer 1).

Fig 1E: I assume that this is at 5% additive. The correlation at 2.5% should also be shown as there the spread in the concentrations (Fig 1B) is much larger. Please add error bars in the plot. Looking at Fig 1B, the error in the concentrations is large compared to the differences between the additives.

We appreciate this suggestion and we have added the correlations and we have also extensively addressed comments from the other reviewers to improve this section.

"Hence, the changes in residue-by-residue perturbation of the chemical environment are strongly correlated with the mechanism of action. " This statement is far to strong I think (see above). There is no insights into any kind of mechanism of action here.

We have now rephrased to say that the changes are “strongly correlated with the effect on phase separation” – removing the word “mechanism”.

"predicted water solubility (logP) of each diol" Why should there be a correlation? 1,4-Butanediol can be mixed with water at any ratio and 1,6-Hexanediol is soluble up to 50% (w/v). According to the legend P is the octanol partition coefficient (how much of the compound goes into octanol and how much into water) I don't see how this tells something about water solubility of the different compounds. P only tells something about the difference in solubility of the compounds in octanol and in water.

We agree this was not well presented. Log P using octanol partitioning coefficient is a highly used and standard measure of hydrophobicity. We removed a confusing mention of "solubility" and replaced the word with hydrophobicity and added additional text to clarify. Despite claims that in the literature that the effectiveness depends on hydrophobicity, we did not find a clear correlation (especially for diols with 5 or more carbons) with this (predicted) measure of hydrophobicity, but we do see clear correlation between effect on phase separation and NMR chemical shift perturbations. Unfortunately, we could not find tabulated data of logP experimentally determined for all hexanediols.

In addition to my confusing regarding the relevance of the partition coefficient I disagree with the statement "similar predicted partition coefficient such as 1,2-hexanediol (which also has significantly lower experimentally-determined solubility than 1,6 hexanediol) and 2,5-hexanediol show different impacts on LLPS. " Why are the log(P)s of 1,2 HD (0.65) and 2,5 HD (0.4) very similar, but is the log(P) of 1,6 HD (0.3) significantly different? A difference of 0.25 can not be called similar, while a difference of 0.1 is called significant.

We have more carefully explained that the log(P) is presented only to show that the correlation is not good and removed these statements that are confusing.

"Importantly, we find that phase separation is modestly decreased but still present even with addition of aliphatic alcohols " To me it looks like there are more "bubbles" after adding any of the alcohols in the presence of RNA (I don't see a decrease in Fig. 2A). Fig. 2A (bottom row) and Fig 2B (right panel) are clearly inconsistent to me. (maybe the turbidity in Fig 2B decreases over time in the presence of alcohols as there is larger condensates that just sink to the bottom). Like this it makes no sense to me.

As described above, we have repeated the experiments on FUS LC and for the RGG we emphasize in the text that the imaging is not provided for quantitative purposes and finally we have adjusted the text to reflect that we are referring to changes in turbidity only. The reviewer is correct that droplets settle over time to make the solution less turbid – and in fact this speed of decrease of turbidity is often indicative of liquid phase separation as opposed to solid-like aggregation. We clarify in the text this fact.

For all panels in Fig 2 a control with only RNA should be done, maybe the RNA phase separates too, independent of the protein.

We do not see evidence of phase separation of the RNA at these conditions in these concentrations, though it is certainly possible. All samples of RNA are buffer exchanged via a spin desalting column in the experiment buffer before use, so any possible micron sized assemblies would be removed.

Again, I see no correlation between Fig 2C and Fig 2D. If at all, I see less bubbles for 5% 1,2 CHD than for no alcohols in Fig 3C, whereas Fig 3D indicates more condensation for 5% 1,2 CHD than for no alcohols. The lack of correlation between complementary experiments is disturbing.

We agree with the reviewer and have decided to remove Figure 2D as the quantification is not reliable to interpret due to the absorbance of the RNA (leading to the large uncertainties).

"Imaging the RGG3- RNA condensates over time showed that droplets remained spherical " Do the droplets remain liquid like, or are they forming hydrogels that can no longer dissolve. In case the droplets matured, it is clear that alcohols have no effect and that they remain the same in shape over time.

Although a detailed analysis of the droplet dynamics could be interesting, as we described above any changes in the phase diagram will also change the viscosity. This work focuses on the impact on the phase diagram and we think changes to the viscosity could be investigated in follow up work.

Fig 3A and 3B. These spectra should be references properly. 95% of the shifts go in one direction. This will mask residue specific differences (in case there are any). The CSPs that are extracted here are not relevant I would say.

Fig 3C, D. See my remark above. In the absence of proper referencing these data appear not very valuable to me.

We refer the reviewer to our comments on chemical shift referencing above. The fact that the shifts are not identical at each position shows that the shifts are not from referencing issues. Also, the CSPs for ^{15}N and ^{13}C are much larger than the referencing corrections.

Fig S4 C and D. The plot contain 37% bars for residues that are only present once in the sequence. As the spreads are very large (as can be seen for the residue types that occur often), this is not very informative.

We agree that the spreads are large and we have now emphasized in the text that the variability is one of the findings we wish to highlight.

"Hence, the elevated R2 values likely report on dynamic interactions leading to phase separation...". One easy (other) explanation is that the viscosity of the solvent changed by the addition of the alcohol. This then leads to changes in the relaxation rates, fully independent of phase separation effects. One could check this by e.g. recording data on an IDR that does not undergo LLPS.

We agree with this suggestion and we have now performed the same NMR spin relaxation experiments on a highly similar sequence that does not phase separate – the phosphomimetic FUS LC 12E that has 12 serine/threonine to glutamate substitutions (see Monahan et al EMBO J 2017). Here we find that R2 is lower to start with (Monahan et al 2017) and importantly does not change as much as observed for the wild-type unmodified FUS LC.

"the ^1H ^{15}N NMR data do not conclusively show if 1,6-hexanediol makes specific contacts that mediate its action or if it primarily alters the solvent environment." I would say that the spectra show that there as no specific contacts. The wording of this sentence is misleading ("do not conclusively show" → "don't show").

We have now updated this section as suggested.

A technical question regarding the buffers. Do the authors take the 0% additive buffer and then pipette in the additives directly? This would lower the salt concentration of course. Or do the authors prepare a buffers such that the buffers with additives have exactly the same same concentration as the buffers without additives?

The additives are dissolved in MES buffer with 150 mM NaCl so the salt and buffer concentrations are not decreased.

I am confused with the errorbars in Fig. 4D. For Ala, there are 2 datapoints (around 0.002 and around 0.05 or so). The average of those makes sense (around 0.026), but the displays error bar (s.d. I assume, it is not mentioned), makes no sense. How is this calculated?

We have addressed this issue and added information that we are showing the standard deviation except for places with two data points where we simply show the range.

Again for Fig 4, the real differences in chemical shifts might be obscured in a general shift of the spectra due to changes in the solvent.

We have addressed this extensively above.

Clearly, I agree, there is no correlation between the amide nitrogen and carbonyl carbon CSPs. But why should there be one? Has this ever been systematically investigated. Of course when there is a single well defined binding site the ^{15}N and ^{13}C CSPs are localized and then correlate, but for the very small shifts here I see no reason for this. Please explain.

If the residue that is perturbed by direct binding is showing something at the $^1\text{H}/^{15}\text{N}$ position, it is reasonable to assume that it would show up at ^{13}C backbone position. Therefore the lack of correlation again suggests that the chemical shifts are changed due to changes of hydrogen bonding and water, not from side chain (hydrophobic) contacts. We have clarified this in the text.

"show a wide range of chemical shifts " I assume the authors mean "show a wide range of chemical shift perturbations".

We agree and we have fixed this issue in the text.

It took me a while to figure out what the labels in panels 4E and F (x-axis) are. Please make this clearer.

We have fixed this issue in the figure.

I strongly disagree with calling Serine, Threonine or Glutamine aliphatic amino acids. Those are polar, uncharged amino acids. Aliphatic amino acids are e.g. Leucine, Valine and Isoleucine (that is a biochemistry I class)

We did not intend to call them aliphatic amino acids, but they have aliphatic carbon chain segments. We have clarified this in the text.

"Surprisingly, many of the backbone $^{13}\text{C}\text{O}$ chemical shifts are larger than the ^{13}C aromatic (i.e. tyrosine) and aliphatic sidechain shift perturbations". What is the conclusion of the authors regarding this statement?

The main conclusion is that the NMR provide evidence that the effect of hexanediol is not via aromatic side chain interactions as surmised by others previously.

Fig 5B definitely needs error bars (that are likely very large).

We have added these error bars. The error bars are large compared to some of the values, but many are much smaller than the values shown.

Fig 5: some primary data should be shown.

We have added spectra and example plots in the supplementary data.

Fig 5. Did the authors take into account that different amino acids behave differently with temperature and have different pKa values? The analysis here seems quite limited.

It is not clear what is meant here by the reviewer regarding different behavior and pKa values. There are no histidine residues the only residues with pKa near the experimental pH. The only

charged residues are two Asp residues, which also are far from the pKa.

Panel 5C is not of much value I would say.

We agree and we have moved this panel to Appendix. We also added primary data as suggested by this reviewer above.

"Still, our results show that NMR-based observations can provide unique insight into the molecular origins of phase separation modulation and could contribute to the rational design of possible therapies for altering disordered protein phase separation." What molecular origins of phase separation have been found in this paper?

We rule out tyrosine/aromatic-specific interaction of hexanediol and it is consistent with our previous findings that many interactions modes and residue types contribute phase separation of these IDRs.

Dear Nick,

Thank you for sending me the point-by-point response for how to address the remaining concerns. I appreciate the proposed experiments and would like to invite you to submit a revised version.

Let me know if we need to discuss anything further

with best wishes

Karin

Karin Dumstrei, PhD
Senior Editor
The EMBO Journal

Use the link below to submit your revision:

Referee #1:

I applaud the authors for revising their manuscript, which I find now improved on many points. I also appreciate the explanations that they have offered for reviewer recommendations that they did not follow, either because the data asked were previously shown or because some of the quantifications were too tedious for presumably little added value on their conclusions.

One point that I do not fully agree with the authors on, and which I think should still be revised before publication is the following:

Reviewer point: For 2.5% and 5% 2,5-HD there seems to be a difference in the effectiveness of droplet dissolution. Several concentrations of alkanediol isomer have been used in the field over the years (usually ranging from 1-10% w/v alkanediol concentration). It would, therefore, be particularly useful if the authors performed a dose response ranging from 1 to 10% for all alkanediol isomers used in this manuscript.

Author response: We have now added an evaluation of the saturation concentration as a function of 1,6 hexanediol concentration from 0.5% to 5% (Figure 1B). We did not go above 5% as for FUS LC we cannot generate easily a higher concentration of FUS LC - i.e. the saturation concentration becomes extraordinarily high at about 5% hexanediol.

Reviewer response: I think the new panel Figure 1B is very useful. I may be missing a point here, but in my mind, altering the hexanediol concentration is more important than changing the protein concentration here.

Author response (continued): We show that for 0.5% to 2.5% the response of the saturation concentration is linear. We also show that the correlation between 1% and 2% hexanediol is linear for all diols tested. Therefore we did not provide additional evaluation for all diols at all concentrations as the amount of effort and sample and replicates needed became too great and we demonstrated effect linearity already several ways.

Reviewer response (continued): Here I respectfully disagree with the authors. Just because a chemical treatment is linear in one concentration range, doesn't mean it will be linear in all the ranges. In fact, it is usually not the case. I think this is an important point, especially given that most cellular (and sometimes also in vitro studies) use higher hexanediol concentrations than tested here. I think testing the behavior of these alkanediols at higher concentrations and their effect on protein phase separation is particularly interesting.

Referee #2:

We appreciate the authors' efforts in revising the manuscript, including incorporating additional data, updating figures, and revising the wording. However, several critical issues remain unresolved, which significantly diminish the likelihood of the paper being published in EMBO J. We summarize these shortcomings as follows:

a. Insufficient control experiments: the manuscript lacks essential control experiments, limiting the evidence to support the authors' conclusions. While the authors have made some attempts to address this concern, we still strongly recommend the author to include 2,5-HD as a negative control in the NMR experiments. This addition would provide stronger supports for the claim that 1,6-HD impairs FUS LC LLPS as described. By comparing the effects of 1,6-HD and 2,5-HD, it would help to exclude the possibility that the observed effects in Figures 3, 4, and 5 are merely noise rather than genuine disruptions of FUS LC LLPS by alkanediols.

b. Inappropriate quantitative and measurement methods: the use of inappropriate quantitative and measurement methods probably results in weak or false conclusions.

1. Regarding the quantitative analysis of the NMR data in Figure 1 and Figure S1, we appreciate the authors' presentation of the effects of different diols on FUS LC LLPS and its NMR signals. However, we have concerns about the approach used, as the authors solely rely on the chemical shift perturbation (CSP) of ^{15}N while disregarding the CSP of ^1H , despite using ^1H -related measurements in Figure 5. This inconsistency raises questions about potential bias and the possibility of distorted conclusions. Furthermore, it is evident that the observed ^1H CSP induced by the diols does not correspond to the extent of their LLPS inhibition. For instance, the ^1H CSP caused by 2.5%/5% 1,4-BD and 2,5-HD is considerably larger than that induced by 1,6-HD. This inconsistency undermines the credibility of the conclusions presented in Figure 1.

2. The quality of the data in Figure 2 does not seem to have significantly improved. The authors state that phase separation is modestly decreased but still present even with the addition of aliphatic alcohols, as indicated by high turbidity values that decrease over time due to droplet settling. In Figure 2B, OD600nm measurements are used. However, it is important to note that the treatment with different diols results in a significant difference in the size and size distribution of FUS-RNA droplets. To accurately quantify the extent of FUS-RNA LLPS in Figure 2, a more precise method similar to that used in Figure 1B and 1D should be employed. These figures demonstrate the proportion of the supernatant to the total protein content, providing a more accurate measure. Considering the complications caused by the presence of RNA, we recommend employing a high-accuracy and similar quantitative method, such as SDS-PAGE grayscale quantitative analysis for protein concentration in the supernatant, in Figure 2B. If the construction of MBP-FUS poses challenges due to the overlap of MBP and FUS in the gel, we suggest using purified FUS-EGFP as an alternative (PMID: 29677514).

c. Certain conclusions exhibit contradictions and suffer from a lack of persuasive argumentation process and logical reasoning.

1. The conclusion that alkanediols alter the solvation of FUS LC and may disrupt hydrophobic interactions and hydrogen bonding lacks detailed descriptions and contains confusing points.

The authors attempt to emphasize that popular simplistic conceptions of alkanediols interacting directly with hydrophobic groups

as "minidetergents" and the notion that distinct alkanediol isomers have qualitatively different effects are not true. However, they also mention that alkanediols disrupt hydrophobic interactions in this section, as well as in the abstract and discussion. It is unclear how alkanediols can disrupt hydrophobic interactions without directly interacting with hydrophobic groups.

Furthermore, the statement in the abstract that "the hydrophobic interactions mediating SYGQ LC domain LLPS are perturbed differently by each diol" lacks experimental data supporting this conclusion. This conclusion may be related to the predicted hydrophobicity data in Figure S1, however, the presented data in Figure S1 does not provide compelling evidence to support this conclusion, as the predicted results lack substantial substantiation.

2. The authors state that interactions between 1,6-hexanediol (1,6-HD) and FUS LC are relatively weak and non-specific. However, it is important to note that the observation of significant CSP in 13CO under 1,6-HD indicates some level of interaction between 1,6-HD and FUS LC. This raises a valid question about the authors' assertion of weak interactions between 1,6-HD and FUS LC.

3. The authors state that they studied the capacity of condensate-modifying agents to inhibit phase separation of FUS in the presence of RNA, aiming to model protein-RNA interactions that contribute to biomolecular condensates formed in cells. However, it is worth noting that 1,6-HD has been shown to efficiently dissolve many RNA-containing granules, such as stress granules, where RNA-protein interactions are crucial for their assembly (PMID: 32302571). Furthermore, studies have reported the ability of 3% 1,6-HD to melt FUS granules in cells (PMID: 34746121). These findings are in contrast to the observation in Figure 2A, where FUS-RNA droplets were not melted by 1,6-HD. This discrepancy suggests that the phenomenon observed in the experimental setup lacks representative significance. It is possible that the use of an artificial system with a single RNA species and excessive RNA may contribute to these observations.

4. The observation that the Q to G mutant of FUS does not impair FUS phase separation (PMID: 29961577) is an important point to consider. It suggests that the sidechain of Q may not play a significant role in FUS phase separation. This raises doubts about whether the hydrogen bonding observed in the sidechain CO of Q under 1,6-HD, as shown in Figure 3, reflects the mechanism by which 1,6-HD disrupts FUS phase separation. It is necessary for the authors to address the discrepancy and provide a clear explanation. They should discuss how the observed hydrogen bonding in the sidechain CO of Q under 1,6-HD aligns with the understanding that the Q to G mutant does not impair FUS phase separation.

Referee #3:

Like the previous version, this version of the manuscript is still highly descriptive (spectra and microscopy images are shown and it is described what happens). However, the title suggest "Molecular insights", that I did not find in the manuscript. The abstract and conclusions claim that hexanediol induces changes in the protein solvation, but this is only suggested indirectly by the data. In general the data is not analyzed very critically and often the conclusion cannot be unquestionably drawn. I thus cannot support publication of this manuscript, despite the high interest of the topic.

Major remarks:

I see no correlation between Figures 1C and D. This point was raised by all reviewers, and despite the response that the microscopy images are only qualitatively, the feeling remains that the two approaches (microscopy, saturation concentration) are reporting on different things.

In Figure 1E it should be clearly stated that one dot is one residue (or more accurately one nitrogen spin).

"In other words, the sequence-specific effects are highly similar for all diols tested, just with a different magnitude, suggesting that all alkanediols affect phase separation by a similar mechanism." should read something like: "In other words, the sequence-specific effects are highly similar for all diols tested, just with a different magnitude, suggesting that all alkanediols interact with FUS in a similar manner, albeit with different affinities" (This then might imply that they affect phase separation similarly).

Figure 1F and S1F. It is rather arbitrary to call correlations Strong or Moderate based on the PCC. I would use the following criteria: $r > 0.8$ is a strong correlation, $0.5 < r < 0.8$ is a moderate correlation and $r < 0.5$ is a weak correlation. In that context all correlations are moderate. Bottom line: there is no reason to suggest that the correlations in figure 1F are strong and that the one in Figure S1F is only moderate. They are all in the same range, with the correlation in Figure S1E being a bit weaker. The discrimination between strong and moderate is even less relevant if one considers that the $\log(P)$ values are only predictions (thus likely making any correlation weaker).

"Importantly, we find that phase separation is modestly decreased but still present even with addition of aliphatic alcohols (Fig 2A)" I do not see in Fig 2A (row + RNA) how the phase separation is changed by the additives. To me all panels look equal with respect to the degree of droplet formation. The 1,2-HD should be removed from the panels in Figure 2A and B for the FL protein as there is still MBP attached (I assume), preventing a comparison. Also, the panels in the top row are missing, which prevents me from judging the data.

For all replicates in all experiments, please clearly indicate if these are technical (doing the same measurements with the same

batch of protein) or biological replicates (with differently purified batches of protein). As FUS seems hard to purify there could be larger variations between different batches.

" and phase separation is not markedly decreased, neither immediately (Fig 2D) nor over time (Appendix Fig S4)." The turbidity does change over time (Fig. 2D), I assume that the authors mean that the changes in turbidity over time are independent of the presence of the alkanediols.

Figure 3A and B. Could the authors analyze the CSPs by combining the ^{15}N and ^1H shifts, this would likely give more accurate data and also a more common way of analyzing interactions.

"though the distribution of shifts for each residue type is large, especially for the most common residue types (serine, glycine, glutamine, and tyrosine)." Why would the CSPs vary more for the more abundant residues? Is it not just because there are more data-points that there appears to be a larger spread? For the small sample size for many residue-types the spread is not a relevant measure for anything.

The authors write that Ala and Thr show the largest CSPs. This is only the case in the nitrogen dimension. However, what the mechanistic basis for this could be. Do those residues interact preferably with 1,6-HD or is this just coincidence, or are those large CSPs just "outliers" and not representative of any general amino-acid specific trend?

" though a similarly large distribution of shifts for glycine, suggesting that 1,6-hexanediol similarly influences the chemical environment of highly distinct sequences" I strongly disagree with this conclusion, unless the authors have an explanation why a similar (random) distribution of the size of CSPs would correlate with similarities in interactions. I would actually conclude from the data that all (glycine) residues interact significantly differently with the 1,6-HD, maybe in a sequence specific manner.

"Supporting this hypothesis, 1,6-HD had little effect on the molecular motions of FUS LC 12E variant, presumably because the introduction of 12 phosphomimetic mutations has already disrupted the molecular contacts upon which 1,6-HD acts (Appendix Fig S6)". I am not sure if the differences in the changes in R1 and R2 (Fig 3E and Fig S6) are actually as large as the authors suggest. The y-scale of the R2 plot is different and to me the change in R2 is more or less between 0.5 and 1 Hz in both cases. One could just plot the change in R2 in both cases to see if the differences are really significant.

Figures 4E and F are obsolete as the CSPs are very small and cannot be determined with any kind of accuracy as they are a superposition of many different resonances from many residues. Also, why are some bars missing (AHB, QHA, SHA, THB but also ACB, TCB)?

"These shifts are modest and not markedly correlated with the backbone ^{15}N perturbations". They are uncorrelated.

"...and similar for side chain and main chain (i.e. α carbon positions)." . The plot (that is obsolete, see above) does show a variation by an order of magnitude, so the side and main-chain CSPs that are presented here are definitely not similar.

"Classifying the CO chemical shift difference by residue type (Fig 4D) shows that polar residues such as serines, threonines, glutamines but also glycines show a wide range of chemical shift perturbations." As for the nitrogen and proton CSPs, the largest spread is observed for the most common residues, which is just as one would expect statistically (as many residue types are only present with very few residues). Because of the limited number of probes for some residue types the spread (standard deviation) is highly uncertain and not a valid measure.

" Furthermore, given that the largest differences are observed for backbone and sidechain positions that form hydrogen bonds, " → I cannot see what this is based on, the authors contradict themselves. The glutamine side-chains shift the least in ^{15}N (compared to the bb) and the most in ^{13}CO (compared to the bb). Both the NH_2 and the CO in the side-chain can hydrogen bond. Also, there are many examples for bb CO, N and H resonances that do not shift, or only very little. Those are very well able to be involved in hydrogen bonds. So, I see no link between CSPs and hydrogen bonding abilities and I disagree with the conclusion that "alkanediols ... disrupt hydrophobic interactions and hydrogen bonding."

Fig 5A and B. The errors (standard deviations) in the slopes are so large that it is hard to see if the change in the slope upon addition of 1,6-HD is significant at all. The addition of the 1,6-HD will cause changes in many things, including viscosity. Maybe the observed (very small) changes in the slopes are just due to changes in solvent properties and completely unrelated to hydrogen bonding effects. I do not trust the conclusion unless proper controls have been made, e.g. with isolated amino acids in the presence and absence of 1,6-HD.

" Taken together, these data suggest that condensate-modifying molecules like hexanediol might enhance the solvation of the protein backbone leading to dissolution of the condensates." This is really far fetched and highly speculative.

Minor remarks:

"the observed chemical shift perturbations are small compared to what would be expected for a tight-binding molecule" → the extend of CSPs is not correlated with the affinity.

"Surprisingly, many of the backbone ¹³C chemical shifts are larger than " → ...¹³C chemical shift changes ...

"This large shift for glutamine side chain is unique given that the ¹⁵N side chain of glutamine is shifted less (~0.04 ppm) (Fig 3F) than the largest backbone ¹⁵N NH₂ (~0.20 ppm). " → I assume "...than the largest backbone ¹⁵N N-H (~0.20 ppm) "

Figure S3, please indicate that the weak band at 25 kDa is the TEV protease (in case that is true).

Supplementary figures come in the wrong order, please correct. Also the main text figures come in different order in the text and in the panels.

Figure S8A: why are there 2 spectra at the second spectrum (a blue and a cyan) on top of another.

Figure S8B: why are the amide signals doublets. Did the authors fail to decouple properly?

Figure S8D: "The high R² values for every position (>0.97 in all cases) suggest that the values extracted represent the true temperature coefficients." The high correlation coefficients only reveal that the resonance frequencies shift linear with temperature.

Remarks regarding the rebuttal letter:

"We also attempted to assess the impact on folded proteins using the RRM of hnRNPA2 as a model folded protein. However we found that even residues in the core shifted, likely due to slight changes in the structure transmitted from the surface - proteins are not static rigid assemblies - and the specific arrangement is perturbed by high percentages of co-solvents." → can this not be seen as an argument that supports a situation where hexandiol has a general influence on all resonances, independent of hydrogen bonding? In other words, does that not imply that the shifts are mainly due to changes in solvent properties and that CSPs are independent of changes in hydrogen bonding.

"We do not see evidence of phase separation of the RNA at these conditions in these concentrations, though it is certainly possible." The authors should show these data in the supplement.

"We agree that the spreads are large and we have now emphasized in the text that the variability is one of the findings we wish to highlight." The variability is not a relevant measure in my eyes (see major remarks). The large variability is correlated with residue abundance. And a large variability in the CSPs just means that the individual residues of a certain residue type react slightly different with the additives.

"If the residue that is perturbed by direct binding is showing something at the ¹H/¹⁵N position, it is reasonable to assume that it would show up at ¹³C backbone position. Therefore the lack of correlation again suggests that the chemical shifts are changed due to changes of hydrogen bonding and water, not from side chain (hydrophobic) contacts. We have clarified this in the text." In case the hydrogen bonding of water with the backbone changes the ¹H-¹⁵N shifts will change. That is clear. But, as the authors state (reasonably assume, I agree), this should then also be observed for the ¹³C backbone. Why is that not the case here then. Hydrogen bonding changes should be picked up by ¹³C, ¹H and ¹⁵N in the backbone.

"Fig 5. Did the authors take into account that different amino acids behave differently with temperature and have different pK_a values? The analysis here seems quite limited.

It is not clear what is meant here by the reviewer regarding different behavior and pK_a values. There are no histidine residues the only residues with pK_a near the experimental pH. The only charged residues are two Asp residues, which also are far from the pK_a." → The question is quite simple: do the amide shifts from different aminoacids react on temperature changes the same, or are there any indication that the temperature dependence of the chemical shifts are residue-type specific? In that case the analysis that the authors performed is an over simplification.

"We rule out tyrosine/aromatic-specific interaction of hexanediol and it is consistent with our previous findings that many interactions modes and residue types contribute phase separation of these IDRs." → I agree with that. That something does not play a role is an important finding, but molecular insights (title) should maybe reveal the mechanistic basis of which (specific) interaction do play a key role. This latter point is unfortunately too weak.

Editorial comments:

Thank you for submitting your revised version to The EMBO Journal. Your study has now been re-reviewed by the original referees and I am afraid that they find that many of the key concerns have not been adequately addressed. Given this, I am afraid that I can't offer to consider publication of the present submission.

I know the importance of the topic and I would like to give you a chance to see if you can add experimental data to address the remaining concerns. Please note that for me to consider another revision I need that the issues raised are addressed with the inclusion of experimental data.

We thank you for the opportunity to respond to the reviewers. We have taken an extended amount of time to respond because we appreciate the importance of the reviewers' concerns. We sought to thoroughly address them with many new experiments to revise, support, and refine the mechanistic claims regarding these compounds and account for the extent of their disruption of phase separation.

In this revised version, we shift the focus to experimentally testing and providing detailed insight into the nature of hexanediol effects on hydrophobic interactions. We have added many new experimental and simulation results, both those requested by reviewers and many others to make the claims clear. Specifically, we have added direct insights through 1) the first experimental measurements of the hydrophobicity of the diols – an important contribution to the field on its own – to enable us to test the correlation between hydrophobicity and disruption of phase separation. 2) Testing and comparing the effects of single alcohols (ethanol and propanol), for which previous data exist on hydrophobic impacts on folding stability, to hexanediols. 3) Investigating the impact of hexanediol and other alcohols on folded protein stability of the model protein lysozyme. 4) Conducting atomistic simulations to probe the contacts between hexanediol and the protein. We have also extensively revised the message presented in the text to reflect these additions as well as clearly present the findings and conclusions.

To directly address the reviewers' concerns, we have also performed the specific experiments requested, including importantly extensive quantification of droplet imaging which we now show is effectively perfectly correlated with our other biochemical measure of disruption of phase separation. We have included additional data on the effect of several different RNAs and different RNA concentrations on FUS phase separation, as well as new experiments investigating hexanediol's effect on polyD+polyK phase separation, which provides a good comparison as it is well established to be mediated by charge-charge interactions. Just as important, we have extensively revised the text and the claims carefully and thoughtfully, based on the reviewer comments, to make sure the claims are well supported. We have removed the discussion of specific hydrogen bonding and the temperature coefficient NMR that the reviewers suggested (and we now agree) is not clear.

In our view, these efforts have significantly improved the mechanistic insight of these findings and paint a consistent and important picture for the field. We are convinced that these findings presented in this thoroughly updated version are of substantial interest to the field, are of excellent quality with extensive support, and address the reviewers' concerns comprehensively.

Editor comments:

The key points are (but not limited to)

1. DIC images in Figure 1 and 2:

KD: We would need some form of quantification

We appreciate the need to directly address the reviewer concerns. One challenge we had encountered while attempting to quantify DIC images is that the alkanediol additives can cause FUS droplets to change how FUS droplets wet the glass slide (depending on the surface treatment, of which we tried several), resulting in irregularly shaped droplets on these glass surfaces. Combined with the non-uniform contrast of DIC imaging (which results in pleasing images but gradients of contrast), this made quantification difficult by canned outlining/picking routines (in Fiji/ImageJ) as well as home written Matlab scripts. To solve this issue, we have added image quantification by fluorescence microscopy, which provided enhanced and uniform contrast. Importantly, we also showed that quantification by fluorescence microscopy matches biochemical results from our previous approach measuring the concentration of protein remaining in the supernatant after centrifugation of droplets (Figure 1F), demonstrating robust findings and jointly validating both approaches.

2. Non-specificity of diols on dissolution of FUS biomolecular condensates:

KD: You provide different possible explanations as to why these differences could occur, independently of the dissolution mechanism of diols proposed. We would need some experiments to test at least some of their proposed hypotheses.

We appreciate the reviewer concerns that the main message of the manuscript was not clear nor was it fully supported. To this end, we embarked on a large series of additional experiments that tackled these important questions, where we focus on the common and logical, but yet-to-be-tested, supposition in the field that hexanediols disrupt phase separation via hydrophobic contacts. Specifically, a main thread in the research on hexanediol is that the hydrophobicity is thought to be important and that 1,6-HD is thought to be more hydrophobic than 2,5-HD, but neither of these hypotheses was ever thoroughly tested. To fill this hole in the field, we have added direct measurements of the hydrophobicity of alkanediol molecules. We also used orthogonal approaches (i.e. folding stability, not just phase separation) to experimentally probe alkanediols' impact on other processes that are also associated with hydrophobic interactions (Figure 4). Additionally, we added extensive, validated molecular dynamics simulation to probe the molecular features of hexanediol solutions and the direct contacts between protein and alkanediol molecules, as well as how this differs for 1,6-HD vs 2,5-HD. These new observations paint a consistent picture of the importance of hydrophobicity that we examine in detail, combined with a critical examination of how hydrophobic and geometric differences between the isomers govern their effect on phase separation.

3. FUS-RNA biomolecular condensates are resistant to diol treatments:

KD: We would need further experiments with different RNAs and concentrations to support this.

We appreciate this concern and have made significant additions with different RNAs and varying RNA concentrations. Specifically, we have improved the quantification both by imaging and biochemistry methods, and have added new data using disordered polyU RNA as well as torula yeast total RNA extract (and find similar effects) to support our claims. We also quantified the effect of 1,6-HD on RGG3+RNA mixtures at varying RNA concentrations and found no significant differences across RNA-to-protein ratios ranging from 0 to 1. Furthermore, our new experiments investigating hexanediol's (lack of) effect on polyD+polyK phase separation further support our claim regarding the resistance to 1,6-HD of some types of phase separating peptides that depend on charged interactions. Finally, we carefully discuss how these findings regarding disordered protein interactions with RNA observed in a test tube relate to RNA containing granules dissolved by 1,6-HD in cells.

Referee #1:

I applaud the authors for revising their manuscript, which I find now improved on many points. I also appreciate the explanations that they have offered for reviewer recommendations that they did not follow, either because the data asked were previously shown or because some of the quantifications were too tedious for presumably little added value on their conclusions.

One point that I do not fully agree with the authors on, and which I think should still be revised before publication is the following:

Reviewer point: For 2,5% and 5% 2,5-HD there seems to be a difference in the effectiveness of droplet dissolution. Several concentrations of alkanediol isomer have been used in the field over the years (usually ranging from 1-10% w/v alkanediol concentration). It would, therefore, be particularly useful if the authors performed a dose response ranging from 1 to 10% for all alkalediol isomers used in this manuscript.

Author response: (note, these outdated responses are from our response to the previous set of reviewer comments) "We have now added an evaluation of the saturation concentration as a function of 1,6 hexanediol concentration from 0.5% to 5% (Figure 1B). We did not go above 5% as for FUS LC we cannot generate easily a higher concentration of FUS LC - i.e. the saturation concentration becomes extraordinarily high at about 5% hexanediol."

Reviewer response: I think the new panel Figure 1B is very useful. I may be missing a point here, but in my mind, altering the hexanediol concentration is more important than changing the protein concentration here.

We appreciate the reviewer's question and we would like to explain our reply. We had commented that "we cannot generate easily a higher concentration of FUS LC - i.e. the saturation concentration becomes extraordinarily high at about 5% hexanediol" because there is already no phase separation for FUS LC in 5% 1,6 HD at a very high concentration of 300 μ M FUS LC, higher than which is not easy to achieve in these assays requiring mixing several solution components. This can be seen in our new Figure 1B, where we quantified the droplet area fraction in fluorescence microscopy. The effect of 1,6-HD on FUS LC linearly increases with 1,6-HD concentration until the area of droplets drops to zero at 5% 1,6-HD.

In this latest version of the manuscript, we have now performed the requested experiment (see below) with 10% 1,6-HD using a longer FUS construct that phase separates with a lower saturation concentration.

(note, these outdated responses are from our response to the previous set of reviewer comments) Author response (continued): "We show that for 0.5% to 2.5% the response of the saturation concentration is linear. We also show that the correlation between 1% and 2% hexanediol is linear for all diols tested. Therefore we did not provide additional evaluation for all diols at all concentrations as the amount of effort and sample and replicates needed became too great and we demonstrated effect linearity already several ways."

Reviewer response (continued): Here I respectfully disagree with the authors. Just because a chemical treatment is linear in one concentration range, doesn't mean it will be linear in all the ranges. In fact, it is usually not the case. I think this is an important point, especially given that most cellular (and sometimes also in vitro studies) use higher hexanediol concentrations than tested here. I think testing the behavior of these alkanediols at higher concentrations and their effect on protein phase separation is particularly interesting.

As we mention in our response in blue above, we do agree that the reviewer has a point. Now we have extended the linear range to 10%. We did this by adding new data using a longer version of FUS LC-RGG1 (FUS 1-284, see Wake et al *bioRxiv* 2024 where we characterize the phase separation of this piece of FUS), which phase separates with a lower saturation concentration and therefore enabled experiments at 10% hexanediol. We observed that the effects are linear in up to 10% hexanediol concentration. We also note that recent cellular studies demonstrate that optimal concentrations of 1,6 hexanediol are approximately 1.5% (Liu et al. 2021) and that higher values are not needed to see the effects, far within the range we are studying here.

Liu X, Jiang S, Ma L, Qu J, Zhao L, Zhu X, Ding J (2021) Time-dependent effect of 1,6-hexanediol on biomolecular condensates and 3D chromatin organization. *Genome Biol* 22: 230

Referee #2:

We appreciate the authors' efforts in revising the manuscript, including incorporating

additional data, updating figures, and revising the wording. However, several critical issues remain unresolved, which significantly diminish the likelihood of the paper being published in EMBO J. We summarize these shortcomings as follows:

a. Insufficient control experiments: the manuscript lacks essential control experiments, limiting the evidence to support the authors' conclusions. While the authors have made some attempts to address this concern, we still strongly recommend the author to include 2,5-HD as a negative control in the NMR experiments. This addition would provide stronger supports for the claim that 1,6-HD impairs FUS LC LLPS as described. By comparing the effects of 1,6-HD and 2,5-HD, it would help to exclude the possibility that the observed effects in Figures 3, 4, and 5 are merely noise rather than genuine disruptions of FUS LC LLPS by alkanediols.

We appreciate the reviewer's question here. In the current manuscript, we present the full ^1H ^{15}N chemical shift perturbations for all tested alkanediols. We also have dramatically revised the manuscript and removed some NMR experiments (such as the temperature coefficient experiment) that did not have a 2,5-HD control originally because we agreed with other reviewer concerns and instead we added orthogonal experiments (see below). Therefore, now the biochemical, biophysical, (new) computational, and NMR experiments have the 2,5-HD control as requested.

b. Inappropriate quantitative and measurement methods: the use of inappropriate quantitative and measurement methods probably results in weak or false conclusions. 1. Regarding the quantitative analysis of the NMR data in Figure 1 and Figure S1, we appreciate the authors' presentation of the effects of different diols on FUS LC LLPS and its NMR signals. However, we have concerns about the approach used, as the authors solely rely on the chemical shift perturbation (CSP) of ^{15}N while disregarding the CSP of ^1H , despite using ^1H -related measurements in Figure 5.

This inconsistency raises questions about potential bias and the possibility of distorted conclusions. Furthermore, it is evident that the observed ^1H CSP induced by the diols does not correspond to the extent of their LLPS inhibition. For instance, the ^1H CSP caused by 2.5%/5% 1,4-BD and 2,5-HD is considerably larger than that induced by 1,6-HD. This inconsistency undermines the credibility of the conclusions presented in Figure 1.

We agree with the reviewer that these issues were not sufficiently addressed in the previous version of the manuscript. The analysis of the ^1H data is more challenging given the higher sensitivity to chemical shift referencing. We have now extensively addressed the chemical shift referencing (see response to Reviewer 3 also) so the data are of high quality for ^1H and the added new analysis using ^1H CSPs induced by alkanediols shows that the ^1H CSPs are indeed also correlated to how potent the individual alkanediols disrupt phase separation, essentially the same as the ^{15}N chemical shift differences we presented before (Figure S8).

Also, we point out that although ^1H chemical shift perturbations are common to examine for discreet ligand binding events in folded proteins where the (relatively dilute) ligand is not reasonably expected to interact extensively with the DSS referencing compound, transient interactions of a very high concentration of a co-solvent may affect the chemical environment of the referencing agent as well in ways that are not easily predictable. In Figure S8, we directly show that despite these challenges the ^1H chemical shift perturbations confirm the data we showed previously for ^{15}N shift perturbations.

We emphasize that we have now also conducted and included several significant additional experiments assessing the function of hexanediols and their differences using many new orthogonal approaches.

2. The quality of the data in Figure 2 does not seem to have significantly improved. The authors state that phase separation is modestly decreased but still present even with the addition of aliphatic alcohols, as indicated by high turbidity values that decrease over time due to droplet settling. In Figure 2B, OD600nm measurements are used. However, it is important to note that the treatment with different diols results in a significant difference in the size and size distribution of FUS-RNA droplets. To accurately quantify the extent of FUS-RNA LLPS in Figure 2, a more precise method similar to that used in Figure 1B and 1D should be employed. These figures demonstrate the proportion of the supernatant to the total protein content, providing a more accurate measure. Considering the complications caused by the presence of RNA, we recommend employing a high-accuracy and similar quantitative method, such as SDS-PAGE grayscale quantitative analysis for protein concentration in the supernatant, in Figure 2B. If the construction of MBP-FUS poses challenges due to the overlap of MBP and FUS in the gel, we suggest using purified FUS-EGFP as an alternative (PMID: 29677514).

We appreciate the reviewer's critique regarding improving the quality of figure 2. We have now added all new data for Figure 2. We have conducted new microscopy experiments (Figure 2A) and have quantified the droplets via fluorescence microscopy for a straightforward quantitative comparison (Figure 2B). We have performed quantitative turbidity and biochemical partitioning assays that are fully consistent (Figure 2C). We have used FUS RGG3 with several different RNAs and several different RNA concentrations without and with 1,6-HD to test our conclusions. We also compare the effect of 1,6-HD on the phase separation of mixtures of polyD and polyK (see above) to test the effect of 1,6-HD on a system that is well established to phase separate via charged interactions.

c. Certain conclusions exhibit contradictions and suffer from a lack of persuasive argumentation process and logical reasoning.

1. The conclusion that alkanediols alter the solvation of FUS LC and may disrupt hydrophobic interactions and hydrogen bonding lacks detailed descriptions and contains confusing points.

The authors attempt to emphasize that popular simplistic conceptions of alkanediols

interacting directly with hydrophobic groups as "minidetergents" and the notion that distinct alkanediol isomers have qualitatively different effects are not true. However, they also mention that alkanediols disrupt hydrophobic interactions in this section, as well as in the abstract and discussion. It is unclear how alkanediols can disrupt hydrophobic interactions without directly interacting with hydrophobic groups.

We appreciate the reviewer's critique. To address the mechanistic questions head on and help move the field beyond "popular simplistic conceptions", we have added direct insight – 1) direct measurements of the hydrophobicity (water-octanol partitioning, logP) of the alkanediols – which to our knowledge has never been done and is an important contribution to the field, 2) testing and comparing the effect of single alcohols (ethanol and propanol) to hexanediols, 3) investigation of hexanediol and other alcohols' impact on folded protein stability, and 4) atomistic simulation to probe the contacts between HD and the protein. We have also thoroughly revised the abstract, results, and discussion in manuscript to better explain and reflect our findings showing that hydrophobicity and isomer geometry play important roles in alkanediol action, and how they do so.

Furthermore, the statement in the abstract that "the hydrophobic interactions mediating SYGQ LC domain LLPS are perturbed differently by each diol" lacks experimental data supporting this conclusion. This conclusion may be related to the predicted hydrophobicity data in Figure S1, however, the presented data in Figure S1 does not provide compelling evidence to support this conclusion, as the predicted results lack substantial substantiation.

We agree this was unclear and have revised the abstract. We have now experimentally measured the water-octanol partitioning coefficient logP as one measure of hydrophobicity. We find it is quite different from the predicted values we and others have previously referenced, hence these data are important for the field. See above regarding how we have conducted additional experiments to probe the connection between alkanediol hydrophobicity, geometry, and impact on phase separation.

2. The authors state that interactions between 1,6-hexanediol (1,6-HD) and FUS LC are relatively weak and non-specific. However, it is important to note that the observation of significant CSP in 13CO under 1,6-HD indicates some level of interaction between 1,6-HD and FUS LC. This raises a valid question about the authors' assertion of weak interactions between 1,6-HD and FUS LC.

This is a good point, and the nature of the direct interactions between FUS LC and 1,6-HD are challenging to probe with structural detail due to their transient nature. In addition to extensive additional experiments, we have studied the interactions using molecular simulation, which provided us with evidence for direct interactions between 1,6-HD and FUS LC. By comparing the effects of 1,6-HD and 2,5-HD in simulations to the experiments and by demonstrating that the simulations reproduce experimental properties of 1,6-HD and 2,5-HD solutions, we can provide direct and validated insight into the molecular features of this interaction that is at the cutting edge of what is feasible.

3. The authors state that they studied the capacity of condensate-modifying agents to inhibit phase separation of FUS in the presence of RNA, aiming to model protein-RNA interactions that contribute to biomolecular condensates formed in cells. However, it is worth noting that 1,6-HD has been shown to efficiently dissolve many RNA-containing granules, such as stress granules, where RNA-protein interactions are crucial for their assembly (PMID: 32302571). Furthermore, studies have reported the ability of 3% 1,6-HD to melt FUS granules in cells (PMID: 34746121). These findings are in contrast to the observation in Figure 2A, where FUS-RNA droplets were not melted by 1,6-HD. This discrepancy suggests that the phenomenon observed in the experimental setup lacks representative significance. It is possible that the use of an artificial system with a single RNA species and excessive RNA may contribute to these observations.

The reviewer makes several good points that we sought to address carefully with additional experiments and changes to the results section and discussion.

We have many conducted further experiments with other types of RNA (polyU, polyA, and torula yeast total RNA) and FUS, various concentrations of RNA, as well as an additional/orthogonal phase separating mixture that is known to be mediated in large part by charge-charge interactions (polyK+polyD peptides). Our data show that phase separation of FUS RGG3 with RNA and polyK with polyD (oppositely charged peptides) are not disrupted by 1,6-HD.

Regarding the potential discrepancy between our *in vitro* observations and the in-cell studies reported in the literature, while interactions between RNA and disordered RGG domains in these biomolecular condensates found in cells may not be affected by 1,6-HD, other types of interactions contributing to phase separation are indeed disrupted. (In other words, just because cellular granules contain RNA does not mean that they are stabilized by RNA in the way that mixtures of FUS disordered domains and RNA made in a test tube.) Specifically, in the case of full-length FUS with RNA (Figure 2A), we may infer that heterotypic RGG-RNA interactions are not affected by 1,6-HD. But, we do observe a decrease in overall turbidity of FUS-full-length with hexanediol addition, likely due to the disruption of homotypic FUS-FUS interactions. Additionally, we also clarify in the discussion that we are not looking at the effect of specific or non-specific RNAs that interact with the folded domains of FUS. Indeed, these interactions may include hydrophobic contacts that are disrupted by 1,6-HD (much like the effect of 1,6-HD on lysozyme stability that we show here for the first time in this revised version). We have included additional text discussing these points extensively and urging caution in using hexanediol sensitivity to infer the mechanism of dissolution in cells.

“As a corollary, the large number of membraneless organelles found to be susceptible to 1,6-hexanediol may also therefore imply that these are primarily stabilized by hydrophobic and not charge-charge interactions. However, these observations that interactions between disordered RGG peptides and unstructured RNA or total RNA extracts are not disrupted by 1,6-HD should not be taken to mean that no cellular RNA-protein interactions are disrupted by 1,6-HD. Indeed, FUS and many other phase-

separating RNA-binding proteins contain RRM and other domains that interact with single stranded RNA not primarily via the charged backbone of RNA but rather via the bases using hydrophobic/aromatic residues (Loughlin *et al.*, 2019), which may be disrupted by 1,6-HD. Thus, the effectiveness of hexanediol is context-dependent and results of cellular experiments adding HD should be interpreted with careful consideration of the condensate composition (e.g. sequence characteristics and nucleic acid partitioning).”

4. The observation that the Q to G mutant of FUS does not impair FUS phase separation (PMID: 29961577) is an important point to consider. It suggests that the sidechain of Q may not play a significant role in FUS phase separation. This raises doubts about whether the hydrogen bonding observed in the sidechain CO of Q under 1,6-HD, as shown in Figure 3, reflects the mechanism by which 1,6-HD disrupts FUS phase separation. It is necessary for the authors to address the discrepancy and provide a clear explanation. They should discuss how the observed hydrogen bonding in the sidechain CO of Q under 1,6-HD aligns with the understanding that the Q to G mutant does not impair FUS phase separation.

We agree with the reviewer’s concerns and have removed this section of text and figures due to a lack of concrete evidence. We have focused the manuscript on the hydrophobic nature of the alkanediols instead, and used validated molecular simulations to analyze the contacts.

Referee #3:

Like the previous version, this version of the manuscript is still highly descriptive (spectra and microscopy images are shown and it is described what happens). However, the title suggest “Molecular insights “, that I did not find in the manuscript. The abstract and conclusions claim that hexandiol induces changes in the protein solvation, but this is only suggested indirectly by the data. In general the data is not analyzed very critically and often the conclusion cannot be unquestionably drawn. I thus cannot support publication of this manuscript, despite the high interest of the topic.

We appreciate and have taken seriously the reviewer’s direct and extensive critique. By adding many new experiments, changing the main message, and completely rewriting most of the manuscript, we hope that we have addressed the reviewer’s concerns to provide a study that goes beyond description and gets to mechanistic conclusions through critical analyses of complementary data.

Major remarks:

I see no correlation between Figures 1C and D. This point was raised by all reviewers, and despite the response that the microscopy images are only qualitatively, the feeling remains that the two approaches (microscopy, saturation concentration) are reporting on different things.

We appreciate the reviewer's concerns and we agree that the images were not helpful. A challenge we encountered while attempting to quantifying the previous DIC images is that the alkanediol additives can cause FUS droplets to wet the glass slide differentially (depending on surface treatments), resulting in irregularly shaped droplets. Combined with low contrast of DIC imaging, this made quantification difficult. To solve this, we conducted entirely new set of experiments and have added image quantification by fluorescence microscopy, which provided enhanced contrast. We also showed that quantification by fluorescence microscopy precisely matches our previous results measuring the concentration of protein remaining in the supernatant after centrifugation of droplets (Figure 1F).

In Figure 1E it should be clearly stated that one dot is one residue (or more accurately one nitrogen spin).

We have now moved this analysis to Figure 3E and addressed this in the text – one “dot” represents one residue's chemical shift deviation.

"In other words, the sequence-specific effects are highly similar for all diols tested, just with a different magnitude, suggesting that all alkanediols affect phase separation by a similar mechanism." should read something like: "In other words, the sequence-specific effects are highly similar for all diols tested, just with a different magnitude, suggesting that all alkanediols interact with FUS in a similar manner, albeit with different affinities" (This then might imply that they affect phase separation similarly).

We have changed the wording here extensively due to many changes in the manuscript. The wording is changed as follows:

“In summary, we observe that the series of alkanediols show the same pattern of effects on FUS LC with different magnitudes that correlate with the extent of impact on phase separation. This observation suggests that all alkanediols, regardless of structure, have similar types of interactions with FUS. Hence, we do not find evidence here for unique interactions only possible by particular alkanediol isomers, though below we probe the factors that govern the extent of impact on phase separation.”

Figure 1F and S1F. It is rather arbitrary to call correlations Strong or Moderate based on the PCC. I would use the following criteria: $r > 0.8$ is a strong correlation, $0.5 < r < 0.8$ is a moderate correlation and $r < 0.5$ is a weak correlation. In that context all correlations are moderate. Bottom line: there is no reason to suggest that the correlations in figure 1F are strong and that the one in Figure S1F is only moderate. They are all in the same range, with the correlation in Figure S1E being a bit weaker. The discrimination between strong and moderate is even less relevant if one considers that the $\log(P)$ values are only predictions (thus likely making any correlation weaker).

We appreciate this point and have extensively changed and added to this section. We have added experimentally measured $\log P$ values to be able to make strong conclusions about the hydrophobicity. Regarding the quantitative correlations, the

correlation for 1% HD is strong by this definition (>0.8). The 2% is essentially the same (0.77). We have added new data measuring the logP values of the alkanediols experimentally using NMR, which are an important contribution. The new correlation using experimentally measured logP has significantly improved. We also observe that alkanediols with terminal hydroxyl groups (1,6-HD, 1,5-PD, 1,4-BD) show an extremely high correlation between hydrophobicity and impact on phase separation, suggesting that hydrophobicity distinguishes these linear alkanediols. We explicitly probe the difference between 2,5-HD and 1,6-HD that is not fully accounted for by hydrophobicity using both molecular simulation in combination with NMR to observe how the geometry contributes to the interactions.

" Importantly, we find that phase separation is modestly decreased but still present even with addition of aliphatic alcohols (Fig 2A)" I do not see in Fig 2A (row + RNA) how the phase separation is changed by the additives. To me all panels look equal with respect to the degree of droplet formation. The 1,2-HD should be removed from the panels in Figure 2A and B for the FL protein as there is still MBP attached (I assume), preventing a comparison. Also, the panels in the top row are missing, which prevents me from judging the data.

We agree with the reviewer's concern and we have fixed the issue by using a different experimental strategy. Instead of performing the MBP tag cleavage in the presence of aliphatic alcohol additives, we now carry out the TEV cleavage in the absence of these additives. After cleavage, we add RNA and alkanediols to the cleaved FUS solution and then quantify the phase separation using fluorescence microscopy. This revised procedure has produced more consistent results and eliminated the interference caused by TEV cleavage in the presence of alkanediols. We have updated Figures 2A and 2B accordingly. We have left the evidence for 1,2-HD disruption of TEV cleavage (but not the other alkanediols) as helpful information for the community.

For all replicates in all experiments, please clearly indicate if these are technical (doing the same measurements with the same batch of protein) or biological replicates (with differently purified batches of protein). As FUS seems hard to purify there could be larger variations between different batches.

These are technical replicates, but since our last revision, we have done biological replicates and saw consistent results between two different protein preps for all FUS experiments. With 10+ years of experience purifying FUS and FUS constructs, our purification methods yield reproducible outcomes with minimal variation between different batches.

" and phase separation is not markedly decreased, neither immediately (Fig 2D) nor over time (Appendix Fig S4)." The turbidity does change over time (Fig. 2D), I assume that the authors mean that the changes in turbidity over time are independent of the presence of the alkanediols.

Yes, however we have taken the reviewer's critique seriously and have directly measured the concentration by UV-Vis that corroborate additional turbidity data, which we instead present as initial turbidity (we remove the analysis over time as we do not think it adds anything to the presentation).

Figure 3A and B. Could the authors analyze the CSPs by combining the ^{15}N and ^1H shifts, this would likely give more accurate data and also a more common way of analyzing interactions.

After performing careful referencing with DSS, we have now added the parallel analysis for ^1H amide chemical shift perturbations (Figure S8) and shown that they yield correlations that are as good as or even higher than the ^{15}N chemical shift perturbations.

"though the distribution of shifts for each residue type is large, especially for the most common residue types (serine, glycine, glutamine, and tyrosine)." Why would the CSPs vary more for the more abundant residues? Is it not just because there are more data-points that there appears to be a larger spread? For the small sample size for many residue-types the spread is not a relevant measure for anything.

We have revised the text to remove statements regarding the distribution of CSPs that we agree are confusing and not sufficiently supported.

The authors write that Ala and Thr show the largest CSPs. This is only the case in the nitrogen dimension. However, what the mechanistic basis for this could be. Do those residues interact preferably with 1,6-HD or is this just coincidence, or are those large CSPs just "outliers" and not representative of any general amino-acid specific trend?

We agree and we have removed analysis where we have too few observations to make claims.

" though a similarly large distribution of shifts for glycine, suggesting that 1,6-hexanediol similarly influences the chemical environment of highly distinct sequences" I strongly disagree with this conclusion, unless the authors have an explanation why a similar (random) distribution of the size of CSPs would correlate with similarities in interactions. I would actually conclude from the data that all (glycine) residues interact significantly differently with the 1,6-HD, maybe in a sequence specific manner.

The point we were attempting to make was that glycines in one sequence that phase separates and glycines in another sequence that does not readily phase separate are all affected. We have thoroughly altered this section to revise this conclusion, which now reads "We observed a similar magnitude of CSPs for FUS RGG3 (Figure S5) as for FUS LC, suggesting that 1,6-HD similarly influences the chemical environment of distinct sequences. Therefore, our CSP analysis suggests that, despite certain residues showing larger CSPs, no clear pattern of interaction emerges."

"Supporting this hypothesis, 1,6-HD had little effect on the molecular motions of FUS LC 12E variant, presumably because the introduction of 12 phosphomimetic mutations has already disrupted the molecular contacts upon which 1,6-HD acts (Appendix Fig S6)". I am not sure if the differences in the changes in R1 and R2 (Fig 3E and Fig S6) are actually as large as the authors suggest. The y-scale of the R2 plot is different and to me the change in R2 is more or less between 0.5 and 1 Hz in both cases. One could just plot the change in R2 in both cases to see if the differences are really significant.

This is an excellent and straightforward suggestion. We have remeasured these values (they remained largely the same) to improve signal to noise to make the comparison easier and replotted to make the point clear. As we show in the new panel in Figure S6 (B and C), the 1,6-HD-induced reduction in R_2 values (ΔR_2) are significantly reduced for the 12E variant when compared to WT FUS LC.

Figures 4E and F are obsolete as the CSPs are very small and cannot be determined with any kind of accuracy as they are a superposition of many different resonances from many residues. Also, why are some bars missing (AHB, QHA, SHA, THB but also ACB, TCB)?

We agree that these overlapped and hence the data may not be reliable. For this reason, we have removed this analysis.

"These shifts are modest and not markedly correlated with the backbone ^{15}N perturbations". They are uncorrelated.

We agree, though we have updated this section extensively so this section is no longer present.

"...and similar for side chain and main chain (i.e. α carbon positions)." . The plot (that is obsolete, see above) does show a variation by an order of magnitude, so the side and main-chain CSPs that are presented here are definitely not similar.

As above, we have removed this analysis.

"Classifying the CO chemical shift difference by residue type (Fig 4D) shows that polar residues such as serines, threonines, glutamines but also glycines show a wide range of chemical shift perturbations." As for the nitrogen and proton CSPs, the largest spread is observed for the most common residues, which is just as one would expect statistically (as many residue types are only present with very few residues). Because of the limited number of probes for some residue types the spread (standard deviation) is highly uncertain and not a valid measure.

We agree with the reviewer that this section was not well founded. We have removed residue-specific analysis of the ^{13}C O chemical shift perturbations (but have retained the data in the Appendix Figure for completeness) and have revised the text to read "Consistent with the picture from the amide positions, we also observed small CSPs

without clear residue-type specificity for backbone carbonyl (^{13}CO) positions in the presence of 1,6-HD (Figure S4). ”

" Furthermore, given that the largest differences are observed for backbone and sidechain positions that form hydrogen bonds, " → I cannot see what this is based on, the authors contradict themselves. The glutamine side-chains shift the least in ^{15}N (compared to the bb) and the most in ^{13}CO (compared to the bb). Both the NH_2 and the CO in the side-chain can hydrogen bond. Also, there are many examples for bb CO, N and H resonances that do not shift, or only very little. Those are very well able to be involved in hydrogen bonds. So, I see no link between CSPs and hydrogen bonding abilities and I disagree with the conclusion that "alkanediols ... disrupt hydrophobic interactions and hydrogen bonding."

We agree that the presentation is confusing and we have removed this claim.

Fig 5A and B. The errors (standard deviations) in the slopes are so large that it is hard to see if the change in the slope upon addition of 1,6-HD is significant at all. The addition of the 1,6-HD will cause changes in many things, including viscosity. Maybe the observed (very small) changes in the slopes are just due to changes in solvent properties and completely unrelated to hydrogen bonding effects. I do not trust the conclusion unless proper controls have been made, e.g. with isolated amino acids in the presence and absence of 1,6-HD.

We agree with the reviewer and have removed this entire section as it causes confusion and is not sufficiently well founded.

" Taken together, these data suggest that condensate-modifying molecules like hexanediol might enhance the solvation of the protein backbone leading to dissolution of the condensates." This is really far fetched and highly speculative.

We agree that this statement is not supported and also not the main point of the findings here. Therefore, we have removed this section from the manuscript to avoid any confusion or speculative statements. What we do focus on are how alkanediol hydrophobicity and isomer geometry contribute to their impact on phase separation, and we use molecular simulation to help visualize the interactions that lead to this effect and probe how the different isomers interact with the protein.

Minor remarks:

" the observed chemical shift perturbations are small compared to what would be expected for a tight-binding molecule" → the extend of CSPs is not correlated with the affinity.

We have now rephrased this to say:

“While the CSPs are smaller than those typically associated with strong binding interactions, they are larger than those observed for the weak interactions of

karyopherin- β 2 with FUS LC (Yoshizawa et al, 2018). The observed CSPs are also not uniform across the sequence. Binning backbone CSPs according to amino acid types showed little to no systematic residue-type specific effects and a broad distribution of changes within each type (Figure 3C)."

"Surprisingly, many of the backbone ^{13}C O chemical shifts are larger than " \rightarrow ... ^{13}C O chemical shift changes ...

We have revised this section to remove these claims.

" This large shift for glutamine side chain is unique given that the ^{15}N side chain of glutamine is shifted less (~ 0.04 ppm) (Fig 3F) than the largest backbone ^{15}N NH $_2$ (~ 0.20 ppm). " \rightarrow I assume "...than the largest backbone ^{15}N N-H (~ 0.20 ppm) "

This paragraph has been revised; the statement has been removed.

Figure S3, please indicate that the weak band at 25 kDa is the TEV protease (in case that is true).

We have added labels for TEV and other bands

Supplementary figures come in the wrong order, please correct. Also the main text figures come in different order in the text and in the panels.

We have fixed this issue.

Figure S8A: why are there 2 spectra at the second spectrum (a blue and a cyan) on top of another.

We have removed this section.

Figure S8B: why are the amide signals doublets. Did the authors fail to decouple properly?

These are high resolution spectra that resolve the $^3J_{\text{HN-HA}}$ coupling (3 to 10 Hz) for non-proline residues. Glycines in disordered proteins appear as a triplet due to coupling to both HAs. But, we have removed this section.

Figure S8D: "The high R^2 values for every position (>0.97 in all cases) suggest that the values extracted represent the true temperature coefficients." The high correlation coefficients only reveal that the resonance frequencies shift linear with temperature.

We have removed this section.

Remarks regarding the rebuttal letter:

"We also attempted to assess the impact on folded proteins using the RRM of

hnRNPA2 as a model folded protein. However we found that even residues in the core shifted, likely due to slight changes in the structure transmitted from the surface - proteins are not static rigid assemblies - and the specific arrangement is perturbed by high percentages of co-solvents." → can this not be seen as an argument that supports a situation where hexandiols have a general influence on all resonances, independent of hydrogen bonding? In other words, does that not imply that the shifts are mainly due to changes in solvent properties and that CSPs are independent of changes in hydrogen bonding.

We agree that this may be true. We have removed discussions regarding hydrogen bonding due to lack of strong evidence.

"We do not see evidence of phase separation of the RNA at these conditions in these concentrations, though it is certainly possible." The authors should show these data in the supplement.

We have added new microscopy images showing that RNA do not form droplets without the protein (Figure S3).

"We agree that the spreads are large and we have now emphasized in the text that the variability is one of the findings we wish to highlight." The variability is not a relevant measure in my eyes (see major remarks). The large variability is correlated with residue abundance. And a large variability in the CSPs just means that the individual residues of a certain residue type react slightly different with the additives.

We agree that variability shows that not all residue positions of the same type are impacted the same. We have removed this.

"If the residue that is perturbed by direct binding is showing something at the 1H/15N position, it is reasonable to assume that it would show up at 13C backbone position. Therefore the lack of correlation again suggests that the chemical shifts are changed due to changes of hydrogen bonding and water, not from side chain (hydrophobic) contacts. We have clarified this in the text." In case the hydrogen bonding of water with the backbone changes the 1H-15N shifts will change. That is clear. But, as the authors state (reasonably assume, I agree), this should then also be observed for the 13C backbone. Why is that not the case here then. Hydrogen bonding changes should be picked up by 13C, 1H and 15N in the backbone.

We agree with this assessment. We have removed this section from the manuscript.

“ Fig 5. Did the authors take into account that different amino acids behave differently with temperature and have different pKa values? The analysis here seems quite limited.’

response: It is not clear what is meant here by the reviewer regarding different behavior and pKa values. There are no histidine residues the only residues with pKa near the

experimental pH. The only charged residues are two Asp residues, which also are far from the pKa." → The question is quite simple: do the amide shifts from different aminoacids react on temperature changes the same, or are there any indication that the temperature dependence of the chemical shifts are residue-type specific? In that case the analysis that the authors performed is an over simplification.

We appreciate the critique and have removed this section.

"We rule out tyrosine/aromatic-specific interaction of hexanediol and it is consistent with our previous findings that many interactions modes and residue types contribute phase separation of these IDRs." → I agree with that. That something does not play a role is an important finding, but molecular insights (title) should maybe reveal the mechanistic basis of which (specific) interaction do play a key role. This latter point is unfortunately too weak.

We have significantly revised our manuscript to include new experimental and simulation data that directly probe the impact of alkanediols on the molecular mechanisms underlying FUS phase separation. These additions have thoroughly revised and strengthened our findings and provide a clearer mechanistic view for the interactions involved.

Dear Nick,

Thank you again for submitting your revised manuscript (EMBOJ-2022-111817R1) to The EMBO Journal for our consideration. It has now been seen by two of the original referees who previously assessed the initial version of your manuscript (their comments are included below). I am glad to say that both referees are very satisfied with the revised manuscript, point out that the previously raised concerns of all three reviewers have been successfully addressed, and now support the publication of this interesting manuscript in The EMBO Journal without any further comments. Given this input, I am glad to let you know that your manuscript has in principle been accepted for publication in our journal.

Before we can proceed with the production of your article, there are some editorial requests/formatting changes that we need from you to address in the final version of your manuscript:

- Please note that the Figures should be removed from the main manuscript file; only their legends should remain in the manuscript (after the References list).
- We noticed that the first four co-authors have been listed as "co-first". Although this is possible if these co-authors contributed equally, it is rather unusual at our journal, and we would thus like to ask you to consider again whether their contributions justify their "co-first" listing. Please also note that we use the CRediT system to specify the contributions of all co-authors in our journal's submission system (see below for more information).
- Our previous messages to the co-author Theodora Myrto Perdikari, when we tried to contact her at "theodora_myrto_perdikari@brown.edu", bounced back. Could you please ask her to update her profile in our manuscript tracking system with a valid e-mail address, or send us her current e-mail address so that we can update her profile?
- All relevant funding information should be provided in the Acknowledgements section of the manuscript and also entered in our manuscript tracking system (eJP) during resubmission of the manuscript. The following item is currently saved in eJP but missing in the manuscript: T32GM136566. The following information is missing in eJP: Pape Adams Postdoctoral Award from the Carney Institute for Brain Science at Brown University and a Milton Safenowitz Postdoctoral Fellowship from the ALS Association; NIGMS training grant at Brown University (T32GM139793); NIGMS training grant at Brown University (T32GM007601).
- Please provide a list of up to 5 relevant keywords after the Abstract of your revised manuscript.
- Before submitting your revision, primary datasets (and computer code, where appropriate) produced in this study need to be deposited in appropriate public databases (see <https://www.embopress.org/page/journal/14602075/authorguide#dataavailability>). The accession numbers, databases, and the specific URLs (links) should be listed in a formal "Data availability" section (placed after Methods). All links should resolve to a webpage where the data can be accessed. In case you have no data that require deposition in a public database, please state so instead of referring to the database: "Our study includes no data deposited in public repositories." under the heading "Data availability".
- Please change the heading of your conflict-of-interest statement to "Disclosure and competing interests statement".
- The author contributions statement should be removed from the manuscript file. Instead, we use CRediT to specify the contributions of each author in the journal submission system. Please feel free to use the free text box to provide more detailed descriptions during submission. See also our guide to authors for more information: <https://www.embopress.org/page/journal/14602075/authorguide#authorshipguidelines>.
- The callouts for Appendix Figures and Tables should be updated to "Appendix Figure S#" and "Appendix Table S#" throughout the manuscript.
- Your Author Checklist appears to have been filled in only partially. Please make sure that all relevant questions are completed (e.g., in the section "Experimental study design and statistics"), and that the sections of the manuscript where the information is available are provided in the last column for all relevant questions (e.g. in the section "Sample definition and in-laboratory replication").
- Materials and methods need to be described in the manuscript using our Structured Methods format, which is now required for all research articles. According to this format, the Methods section includes a single "Reagents and Tools Table" -listing key reagents, experimental models, software and relevant equipment including their sources and relevant identifiers- followed by a "Methods and Protocols" section describing the methods. Please download and fill our Reagents and Tools Table template (.docx), which you can find in our author guide: <https://www.embopress.org/page/journal/14602075/authorguide#structuredmethods>. When submitting your revised manuscript, please do not include the Reagents and Tools Table in the Methods section of the manuscript but upload it as a separate file

choosing the file type "Reagent Table".

- At EMBO Press we now ask authors to provide source data for the main manuscript Figures. Our source data coordinator will contact you to discuss which Figure panels we would need source data for and will also provide you with helpful tips on how to upload and organize the files.

- Please re-order the manuscript sections as follows: Title page - Abstract & Keywords - Introduction - Results - Discussion - Methods - Data Availability - Acknowledgments - Disclosure Statement and Competing Interests - References - Figure Legends - (Main Tables with legends, if there are any) - Expanded View Figure Legends (if there are any).

- During our routine pre-acceptance checks, our data editors have raised a number of queries regarding figures, data, and legends. You can find them as comments in the attached Word file (please note that this is a copy of the previous version of your manuscript). You are kindly requested to address all queries completely in the final version of your manuscript that you will resubmit to our online system.

Please also note that as part of the EMBO publications' Transparent Editorial Process, The EMBO Journal publishes online a Peer Review File along with each accepted manuscript. This File will be published in conjunction with your paper and will include the referee reports, your point-by-point response and all pertinent correspondence relating to the manuscript. You can opt out of this by letting the editorial office know (contact@embojournal.org). If you do opt out, the Peer Review File link will point to the following statement: "No Peer Review File is available with this article, as the authors have chosen not to make the review process public in this case."

We look forward to seeing a final version of your manuscript as soon as possible. Please let us know if you have any questions and use this link to submit your revision: <https://emboj.msubmit.net/cgi-bin/main.plex>.

Best wishes,

Ioannis

Referee #1:

I applaud the authors for this lengthy and in-depth revision of their work and the thorough consideration of all the Referees' critique. My concerns are fully addressed in this version with additional experiments and fully updated Figures. I also see that the authors have responded appropriately to the concerns of all other referees, in most cases with the addition of the exact experiment or editorial revision asked. This includes the particularly critical reviewer #3. In my view the current manuscript is now suitable for publication in EMBO Journal.

Referee #2:

The authors addressed my concerns with additional experiments. I don't have further concern.

All editorial and formatting issues were resolved by the authors.

Dear Nick,

Congratulations on an excellent work! I am very pleased to inform you that your manuscript has now been accepted for publication in The EMBO Journal. Thank you very much for your comprehensive and thorough revision addressing the initially raised concerns of the referees, and also for addressing all editorial and formatting requests.

If you have any questions, please do not hesitate to contact the Editorial Office. Thank you for your contribution to The EMBO Journal. Working with you has been a pleasure!

Best regards,

Ioannis
